# Statistical Analysis of an Adversarial Bayesian Weak Supervision Method

**Steven An**
Computer Science Department
University of California, San Diego
La Jolla, CA 92037
`sla001@ucsd.edu`

## Abstract

Programmatic Weak Supervision (PWS) aims to reduce the cost of constructing large high quality labeled datasets often used in training modern machine learning models. A major component of the PWS pipeline is the *label model*, which amalgamates predictions from multiple noisy weak supervision sources, i.e. *labeling functions* (LFs), to label datapoints. While most label models are either probabilistic or adversarial, a recently proposed label model achieves strong empirical performance without falling into either camp. That label model constructs a polytope of plausible labelings using the LF predictions and outputs the "center" of that polytope as its proposed labeling. In this paper, we attempt to theoretically study that strategy by proposing Bayesian Balsubramani-Freund (BBF), a label model that implicitly constructs a polytope of plausible labelings and selects a labeling in its interior. We show an assortment of statistical results for BBF: log-concavity of its posterior, its form of solution, consistency, and rates of convergence. Extensive experiments compare our proposed method against twelve baseline label models over eleven datasets. BBF compares favorably to other Bayesian label models and label models that don't use datapoint features – matching or exceeding their performance on eight out of eleven datasets.

## 1 Introduction

Large scale deep learning models often require a labeled training set of substantial size. However, hiring experts to label every training point is not only expensive, but time consuming. Programmatic Weak Supervision (PWS) aims to ameliorate this problem by cheaply generating labels. This is done via combination of multiple weak supervision sources, called *labeling functions* (LFs). LFs can be computer programs [Ratner et al., 2016], crowdsourcing workers, simple classifiers (like linear classifiers), etc. Since LFs are noisy and provide contradictory predictions on a datapoint, it is incumbent on the *label model* to combine the LF predictions to produce a single labeling of the data.

The construction of the label model serves as one of the main challenges in the PWS pipeline. Most label models are either probabilistic or adversarial. Probabilistic label models [Dawid and Skene, 1979, Ratner et al., 2016, Zhang et al., 2023] often specify a generative process for the labels/LF outputs and compute the posterior label distribution for each datapoint. Much work has been devoted to this approach and there are many examples of probabilistic label models that perform well empirically. Adversarial label models [Balsubramani and Freund, 2015a, Mazzetto et al., 2021b, Mazuelas et al., 2022] assume the existence of an adversary who assigns the labels, construct a polytope of plausible labelings, and then predict a minimax optimal labeling to guard against the worst case. While they come with theoretical guarantees, e.g. An and Dasgupta [2024], these label models require (small amounts of) labeled data to construct the polytope. Arachie and Huang [2019]

39th Conference on Neural Information Processing Systems (NeurIPS 2025).

propose a label model that is adversarial and unsupervised, though we are unaware of theoretical guarantees for their method. There are also label models that fall into neither camp, e.g. Hyper Label Model (HyperLM) [Wu et al., 2023], which constructs a polytope of labelings in an unsupervised fashion and then picks the labeling that is in the "center". While HyperLM is a strong performer in real-world settings, we are unaware of theoretical guarantees regarding its properties and performance. In this paper, we attempt to bridge the gap by proposing Bayesian Balsubramani-Freund (BBF), a Bayesian adversarial label model that is unsupervised, selects labelings like HyperLM, but also has theoretical guarantees such as consistency.

Say we have $N$ datapoints $X = \{x_1, \ldots, x_N\}$ where $x_i$'s label is $z_i$ in $\mathcal{Z} = \{1, \ldots, K\} = [K]$. We wish to infer the underlying conditional label probability $\eta(j \mid x) = \Pr(z = j \mid x), j \in \mathcal{Z}$, for each $x \in X$. If our task is to determine whether an email is spam, $\eta(j \mid x)$ might have large mass for a single $j$. On the other hand, if the question is whether someone will default on a loan, we may imagine $\eta(j \mid x)$ being closer to uniform due to inherent uncertainty. Over all labels and datapoints, we compactly write $\eta(j \mid x)$ as $\vec{\eta} := (\eta(j \mid x_i) : j \in [K], i \in [N]) \in \Delta_K^N$ where $\Delta_K$ is the set of distributions over $K$ elements and $\Delta_K^N$ is the concatenation of $N$ of those distributions. Our goal is to pick a distribution from $\Delta_K^N$ to approximate $\vec{\eta}$.

BBF approximates $\vec{\eta}$ by taking the mode of its posterior label distribution, which is itself specified by its generative process. First, BBF draws the distribution of labels on $X$, after which *all* datapoint labels are drawn simultaneously. Then, for each LF, its accuracy on $X$ is drawn, following which the predictions it makes (on $X$) are drawn simultaneously. That is, once the accuracy $b$ of LF $h$ on $X$ is drawn, any labeling of the $N$ datapoints that attains accuracy $b$ on $X$ is equally likely to be selected as $h$'s predictions. Compare this to a simplified version of other generative processes in the literature: once the datapoint labels and LF accuracies are drawn, whether an LF's prediction matches the label on a datapoint is a biased coin flip. Indeed, the frequentist versions of these models have a similar difference: Balsubramani-Freund (BF) [Balsubramani and Freund, 2015a] versus One-Coin Dawid-Skene [Li and Yu, 2014].

In this paper, we'll show that not only does BBF's prediction $\vec{g}^{bbf}$ have some nice properties, it has strong empirical performance.

1. (Log-Concavity of Posterior) Given a small modification, BBF's posterior label distribution is log-concave.
2. (Form of Solution) $\vec{g}^{bbf}$ belongs to the same exponential family $\mathcal{G}$ as the BF solutions, defined in terms of the LF outputs.
3. (Consistency) When the LF accuracy priors concentrate on the empirical accuracy, $\vec{g}^{bbf}$ converges to $\vec{g}^* \in \mathcal{G}$ that best approximates $\vec{\eta}$ in KL divergence.
4. (Rates of Convergence) We provide a bound for the rate at which $\vec{g}^{bbf}$ approaches $\vec{g}^*$.
5. (Empirical Results) Comparison of $\vec{g}^{bbf}$ against twelve baseline label models on eleven real datasets. Consistency on real datasets is also shown.

## 2 Related Work

The PWS pipeline ranges from the creation of the LFs, aggregation of their predictions via *label model*, to the training of an *end model* using the label model's labeling of the data. For a good survey that discusses all of these aspects, see Zhang et al. [2022]. Since we focus on the label model in this paper, we'll give a brief overview of them.

One line of work involving probabilistic label models starts from [Dawid and Skene, 1979], where each LF is modeled as a confusion matrix and LF predictions are independent of each other given the datapoint label. The Dawid-Skene model has been lifted into the Bayesian setting and repeatedly generalized: [Kim and Ghahramani, 2012], [Li et al., 2019], [Zhang et al., 2023]. On the frequentist front, Ratner et al. [2016] builds upon Dawid and Skene's model by representing dependencies between LFs in a factor graph. Other probabilistic label models have been studied too, e.g. [Fu et al., 2020, Kuang et al., 2022, Shin et al., 2022].

Adversarial labels models can take the form of minimax games where we the learner attempt to minimize the loss of our prediction (i.e. the label model's aggregated labeling) while an adversary tries to maximize our loss. A big part of these models is representing the subset of feasible labels,

i.e. a set of labels for which the true underlying conditional label probabilities $\eta$ lie. This was studied by Balsubramani and Freund [2015a,b] for 0-1 loss, [An and Dasgupta, 2024] for log loss, where the set of feasible labels is gotten by bounding the accuracies of individual LFs. Arachie and Huang [2021] estimate the LF accuracies and then try to find the minimum norm labeling that induces those accuracies from the LF predictions. Alternate formulations of the minimax game can be found. In [Mazuelas et al., 2020], uncertainty sets are considered, allowing for more general constraints on the set of feasible labels. Mazuelas et al. [2022] consider generalized entropy as the loss function for the minimax game, which includes 0-1 and log loss as special cases. Mazuelas et al. [2020, 2022], and An and Dasgupta [2024], provide convergence results in terms of how well the LF accuracies are estimated, but only the latter show consistency. Mazzetto et al. [2021a] provide a generalization bound, though they require datapoints be drawn i.i.d. In contrast, the result of Arachie and Huang [2021] result requires that one know the actual LF accuracies on the unlabeled datapoints and for there to be an increasing number of LFs to drive down the prediction error. In this paper, we consider a fixed set of LFs and our analysis does not require the actual LF accuracies to be known.

Label models that are neural networks have also been proposed. For example, Wu et al. [2023] trained a graph NN that implicitly constructs a polytope of labelings using LF outputs and returns the "center" as the aggregated labeling. On the other hand, Cachay et al. [2021] and Ren et al. [2020] incorporate datapoint features along with the usual LF predictions in computing the aggregated labeling. These last two works involve *joint models*, a unified model that does the work of both label and end models.

## 3 Preliminaries

We now provide the formal setup for the problem and briefly discuss HyperLM and BF. We follow the notation of Kim and Ghahramani [2012], Li et al. [2019], Zhang et al. [2023].

We desire to label $N$ unlabeled datapoints $X = \{x_1, \ldots, x_N\}$ by combining the predictions from $W$ LFs, $h^{(1)}, \ldots, h^{(W)}$. These LFs can predict one of the $K$ labels in $\mathcal{Z} = \{1, \ldots K\}$ or abstain: i.e. $h^{(w)} : X \to \mathcal{Z} \cup \{?\}$, "?" denoting abstention. In particular, our analysis makes no assumptions about the LFs, e.g. no restrictions on their multiplicity, no requirements about their independence given certain events, etc. We'll abbreviate $h^{(w)}(x_i)$ by $y_{iw}$. For every datapoint $x_i$, we wish to infer the ($K$ dimensional) underlying conditional label distribution $\vec{\eta}_i \in \Delta_K$. $\eta_{ij} = \Pr(z_i = j \mid x_i)$, the true probability of class $j$ for $x_i$. Following Mazuelas et al. [2020],

$$\vec{\eta} := (\vec{\eta}_1, \ldots, \vec{\eta}_N) \in \Delta_K^N,$$

a vector of length $NK$. More notation will be introduced when necessary.

### 3.1 Hyper Label Model

To select its labeling, HyperLM [Wu et al., 2023] first constructs a finite set $U$ of plausible labelings. Then, the labeling whose squared Euclidean distance is minimized with respect to all elements of $U$ is selected and used. Formally, suppose for this section only that $\mathcal{Z} = \{\pm 1\}$. $U$ is a subset of all possible hard labels $\mathcal{Z}^N$ such that the there are at least $W/2$ LFs that are better than random on each class. That is, for class $+1$ (and $-1$ respectively), there are at least $W/2$ LFs that get $> 50\%$ of their predictions right when they predict $+1$ ($-1$ respectively). Then, the HyperLM labeling is

$$\vec{g}^{hlm} = \frac{1}{|U|} \sum_{\vec{z} \in U} \vec{z} = \argmin_{\vec{g} \in [-1,1]^N} \frac{1}{|U|} \sum_{\vec{z} \in U} ||\vec{z} - \vec{g}||_2^2.$$

By taking the convex hull of $U$, one can get a polytope of plausible labelings. Then, $\vec{g}^{hlm}$ can be thought of as being the "center". To get $\vec{g}^{hlm}$ in practice, the above authors train a graph neural network that takes LF predictions as input. HyperLM is empirically effective, e.g. see [Zhang et al., 2023], but Wu et al. [2023] did not present theoretical results about it's performance or other properties like consistency. We hope to better understand HyperLM's strategy in this paper.

### 3.2 Balsubramani-Freund with Log-Loss

The Balsubramani-Freund model (BF) also constructs a set of feasible labelings $P$ and then picks an element to use as its labeling. It has nice theoretical guarantees [An and Dasgupta, 2024], but

there are two drawbacks. Not only does it need labeled data to construct $P$, not having enough labels makes $P$ big and BF's performance poor.

We start by constructing a subset $P \subset \Delta_K^N$ of all possible labelings such that the true labeling $\vec{\eta}$ lies inside $P$. To approximate $\vec{\eta}$, we play a zero-sum minimax game as the learner, who tries to minimize the worst case loss a potential adversary can inflict on us. That is, if we select labeling $\vec{g} \in \Delta_K^N$, the adversary selects $\vec{z} \in P$ that maximizes $-\vec{z}^\top \log \vec{g}$.

We can construct $P$ by using $L$ labeled points to bound each LF's accuracy on $X$ with error at most $O(1/\sqrt{L})$. Formally, if $h^{(w)}$ makes $N_w \leq N$ predictions, we can with $L$ labeled points construct

$$b_w - \epsilon_w \leq b_w^* = \frac{1}{N_w} \sum_{i=1}^{N} \sum_{j=1}^{K} \eta_{ij} \mathbf{1}(y_{iw} = j) \leq b_w + \epsilon_w \tag{1}$$

with a binomial confidence interval, where $b_w^*$ is $h^{(w)}$'s accuracy on $X$. Something similar can be done with the class frequencies: $\omega_j^* = \frac{1}{N} \sum_{i=1}^{N} \eta_{ij}$ in the interval $[\omega_j - \xi_j, \omega_j + \xi_j]$.

These constraints can easily be written in matrix form. Abuse notation and let $\vec{y}_{iw}$ be the one-hot encoding of $h^{(w)}(x_i)$. Then, $\vec{y}_{iw} = (\mathbf{1}(h^{(w)}(x_i) = 1), \ldots, \mathbf{1}(h^{(w)}(x_i) = K))$ where $\vec{y}_{iw}$ is all zeros if $h^{(w)}(x_i) = ?$. We write all of $h^{(w)}$'s predictions as $\vec{y}_w = (\vec{y}_{1w}, \ldots, \vec{y}_{Nw}) \in (\Delta_K \cup \{\vec{0}_K\})^N$. Note that the $\vec{y}_w$ vectors are the only vectors of length $NK$ that have special indexing with three indices. Other vectors of length $NK$ (in $\Delta_K^N$) will follow the same style of indexing as $\vec{\eta}$. Also, we write $\vec{e}_j$ to mean the $j^{th}$ canonical basis vector in $K$ dimensions concatenated $N$ times. To write out constraints, compactly, define $A \in [0,1]^{(W+K) \times NK}$ row wise where

$$A = \begin{cases} \frac{1}{N_m} \vec{y}_m & \text{for } m \leq W \\ \frac{1}{N} \vec{e}_{m-W} & \text{otherwise.} \end{cases}$$

Finally, for brevity, define $\vec{b} := (b_1, \ldots, b_W, \omega_1, \ldots, \omega_K)$. Similarly, $\vec{\epsilon} := (\epsilon_1, \ldots, \epsilon_W, \xi_1, \ldots, \xi_K)$. Then for elementwise inequalities, $P$ can be written as

$$P = \{\vec{z} \in \Delta_K^N : \vec{b} - \vec{\epsilon} \leq A\vec{z} \leq \vec{b} + \vec{\epsilon}\}. \tag{2}$$

Now, the zero-sum minimax game referred to earlier can be formally written as

$$\max_{\vec{g} \in \Delta_K^N} \min_{\vec{z} \in P} \vec{z}^\top \log \vec{g} = \min_{\vec{z} \in P} \vec{z}^\top \log \vec{z} \tag{3}$$

where the equality is shown by An and Dasgupta [2024] and proven by them to be an easily optimizable convex program. Formally speaking, when one doesn't have enough labeled data, the elements of $\vec{\epsilon}$ are large, meaning $P$ is large and BF is overly pessimistic. We will soon see how BBF addresses both drawbacks of BF by implicitly constructing a polytope $P$ and adopting a label selection strategy similar to that of HyperLM's.

## 4 Our Proposed Method

We now described our proposed label model, Bayesian Balsubramani-Freund (BBF). At a high level, we put prior distributions on quantities in BF that are point estimates, specifically the class frequencies and LF accuracies. We'll describe the generative process, present the joint distribution, and discuss how to find BBF's posterior mode, which we'll take as BBF's prediction.

### 4.1 The Generative Process

To generate the labels and LF predictions, BBF is similar to iBCC [Kim and Ghahramani, 2012] in that follows the same order (labels first, then LF predictions) and parameterizes LFs by a few parameters. Rather than model the underlying LF accuracy $\Pr(h^{(w)}(X) = Y)$ like in iBCC, BBF tries to model the accuracy of an LF on $X$, i.e. $b_w^*$. (And similarly for $\Pr(Y = j)$, $\omega_j^*$.) We'll later see BBF's generative process and a simplified version of iBCC's generative process side by side.

The first step in BBF's generative process is to draw the class frequency distribution $\vec{\tau}$ from a Dirichlet prior with hyperparameters $\vec{\alpha} \in \mathbb{R}_{>0}^K$. Then, the labels $\vec{z} \in \Delta_K^N$ are drawn. We'll want $\vec{z}$ to assign $\tau_j N$ total mass for each class $j \in [K]$. That is, $\vec{z}$ will be drawn from

$$\mathcal{L}_{\vec{\tau}} := \left\{ \vec{z} \in \Delta_K^N : \frac{1}{N} \sum_{i=1}^N z_{ij} = \tau_j \text{ for each } j \in [K] \right\}.$$

Rather than draw labelings at random from $\mathcal{L}_{\vec{\tau}}$, we'll favor labelings with large entropy. This is necessary for our theoretical results. Our distribution for labelings will be denoted $Entropy(\mathcal{L}_{\vec{\tau}})$ which assigns density $\Pr(\vec{z} \mid \vec{\tau}) \propto \exp(-\vec{z}^\top \log \vec{z})$ for $\vec{z} \in \mathcal{L}_{\vec{\tau}}$ and 0 otherwise.

The next step is to draw the accuracies for each LF. A first pass approach is to have a Beta prior with hyperparameters $\vec{\rho}_w \in \mathbb{R}_{>0}^2$ for LF $h^{(w)}$'s accuracy $b_w$ on $X$. This won't work in general because a Beta random variable's support is $[0, 1]$, but the choice of $\vec{z}$ can make certain accuracies impossible. For example, if $\vec{z} = \vec{1}_{NK}/K$, then regardless of what $h^{(w)}$ predicts, its accuracy can only be $1/K$. On the other hand, if $z_{11} = 1$ and $z_{ij} = 1/K$ for $i \geq 2$ and $j \in [K]$, an LF that only predicts on $x_1$ can have accuracies in $[0, 1]$ while an LF that doesn't abstain must have accuracy in the range $[(N-1)/NK, (N-1)/NK + 1/N]$. Thus, we need to restrict the range of accuracies that can be generated. For a fixed labeling $\vec{z}$, define the range of permissible accuracies with

$$b_{max} := \max_{i \in [N], j \in [K]} z_{ij} \quad \text{and} \quad b_{min} := \min_{i \in [N], j \in [K]} z_{ij}.$$

With this, we draw $b_w$ from $Beta_{\vec{z}}(\vec{\rho}_w)$, which is a $Beta(\vec{\rho}_w)$ with support restricted to $[b_{min}, b_{max}]$ and density renormalized. Note that the number of predictions $h^{(w)}$ makes can depend on $b_w$.

Finally, we draw predictions for each of the LFs. For readability, abuse notation and let $N: (\Delta_K \cup \{\vec{0}_K\})^N \to \mathbb{R}_{\geq 0}$ be the function that counts the number of predictions made: $N(\vec{y}) = \vec{y}^\top \vec{1}_{NK}$. $h^{(w)}$'s prediction is randomly drawn from the set of all possible predictions that achieve accuracy $b_w$:

$$\mathcal{L}_{b_w, \vec{z}} := \left\{ \vec{y} \in (\Delta_K \cup \{\vec{0}_K\})^N : \frac{\vec{y}^\top \vec{z}}{N(\vec{y})} = b_w \right\}.$$

We now summarize the above and compare it to a simplified version of iBCC's generative process. There, each LF is a biased coin and the datapoint labels/LF predictions are deterministic: $z_i, y_{iw} \in \mathcal{Z}$. I.e. the distribution of $y_{iw} = h^{(w)}(x_i)$ with $z_i = j$ is described by $v(b_w, j) \in \Delta_K$, which has $b_w$ in the $j^{th}$ position and $(1 - b_w)/(K - 1)$ elsewhere.

BBF Generative Process:

1. $\vec{\tau} \sim Dirichlet(\vec{\alpha})$

2. $\vec{z} \sim Entropy(\mathcal{L}_{\vec{\tau}})$

3. For each LF $w \in [W]$:

    (a) $b_w \sim Beta_{\vec{z}}(\vec{\rho}_w)$

    (b) $\vec{y}_w \sim Uniform(\mathcal{L}_{b_w, \vec{z}})$

One-Coin iBCC Generative Process

1. $\vec{\tau} \sim Dirichlet(\vec{\alpha})$

2. For each datapoint $i \in [N]$:

    (a) $z_i \sim Categorical(\vec{\tau})$

3. For each LF $w \in [W]$:

    (a) $b_w \sim Beta(\vec{\rho}_w)$

    (b) For each datapoint $i \in [N]$:

        i. $y_{iw} \sim Categorical(v(b_w, z_i))$

## 4.2 The Joint Distribution

Now, we write out the joint distribution for BBF. Write $\vec{b} \in [0, 1]^W$ the vector of LF accuracies, $\vec{\rho}$ the $2W$ Beta hyperparameters, and $Y$ the set of all LF predictions. We have that

$$\Pr(\vec{z}, \vec{b}, \vec{\tau}, Y \mid \vec{\alpha}, \vec{\rho}) \propto \Pr(Y \mid \vec{z}, \vec{b}) \Pr(\vec{b} \mid \vec{z}, \vec{\rho}) \Pr(\vec{z} \mid \vec{\tau}) \Pr(\vec{\tau} \mid \vec{\alpha}).$$

We now write each of the terms therein. First, $\Pr(\vec{\tau} \mid \vec{\alpha}) \propto \prod_{j=1}^K \tau_j^{\alpha_j - 1}$. Then,

$$\Pr(\vec{z} \mid \vec{\tau}) \propto \exp(-\vec{z}^\top \log \vec{z}) \prod_{j=1}^K \mathbf{1}\left(\frac{1}{N} \vec{e}_j^\top \vec{z} = \tau_j\right), \quad \Pr(Y \mid \vec{z}, \vec{b}) \propto \prod_{w=1}^W \mathbf{1}\left(\frac{1}{N(\vec{y})} \vec{y}_w^\top \vec{z} = b_w\right).$$

To write the probability of the LF accuracies, we will need the Beta distribution's CDF at $r \in [0, 1]$, written as $I_r = I_r(\alpha, \beta)$ for short. To improve readability we drop the hyperparameters, but note that in each term of the following product, the difference of CDFs is specific to each LF.

$$\Pr(\vec{b} \mid \vec{z}, \vec{\rho}) \propto \prod_{w=1}^{W} \frac{b_w^{\rho_{w1}-1}(1 - b_w)^{\rho_{w2}-1}}{I_{b_{max}} - I_{b_{min}}}$$

### 4.3 Computing the Posterior Label Distribution

In practice, we know the LF predictions $Y = \{y_{iw} : w \in [W], i \in [N]\}$ (along with the hyperparameters) and want to know the posterior label distribution. Thus, we are interested in the quantity $\Pr(\vec{z}, \vec{b}, \vec{\tau} \mid Y, \vec{\alpha}, \vec{\rho})$. It turns out that this quantity is only non-zero when every element of $\vec{b}$ and $\vec{\tau}$ are deterministic functions of $\vec{z}$ (when $Y$ is given). This is because the indicator functions in the joint distribution ensure the above is non-zero only when $\tau_j = \vec{e}_j^\top \vec{z}/N$ and $b_w = \vec{y}_w^\top \vec{z}/N(\vec{y})$ for every $j \in [K]$ and $w \in [W]$. Now, if we use $\tau_j, b_w$ to replace the dot products, we have

$$\Pr(\vec{z}, \vec{b}, \vec{\tau} \mid Y, \vec{\alpha}, \vec{\rho}) \propto \exp(-\vec{z}^\top \log \vec{z}) \times \prod_{j=1}^{K} \tau_j^{\alpha_j-1} \prod_{w=1}^{W} \frac{b_w^{\rho_{w1}-1}(1 - b_w)^{\rho_{w2}-1}}{I_{b_{max}} - I_{b_{min}}}. \quad (4)$$

In the Appendix, we show that the removal of the difference of Beta CDFs will make the RHS log-concave in $\vec{z}$ (with an appropriate choice of hyperparameters).

**Lemma 4.1.** *The modified posterior label distribution for our proposed model written below is log-concave in $\vec{z}$ when all elements of $\vec{\alpha}, \vec{\rho} \geq 1$.*

$$\exp(-\vec{z}^\top \log \vec{z}) \prod_{j=1}^{K} \left( \frac{\vec{e}_j^\top \vec{z}}{N} \right)^{\alpha_j-1} \times \prod_{w=1}^{W} \left( \frac{\vec{y}_w^\top \vec{z}}{N(\vec{y}_w)} \right)^{\rho_{w1}-1} \left( 1 - \frac{\vec{y}_w^\top \vec{z}}{N(\vec{y}_w)} \right)^{\rho_{w2}-1}$$

BBF's prediction $\vec{g}^{bbf}$ will be the mode of the modified posterior distribution, i.e. the solution to

$$\max_{\vec{z} \in \Delta_K^N} \left[ -\vec{z}^\top \log \vec{z} + \sum_{j=1}^{K} (\alpha_j - 1) \log \left( \frac{\vec{e}_j^\top \vec{z}}{N} \right) \right.$$
$$\left. + \sum_{w=1}^{W} \left[ (\rho_{w1} - 1) \log \left( \frac{\vec{y}_w^\top \vec{z}}{N(\vec{y}_w)} \right) + (\rho_{w2} - 1) \log \left( 1 - \frac{\vec{y}_w^\top \vec{z}}{N(\vec{y}_w)} \right) \right] \right] \quad (5)$$

which is a concave maximization problem. This can be solved using an off the shelf convex solver. We note that maximizing the modified log posterior without both holding all predictions $Y$ in memory and simultaneously optimizing over all $NK$ variables is an open problem.

## 5 Statistical Analysis of Our Proposed Model

We now argue that BBF implicitly constructs a polytope $P \subset \Delta_K^N$ of plausible labelings and selects a labeling inside as its prediction, a la HyperLM. To start, compare BBF's objective (Equation 5) with BF's objective (Equation 3). Both are entropy maximization problems– BF constrains $\vec{z} \in P$ while BBF has no such constraints, but has extra terms in the objective. Observe that we can rewrite the BF problem as $\max_{\vec{z} \in \Delta_K^N}[-\vec{z}^\top \log \vec{z} + M\mathbf{1}(\vec{z} \in P)]$ where $M$ is an extremely large constant. In order for $\vec{z}$ to be optimal $M$ must appear, meaning $\vec{z}$ must be in $P$. We want to argue that the sum of log-pdfs of a Dirichlet and multiple Betas in the BBF objective serve as a continuous approximation to $M\mathbf{1}(\vec{z} \in P)$. Call $\vec{e}_j^\top \vec{z}/N$ and $\vec{y}_w^\top \vec{z}/N(\vec{y}_w)$ the class frequency/LF accuracy induced by $\vec{z}$. The sum of log-pdfs is maximized when the induced class frequencies/LF accuracies are close to the modes of the Dirichlet/Beta distributions. I.e. close to $(\alpha_j - 1)/(\sum_{j'=1}^{K} \alpha_{j'} - K)$, $(\rho_{w1} - 2)/(\rho_{w1} + \rho_{w1} - 2)$ respectively for each $j \in [K], w \in [K]$. Thus, for BBF, we may regard its polytope $P'$ as the $\vec{z}$'s whose induced accuracies are not too far from the mode. For LF accuracies, that is $|\vec{y}_w^\top \vec{z}/N(\vec{y}_w) - (\rho_{w1} - 2)/(\rho_{w1} + \rho_{w1} - 2)| \leq \epsilon_w$ for some $\epsilon_w$. This is exactly the form of inequality used to specify BF's polytope (Equation 1). For sufficiently large $\vec{\alpha}, \vec{\rho}, \vec{g}^{bbf}$ will induce

class frequencies/LF accuracies close to the modes, which means $g^{\vec{b}bf}$ will be in the interior of $P'$. Note that unlike BF, BBF's performance is affected by duplicate LFs in most cases. It is only in a special case that BBF is not affected by duplicate LFs. However, the following results for BBF do not depend on having a set of unique LFs. With that, we are ready to analyze BBF's form of solution, consistency, and its convergence rates.

## 5.1  $\vec{g}^{bbf}$ is a Weighted Majority Vote

Here, we show that $\vec{g}^{bbf}$ is completely specified by $W + K$ real numbers and the LF predictions. In particular, for any datapoint $x_i$, $\vec{g}^{bbf}_i$ is the softmax of a weighted majority vote. Let $\vec{\theta} \in \mathbb{R}^{W+K}$ be our weights, the first $W$ being for the LFs, the last $K$ for the class frequencies. We are claiming that

$$
g^{bbf}_{ij} = \frac{\exp\left(\sum_{w=1}^{W} \theta_w y_{iwj} + \theta_{W+j}\right)}{\sum_{j'=1}^{K} \exp\left(\sum_{w=1}^{W} \theta_w y_{iwj'} + \theta_{W+j'}\right)} = \frac{\exp((A^\top \vec{\theta})_{ij})}{\sum_{j'=1}^{K} \exp((A^\top \vec{\theta})_{ij'})}.
$$

for some $\vec{\theta}$. By varying $\vec{\theta}$ over the reals, one gets an exponential family of distributions:

$$
\mathcal{G} = \{\vec{g}^{(\vec{\theta})} : \vec{\theta} \in \mathbb{R}^{W+K}\} \quad \text{where} \quad g^{(\vec{\theta})}_{ij} \propto \exp((A^\top \vec{\theta})_{ij}).
$$

**Lemma 5.1.** *For fixed LF predictions and hyperparameters $\vec{\alpha}, \vec{\rho} \geq 1$ elementwise, the BBF prediction $\vec{g}^{bbf}$ gotten by solving Equation 5 is such that $\vec{g}^{bbf} \in \mathcal{G}$.*

Now, we analyze the behavior of $\vec{g}^{bbf}$ as $\vec{\alpha}, \vec{\rho}$ change and the LF predictions/$\vec{\eta}$ remain fixed.

## 5.2  Consistency

While BBF's prediction must lie in $\mathcal{G}$, $\vec{\eta}$ in general doesn't lie in $\mathcal{G}$. Thus, the best we can hope for is to get the best approximator of $\vec{\eta}$ from $\mathcal{G}$, i.e. $\vec{g}^* = \vec{g}^{(\vec{\theta}^*)} = \arg\min_{\vec{g} \in G} d(\vec{\eta}, \vec{g})$ for distance function $d(\cdot, \cdot)$. KL divergence will be the distance we choose. For $\vec{\mu}, \vec{\nu} \in \Delta_K^N$, recall they are each concatenations of $N$ distributions, so

$$
d(\vec{\mu}, \vec{\nu}) = \sum_{i=1}^{N} KL(\vec{\mu}_i, \vec{\nu}_i) = \sum_{i=1}^{N} \sum_{j=1}^{K} \mu_{ij} \log\left(\frac{\mu_{ij}}{\nu_{ij}}\right).
$$

Now we'll informally argue that BBF is consistent insomuch as we can make $\vec{g}^{bbf}$ tend to $\vec{g}^*$. In the beginning of this section, we had argued that BBF implicitly constructs a polytope $P'$ and picks a labeling from inside $P'$. It turns out for a specific polytope and a specific strategy of selecting a labeling from inside, one can obtain $\vec{g}^*$. Consider the polytope $P^* = \{\vec{z} \in \Delta_K^N : A\vec{z} = \vec{b}^*\}$, the set of all labelings that induce the empirical LF accuracies/empirical class frequencies. An and Dasgupta [2024] showed (in Theorem 6) that if BF is given $P^*$, then BF will return $\vec{g}^*$. Our result for BBF shows how to make the implicit polytope of BBF to tend to $P^*$ whereupon $\vec{g}^{bbf} \to \vec{g}^*$. In the Appendix, we show that BBF satisfies a modified version of BF's KKT conditions (for a fixed implicit polytope) where the complementary slackness condition is not satisfied. However, as we make BBF's implicit polytope tend to $P^*$, the complementary slackness conditions will be satisfied, meaning BBF replicates the behavior of BF with $P^*$. Only then is BBF unaffected by duplicate LFs.

To make BBF's implicit polytope tend to $P^*$, we need to make it so whenever the induced class frequencies/LF accuracies are not close to $w_j^*, b_w^*$ the resulting contribution from the sum of the log-pdfs is small. For convenience, define $\alpha_0 = \sum_{j=1}^{K} \alpha_j$.

**Theorem 5.2.** *If for each $w \in [W]$, $\rho_{w1}, \rho_{w2} \to \infty$, $|\rho_{w1} - 1 - b_w^*(\rho_{w1} + \rho_{w2} - 2)| \to 0$ and for each $j \in [K]$, $\alpha_j \to \infty$ and $|\alpha_j - 1 - \omega_j^*(\alpha_0 - K)| \to 0$, then $\vec{g}^{bbf} \to \vec{g}^*$.*

Note that this is not the Bayesian notion of consistency. There, one has appropriately selected priors with fixed hyperparameters and the requirement is the number of observations (e.g. LF predictions) tends to infinity [Miller, 2018]. For BBF, our proposed model only has 1 observation, the set of all LF predictions on $X$, meaning we cannot take advantage of the vast literature on Bayesian consistency.

## 5.3 Rates of Convergence

We'll now see $\vec{g}^{bbf}$'s rate of convergence to $\vec{g}^*$. Specially, we'll bound $d(\vec{g}^*, \vec{g}^{bbf})$ by a function of the weights that produce $\vec{g}^*, \vec{g}^{bbf}$, call them $\vec{\theta}^*, \vec{\theta}^{bbf}$, and how well $\vec{g}^{bbf}$ estimates the actual LF accuracies/class frequencies.

**Theorem 5.3.** *Suppose $\vec{\alpha}, \vec{\rho} \geq 1$ elementwise. Then for elementwise absolute value,*

$$d(\vec{g}^*, \vec{g}^{bbf}) \leq (|\vec{\theta}^*| + |\vec{\theta}^{bbf}|)^\top |\vec{b}^* - A\vec{g}^{bbf}|.$$

*When the conditions of Theorem 5.2 hold, $\vec{\theta}^{bbf} \rightarrow \vec{\theta}^*$.*

It's unclear to us how one bounds $\vec{\theta}^{bbf}$ or how well $A\vec{g}^{bbf}$ approximates $b^*$, meaning we cannot say on what order BBF's convergence rate is. We note that duplicate LFs will increase the size of the bound as each duplicate LF adds an extra term. Now, if an unsupervised instantiation of $\vec{\alpha}, \vec{\rho}$ induces $\vec{g}^{bbf}$ such that $A\vec{g}^{bbf}$ is close to $b^*$, then the excess error will be small. Experimental results show that BBF performs well in the unsupervised setting and often has small excess error compared to $\vec{g}^*$.

# 6 Experiments

Here, we compare the performance of BBF against other baseline methods on eleven real datasets while also showing BBF's consistency on real datasets. Experiments were run in Python 3.6.13 (PSF) [Van Rossum and Drake, 2009] mainly using NumPy 1.19.5 (modified BSD) [Harris et al., 2020] with an AMD Ryzen R9 5950x, 128GB RAM, and an Nvidia RTX 2080Ti. One may find the code in the supplementary materials or at `https://github.com/stevenan5/bayesian-bf-neurips-2025`.

## 6.1 Methods

We compare our proposed method with twelve other methods, representing different approaches to creating the label model. For our method BBF, we set the prior hyperparameters as $\vec{\alpha} = \vec{1}_K$, $\vec{\rho}_w = (4, 1)$ for each $w \in [W]$. This follows the initialization of [Li et al., 2019] for their Bayesian method, i.e. we assume each LF makes $4$ correct predictions and $1$ wrong prediction. Note that there are no requirements on the LF accuracies for BBF. Across all LFs used in the experiments, the accuracies range from $0.0366$ to $1$. To compute the BBF prediction, we optimize Equation 5 using CVXPY 1.4.1 (Apache 2.0) [Diamond and Boyd, 2016] by way of MOSEK 10.1.16 (personal academic license) [MOSEK ApS, 2022] with default parameters.

The other twelve methods are from the WRENCH benchmark version 1.1.2rc0 (Apache 2.0) [Zhang et al., 2021]. They are: majority vote (MV), Dawid-Skene (DS) [Dawid and Skene, 1979], Data Programming (DP) [Ratner et al., 2016], a generalization of DS that accounts for some LF dependencies, Flying Squid (FS) [Fu et al., 2020], a fast method that uses an Ising model, MeTaL [Ratner et al., 2019], a matrix completion type method, iBCC [Kim and Ghahramani, 2012], the Bayesian generalization of DS, EBCC [Li et al., 2019], a generalization of iBCC, but where each LF is modeled by multiple confusion matrices, FABLE [Zhang et al., 2023], a generalization of EBCC which can account for datapoint features, Constrained Label Learning (CLL) [Arachie and Huang, 2021], a polytope based method with an adversarial flavor, Hyper Label Model (HyperLM) [Wu et al., 2023], a graph neural network based method, WeaSEL [Cachay et al., 2021], and Denoise [Ren et al., 2020], two neural network based methods using attention with different losses. We ported CLL into WRENCH and following Arachie and Huang [2021], we initialized LF errors to be $0.01$ if the LF coverage was low (i.e. low number of datapoints predicted on) and $\frac{1}{K}$ if the LF coverage was high. For all other methods that require initialization, we use the defaults provided in WRENCH.

## 6.2 Datasets

We used eleven datasets from WRENCH, which also provided the LF predictions. The following datasets had licenses we could find online and were different from WRENCH's license: YouTube, SMS (CC BY 4.0), SemEval (CC BY 3.0). For FABLE, Denoise, WeaSEL, we use the original datapoint features, unless they were textual, in which case RoBERTa [Liu et al., 2019] was used to extract them. The methods are evaluated in a transductive setting. The provided train/validation/test

Table 1: Some dataset statistics. Note that we only count datapoints with at least one LF prediction.

| Dataset | IMDB | Youtube | SMS | CDR | Yelp | Commercial | Tennis | TREC | SemEval | ChemProt | AG News |
|---|---|---|---|---|---|---|---|---|---|---|---|
| # Class ($K$) | 2 | 2 | 2 | 2 | 2 | 2 | 2 | 6 | 9 | 10 | 4 |
| # LF ($W$) | 5 | 10 | 73 | 33 | 8 | 4 | 6 | 68 | 164 | 26 | 9 |
| # Datapoints ($N$) | 21,914 | 1,843 | 2,246 | 12,562 | 31,512 | 81,105 | 8,803 | 5,738 | 2,527 | 13,768 | 82,591 |

Table 2: Comparison of our proposed method with baseline methods (higher is better). Bolded entries are indistinguishable via two-tailed paired t-test with $p = 0.05$.

| Dataset | IMDB | YouTube | SMS | CDR | Yelp | Commercial | Tennis | TREC | SemEval | ChemProt | AG News |
|---|---|---|---|---|---|---|---|---|---|---|---|
| Loss | F1 | F1 | F1 | F1 | F1 | F1 | F1 | 0-1 | 0-1 | 0-1 | 0-1 |
| MV | 75.19 | 84.56 | **78.88** | 67.28 | 78.22 | 84.46 | 83.83 | 54.51 | 79.07 | 54.67 | 81.38 |
| DS | 65.08 | 83.56 | 65.30 | 57.90 | 75.20 | 88.31 | 83.56 | 47.66 | 73.53 | 52.66 | 81.79 |
| DP | 74.84 | 78.62 | 57.82 | 41.78 | 71.61 | 78.61 | 83.61 | 45.85 | 72.78 | 56.17 | 81.71 |
| FS | 75.51 | 83.07 | 70.55 | 69.12 | 78.67 | 80.86 | 83.31 | 50.10 | 12.15 | 52.37 | 81.28 |
| MeTaL | 74.78 | 76.34 | 48.39 | 17.69 | 70.14 | 78.62 | 83.62 | 43.73 | 54.58 | 55.86 | 81.84 |
| iBCC | 75.33 | 0.00 | 0.00 | 0.00 | 75.83 | 76.83 | 84.16 | 21.02 | 30.19 | 35.09 | 26.57 |
| EBCC | 75.32 | 0.00 | 0.00 | 24.81 | 76.57 | 76.83 | 84.19 | 21.02 | 30.19 | 35.09 | 27.81 |
| CLL | 74.80 | 84.79 | **78.83** | 64.33 | 77.80 | 36.38 | 42.31 | 59.34 | **84.21** | 52.37 | 80.97 |
| HyperLM | 75.29 | **91.40** | **78.90** | 70.54 | 78.73 | 82.60 | **84.21** | **60.02** | **84.21** | 52.32 | 81.38 |
| WeaSEL | 53.07 | 61.85 | 26.45 | 38.50 | 61.91 | 40.00 | 5.74 | 20.81 | 11.14 | 21.41 | 32.29 |
| Denoise | **81.05** | 79.84 | **79.06** | **70.94** | **80.97** | **89.71** | 51.12 | 49.81 | 73.53 | **57.85** | **87.61** |
| Fable | 74.97 | 87.80 | 78.75 | 62.40 | 77.03 | 85.91 | 83.59 | 53.64 | 73.61 | 54.42 | 81.38 |
| BBF | 74.67 | 89.71 | 78.77 | 62.39 | 77.67 | 85.02 | 79.87 | 59.88 | **84.21** | 52.47 | 81.40 |
| Improve | 5.86 | 6.84 | 0.18 | 3.66 | 2.75 | 5.25 | 0.38 | 5.51 | 5.14 | 3.18 | 6.23 |
| $g^*$ | 75.53 | 94.22 | 78.83 | 72.25 | 78.83 | 88.31 | 83.43 | 63.75 | 84.21 | 61.26 | 82.24 |

splits are combined and points with no LF predictions are removed. Each label model is given LF predictions on the datapoints (and the datapoint features themselves if used). No labeled data is provided to the label models. Label models are expected to be initialized and provide a labeling without using any labels. Labels are only used to judge the quality of each label model's predictions by way of F1 score/0-1 loss. Tasks covered by the datasets range from sentiment/spam/bio relation/video frame/chemical relation/topic classification. See Table 1 for some statistics.

## 6.3 Comparison with Other Label Models

For every method that involves randomness, we run it ten times and display the average performance in Table 2. Bolded entries represent ones that are indistinguishable from the best result via two-tailed paired t-test with $p = 0.05$. WeaSEL, Denoise, and FABLE are the only methods that use datapoint features – all others only use LF predictions. The second to last row, denoted "Improve" shows the improvement of the best label model over majority vote. The last row shows $\vec{g}^*$ (computed via Theorem 6 of [An and Dasgupta, 2024]), a proxy for the ceiling of BBF's performance. See the Appendix for the complete results (0-1 loss on all datasets, standard deviations of losses).

Against the highest performing Bayesian method FABLE, BBF often matches or outperforms, only significantly underperforming on Tennis. This is notable because FABLE uses datapoint features and is much more expressive than BBF. On Tennis, the three LFs that each predicted on $99\%$ of the datapoints closely matched each other, i.e. had $> 85\%$ matching predictions. This effectively reduces the number of constraints on BBF's implicit polytope because three constraints are almost duplicates.

Compared to HyperLM, BBF is competitive except for CDR and Tennis. On CDR, the number of LF predictions per datapoint was very low, averaging 2.3 predictions per datapoint despite there being 33 LFs. Two predictions per datapoint means BBF's implicit polytope has very few constraints per datapoint, likely diminishing the performance.

Against Denoise, we see the limitations of representing an LF by its accuracy – even with $\vec{g}^*$, BBF is worse on IMDB, SMS, Yelp, etc. For ChemProt specifically, we think BBF is worse than Denoise because the average LF accuracy is just under $0.47$, which differs greatly from our mean prior accuracy of $0.8$. For almost every other dataset, the average accuracy of the LFs was within $0.11$ of $0.8$. SemEval was the outlier with average LF accuracy of $0.97$.

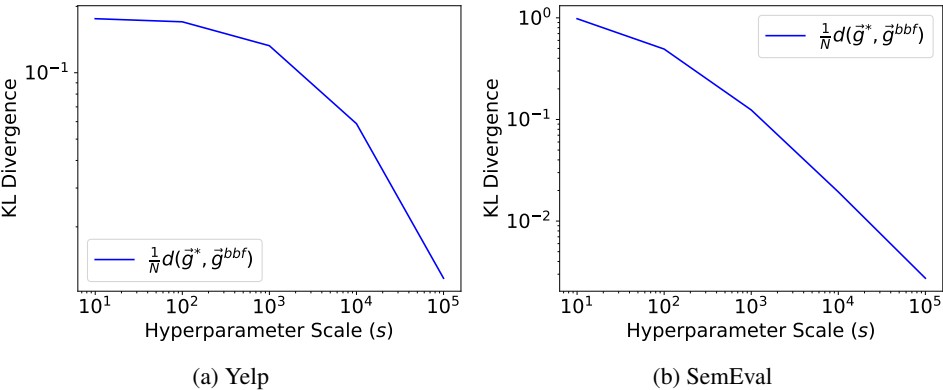

Figure 1: Demonstrations of BBF's consistency on real datasets.

For all other methods, BBF has similar if not better performance on a large majority of the datasets. The exceptions are on CDR against FS, Commercial against DS, Tennis against almost every method, and ChemProt against DP/FABLE. This is evidence of BBF's strength as a label model.

Since BBF employs a similar strategy to HyperLM, we now quickly discuss the performance differences between HyperLM and Denoise. On IMDB, Commercial, ChemProt, AGNews, HyperLM performs worse than Denoise, so much so that it's close to majority vote in performance. We hypothesize that the "center" of the polytope constructed by HyperLM might be close to the majority vote labeling. This is because we could not find a pattern between whether HyperLM's assumptions were met, the number and/or quality of the LFs provided, and whether HyperLM underperformed against Denoise. In the cases where Denoise underperformed against HyperLM, we believe this is because Denoise might get stuck in poor local optima as its strategy for learning is similar to that of bootstrapped co-training [Blum and Mitchell, 1998].

### 6.4 Consistency of BBF

To show that BBF is consistent on real datasets, we choose $\alpha_j = s\omega_j^* + 1$, $\vec{\rho}_w = (sb_w^* + 1, s(1 - b_w^*) + 1)$ with $s > 0$. By having $s$ take values in $\{10^1, 10^2, \ldots, 10^5\}$, we simulate the conditions of Theorem 5.2. Figure 1 shows $d(\vec{g}^*, \vec{g}^{bbf})/N$ on a log-log scale graph on two datasets. We see the convergence is at worst linear as a function of $s$. See Appendix for complete consistency results.

## 7 Limitations

We have seen a new label model that is adversarial, unsupervised, has strong empirical performance and enjoys nice theoretical guarantees. This was done by combining HyperLM and BF's strategies.

However, there are still some unsolved problems. On the empirical front, one must hold $O(WN)$ items in memory and optimize $NK$ variables simultaneously to get $\vec{g}^{bbf}$. This is infeasible when $N$ is large. We also noticed in our experiments that $||\vec{g}^{bbf}||_\infty$ was small, meaning BBF can be poorly calibrated. On the theoretical front, we are not able to control $\bar{\theta}^{bbf}$, meaning we do not know at which rate $\vec{g}^{bbf} \to g^*$ a priori. Due to $\vec{g}^{bbf}$ not being a minimax optimal solution, we lose the guarantee that $-\vec{\eta}^\top \log \vec{g}^{bbf} \leq -\vec{g}^{bbf\top} \log \vec{g}^{bbf}$. I.e. we don't have a bound for the cross entropy between $\vec{\eta}$ and $\vec{g}^{bbf}$ which is only in terms of $\vec{g}^{bbf}$.

## Acknowledgments and Disclosure of Funding

The author thanks the National Science Foundation for its support under grant IIS-2211386, Sanjoy Dasgupta for suggesting the Balsubramani-Freund model be given a Bayesian generalization, and the anonymous reviewers for their time and feedback, which indubitably improved the paper.

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

## A   Appendix Overview

Here, we provide the missing proofs along with extra experimental results. After a brief notation recap, the rest of the Appendix is organized as follows:

- Section C, results involving the Bayesian Balsubramani-Freund (BBF) posterior label distribution, log-concavity (Lemma 4.1).
- Section D, an overview of the proof strategy used to show the remaining theoretical results. A table of contents detailing major steps of the main proof is provided there. The proofs will be written for LFs that don't abstain, but they'll work for LFs that do abstain.
- Section E, extra experimental results.
    - Subsection E.1, extra tables showing 0-1 loss for all datasets as well as the standard deviation of losses.
    - Subsection E.2, extra experimental results showing BBF's consistency on real datasets.

## B   Notation Recap

We would like to label $N$ points in the set $X = \{x_1, \ldots, x_N\}$ which each have one of $K$ labels. The set of labels in consideration is $\mathcal{Z} = \{1, \ldots, K\} = [K]$. At our disposal are $W$ labeling functions (LFs) $h^{(1)}, \ldots h^{(W)} \colon X \to \mathcal{Z} \cup \{?\}$ where "?" denotes abstention. We seek to infer the true conditional label probabilities for each point $x_i$:

$$\eta_{ij} = \Pr(z = j \mid x_j) \quad \text{so that} \quad \vec{\eta}_i = (\eta_{i1}, \ldots, \eta_{iK}) \in \Delta_K.$$

For convenience, we'll concatenate all $N$ of these distributions:

$$\vec{\eta} = (\vec{\eta}_1, \ldots, \vec{\eta}_N) \in \Delta_K^N.$$

Similarly, we'll represent LF predictions by $N$ probability distributions which are each over $K$ elements. For datapoint $x_i$, the one-hot encoding of LF $h^{(w)}$'s prediction is denoted

$$\vec{y}_{iw} = (\mathbf{1}(h^{(w)}(x_i) = 1), \ldots, \mathbf{1}(h^{(w)}(x_i) = K)) \in \Delta_K \cup \{\vec{0}_k\}$$

where $\vec{0}_k$ is a zeros vector of length $K$. We note that for the theory, $\vec{y}_{iw}$ doesn't need to be a vector of zeros and ones. It can be a bona fide distribution without any effect to the results shown below. The set of all predictions for this LF is

$$\vec{y}_w = (\vec{y}_{1w}, \ldots, \vec{y}_{Nw}) \in (\Delta_K \cup \{\vec{0}_K\})^N.$$

Two sets of important quantities are the LF accuracies and the class frequency distribution. Suppose $h^{(w)}$ predicts on $N_w \leq N$ points on $X$. What we refer to as the actual accuracy of LF $h^{(w)}$ on $X$ is

$$b_w^* = \frac{1}{N_w} \sum_{i=1}^{N} \sum_{j=1}^{K} \eta_{ij} \mathbf{1}(h^{(w)}(x_i = j)).$$

In dot product form, this is $\vec{y}_w^\top \vec{\eta} / N_w$. The actual class frequency distribution on $X$ is

$$\omega_j^* = \frac{1}{N} \sum_{i=1}^{N} \eta_{ij}.$$

Letting $\vec{e}_j \in \Delta_K^N$ be the concatenation of $N$ copies of the $j^{th}$ canonical basis vector in $K$ dimensions, the above can be written as $\vec{e}_j^\top \vec{\eta}/N$. We sometimes call the starred quantities empirical – i.e. the empirical LF accuracies or the empirical class frequency distribution.

The computation of these values can be written as a matrix vector multiplication. This matrix will be called $A \in [0,1]^{(W+K) \times NK}$. Let the first $W$ rows be $\vec{y}_w/N_w$ for $w \in [W]$ and $\vec{e}_j$ ($j \in [K]$) for rows $W+1, \ldots, W+K$. Then, $A\vec{\eta} \in [0,1]^{W+K}$ is $(b_1^*, \ldots, b_W^*, \omega_1^*, \ldots, \omega_K^*)$. To save space, we'll define

$$\vec{b}^* = (b_1^*, \ldots, b_W^*, \omega_1^*, \ldots, \omega_K^*).$$

This will also be done for error bars: $\vec{\epsilon} \in [0,1]^{W+K}$ where we might write the elementwise inequality $\vec{b}^* - \vec{\epsilon} \le A\vec{\eta} \le \vec{b}^* + \vec{\epsilon}$.

The Dirichlet prior (for BBF's class frequency) has hyperparameters $\alpha \in \mathbb{R}_{>0}^K$. The Beta priors (for LF accuracies) has hyperparameters $\vec{\rho}_w \in \mathbb{R}_{>0}^2$. Variables that are important for the proofs below are $\vec{\tau}$, the generated class frequency distribution, and $\vec{b}$, the generated LF accuracies. The generative process for BBF is as follows:

1. $\vec{\tau} \sim Dirichlet(\vec{\alpha})$
2. $\vec{z} \sim Entropy(\mathcal{L}_{\vec{\tau}})$
3. For each LF $w \in [W]$:
   (a) $b_w \sim Beta_{\vec{z}}(\vec{\rho}_w)$
   (b) $\vec{y}_w \sim Uniform(\mathcal{L}_{b_w, \vec{z}})$

## C   BBF Posterior Label Distribution Concavity (Lemma 4.1)

We'll first manipulate the posterior distribution explicitly to bring about the terms shown in the main paper. Then, a counterexample will be given to show that the BBF posterior is not log-concave in $\vec{z} \in \Delta_K^N$ (the labeling). Finally, we show how a modification makes the posterior log concave.

Let $Y$ be the LF predictions from our $W$ LFs, $\vec{\alpha} \in \mathbb{R}^K$ be the class frequency Dirichlet hyperparameters, $\vec{\rho}$ be the Beta hyperparameters for each LF. If we include everything, the posterior probability for our parameters $\vec{z}$ (labeling), $\vec{b}$ (LF accuracies), $\vec{\tau}$ (class frequencies) is

$$\Pr(\vec{z}, \vec{b}, \vec{\tau} \mid Y, \vec{\alpha}, \vec{\rho}) = \frac{\Pr(\vec{z}, \vec{b}, \vec{\tau}, Y, \vec{\alpha}, \vec{\rho})}{\Pr(Y, \vec{\alpha}, \vec{\rho})} = \frac{\Pr(\vec{z}, \vec{b}, \vec{\tau}, Y, \vec{\alpha}, \vec{\rho})}{\Pr(Y \mid \vec{\alpha}, \vec{\rho})\Pr(\vec{\alpha}, \vec{\rho})} \propto \frac{\Pr(\vec{z}, \vec{b}, \vec{\tau}, Y, \vec{\alpha}, \vec{\rho})}{\Pr(\vec{\alpha}, \vec{\rho})}.$$

We'll now repeatedly apply the chain rule to the numerator, then drop variables which are independent.

$$\Pr(\vec{z}, \vec{b}, \vec{\tau}, Y, \vec{\alpha}, \vec{\rho}) = \Pr(Y \mid \vec{z}, \vec{b}, \vec{\tau}, \vec{\alpha}, \vec{\rho})\Pr(\vec{b} \mid \vec{z}, \vec{\tau}, \vec{\alpha}, \vec{\rho})\Pr(\vec{z} \mid \vec{\tau}, \vec{\alpha}, \vec{\rho})\Pr(\vec{\tau} \mid \vec{\alpha}, \vec{\rho})\Pr(\vec{\alpha}, \vec{\rho})$$

Since the generation of the LF predictions $Y$ only depends on $\vec{z}$ and $\vec{b}$, all other terms can be dropped. Similarly, $\vec{b}$ only depends on $\vec{z}$ and $\vec{\rho}$, $\vec{z}$ only depends on $\vec{\tau}$, and $\vec{\tau}$ only depends on $\vec{\alpha}$. The above is then equal to

$$\Pr(\vec{z}, \vec{b}, \vec{\tau}, Y, \vec{\alpha}, \vec{\rho}) = \Pr(Y \mid \vec{z}, \vec{b})\Pr(\vec{b} \mid \vec{z}, \vec{\rho})\Pr(\vec{z} \mid \vec{\tau})\Pr(\vec{\tau} \mid \vec{\alpha})\Pr(\vec{\alpha}, \vec{\rho}).$$

We have

$$\Pr(\vec{z}, \vec{b}, \vec{\tau} \mid Y, \vec{\alpha}, \vec{\rho}) \propto \Pr(Y \mid \vec{z}, \vec{b})\Pr(\vec{b} \mid \vec{z}, \vec{\rho})\Pr(\vec{z} \mid \vec{\tau})\Pr(\vec{\tau} \mid \vec{\alpha}).$$

We can now show what each term is proportional to. Let $\mathbf{1}(\cdot)$ be the indicator function and recall that $\vec{y}_w \in \Delta_K^N$ is LF $w$'s predictions. Also, recall that $\vec{e}_j$ is the $j^{th}$ canonical basis vector in $K$ dimensions concatenated $N$ times. (Elements $(i-1)K + j$ are 1 for $i \in [N]$, all other elements are 0.) Recall that we've abused notation and have $N(\cdot)$ count the number of predictions an LF makes. I.e. for $\vec{y} \in (\Delta_K^N \cup \{\vec{0}_K\})^N$ we have $N(\vec{y}) = \vec{y}^\top \vec{1}_{NK}$. Observe the following:

$$\Pr(Y \mid \vec{z}, \vec{b}) \propto \prod_{w=1}^{W} \mathbf{1}\left(\frac{1}{N(\vec{y}_w)}\vec{y}_w^\top \vec{z} = b_w\right), \quad \Pr(\vec{z} \mid \vec{\tau}) \propto \exp(-\vec{z}^\top \log \vec{z}) \prod_{j=1}^{K} \mathbf{1}\left(\frac{1}{N}\vec{e}_j^\top \vec{z} = \tau_j\right),$$

Let $\rho_{w1}$, $\rho_{w2}$ represent the first ($\alpha$) and second ($\beta$) Beta hyperparameters for LF $h^{(w)}$.

$$\Pr(\vec{\tau} \mid \vec{\alpha}) \propto \prod_{j=1}^{K} \tau_j^{\alpha_j - 1}, \qquad \Pr(\vec{b} \mid \vec{z}, \vec{\rho}) \propto \prod_{w=1}^{W} \frac{b_w^{\rho_{w1}-1}(1 - b_w)^{\rho_{w2}-1}}{I_{b_{max}} - I_{b_{min}}}$$

where

$$b_{max} := \max_{i \in [N], j \in [K]} z_{ij} \qquad \text{and} \qquad b_{min} := \min_{i \in [N], j \in [K]} z_{ij}$$

and $I_x$ is shorthand for the Beta CDF with hyperparameters $\alpha, \beta$:

$$I_x = I_x(\alpha, \beta) = \frac{B(x; \alpha, \beta)}{B(\alpha, \beta)} = \frac{1}{B(\alpha, \beta)} \int_0^x t^{\alpha-1}(1-t)^{\beta-1} dt.$$

Putting everything together, the posterior is proportional to the following

$$\Pr(\vec{z}, \vec{b}, \vec{\tau} \mid Y, \vec{\alpha}, \vec{\rho}) \propto \prod_{w=1}^{W} \mathbf{1}\left(\frac{1}{N(\vec{y}_w)} \vec{y}_w^\top \vec{z} = b_w\right) \prod_{w=1}^{W} \frac{b_w^{\rho_{w1}-1}(1 - b_w)^{\rho_{w2}-1}}{I_{b_{max}} - I_{b_{min}}}$$

$$\times \exp(-\vec{z}^\top \log \vec{z}) \prod_{j=1}^{K} \mathbf{1}\left(\frac{1}{N} \vec{e}_j^\top \vec{z} = \tau_j\right) \prod_{j=1}^{K} \tau_j^{\alpha_j - 1}.$$

Now, we show that the posterior distribution as stated above is not log-concave. Specifically, the term $1/(I_{b_{max}} - I_{b_{min}})$ is what gives us trouble as that difference depends on $\vec{z}$.

**Lemma C.1.** *The posterior parameter distribution (over $\vec{z}, \vec{b}, \vec{\tau}$) of our model is not log-concave in $\vec{z}$.*

*Proof.* We consider the case where $\Pr(\vec{z}, \vec{b}, \vec{\tau} \mid Y, \vec{\alpha}, \vec{\rho}, L) > 0$, meaning $\vec{b}$ and $\vec{\tau}$ can be written in terms of $\vec{z}$. The problem comes from the distributions from which the accuracies are drawn, $Beta_{\vec{z}}(\alpha, \beta)$. Suppose we fixed the LF index $w$ and dropped its index in $\vec{y}_w$. Also, say $\alpha = \rho_{w1}$ and $\beta = \rho_{w2}$ to make things cleaner. Finally, lets suppose that LF $h^{(w)}$ predicts on all $N$ points. The pdf of $Beta_{\vec{z}}(\alpha, \beta)$ for $\frac{1}{N} \vec{y}^\top \vec{z} \in [0, 1]$ is then

$$\frac{(\frac{1}{N} \vec{y}^\top \vec{z})^{\alpha-1}(1 - \frac{1}{N} \vec{y}^\top \vec{z})^{\beta-1}}{(I_{b_{max}} - I_{b_{min}})B(\alpha, \beta)}.$$

For this distribution to be log concave, the logarithm of the pdf must be concave. Only looking at the non-constant terms,

$$(\alpha - 1) \log\left(\frac{1}{N} \vec{y}^\top \vec{z}\right) + (\beta - 1) \log\left(1 - \frac{1}{N} \vec{y}^\top \vec{z}\right) - \log(I_{b_{max}} - I_{b_{min}})$$

has to be concave.

We can show an instantiation of the above fails to satisfy the inequality required for concavity. Define $f(\vec{z}; \alpha, \beta, \vec{y})$ as the above. (Recall $I_{b_{max}}, I_{b_{min}}$ are functions of $\vec{z}$.) Take $\alpha = 10$, $\beta = 6$, $\theta = 0.5$, $\vec{z} = (0.9, 0.1)^\top$, $\vec{z}' = (0.55, 0.45)^\top$, and $\vec{y} = (0.5, 0.5)^\top$ ($K = 2$, $N = 1$). For the pdf of $Beta_{\vec{z}}(\alpha, \beta)$ to be log concave, we must have

$$f(\theta \vec{z} + (1 - \theta)\vec{z}'; \alpha, \beta, \vec{y}) \geq \theta f(\vec{z}; \alpha, \beta, \vec{y}) + (1 - \theta)f(\vec{z}'; \alpha, \beta, \vec{y}).$$

For the given values, the right hand side is *bigger* than the left hand side by about $0.612$. Thus, we have an instance where the required inequality for log-concavity does not hold. $\square$

If we just remove the $1/(I_{b_{max}} - I_{b_{min}})$ term from the Bayesian BF parameter posterior (Equation 4), then the resulting expression will be log concave. We'll still call the result a "posterior".

**Lemma C.2** (Lemma 4.1). *If we remove the normalization constant for $Beta_{\vec{z}}(\alpha, \beta)$, i.e. the $I_{b_{max}} - I_{b_{min}}$ quantity, then the resulting "posterior" as follows is log-concave in $\vec{z}$ given that each element of $\vec{\rho}$ and $\vec{\alpha}$ is at least 1.*

$$\exp(-\vec{z}^\top \log \vec{z}) \prod_{w=1}^{W} \left(\frac{1}{N(\vec{y}_w)} \vec{y}_w^\top \vec{z}\right)^{\rho_{w1}-1} \left(1 - \frac{1}{N(\vec{y}_w)} \vec{y}_w^\top \vec{z}\right)^{\rho_{w2}-1} \prod_{j=1}^{K} \left(\frac{1}{N} \vec{e}_j^\top \vec{z}\right)^{\alpha_j - 1}$$

The condition stated in the above lemma is essentially the condition for a Beta/Dirichlet pdf to be log-concave.

Before showing this, we need a supporting lemma.

**Lemma C.3.** *Suppose $r \geq 0$, $\vec{z} \in (\Delta_K \cup \vec{0}_K)^N$, and $\frac{1}{N(\vec{y})}\vec{y}^\top \vec{z} > 0$. Then,*

$$f(\vec{y}) = \left(\frac{1}{N(\vec{y})}\vec{y}^\top \vec{z}\right)^r$$

*is log-concave in $\vec{z}$.*

*Proof.* Since $r$ is non-negative, it suffices to show that

$$\log\left(\frac{1}{N(\vec{y})}\vec{y}^\top \vec{z}\right)$$

is concave. From Boyd and Vandenberghe [2004], page 84, we have a non-decreasing function (log) and a concave function (dot product). Therefore, their composition is concave in $\vec{z}$. □

*Remark C.4.* The above Lemma also holds if the argument inside the logarithm is $1 - \vec{y}^\top \vec{z}/N(\vec{y})$ because this is also concave in $\vec{z}$. For our purposes, that difference will always be non-negative.

*Proof of Lemma C.2.*

$$\exp(-\vec{z}^\top \log \vec{z}) \prod_{w=1}^{W}\left(\frac{1}{N(\vec{y})}\vec{y}_w^\top \vec{z}\right)^{\rho_{w1}-1}\left(1 - \frac{1}{N(\vec{y})}\vec{y}_w^\top \vec{z}\right)^{\rho_{w2}-1}\prod_{j=1}^{K}\left(\frac{1}{N}\vec{e}_j^\top \vec{z}\right)^{\alpha_j-1}. \quad (6)$$

To show that the posterior is log-concave, it suffices to show every element of the product is log-concave as the product of log-concave functions remains log-concave.

To begin, we take the exponential term on the left hand side ($-\vec{z}^\top \log \vec{z}$ term). This is clearly log-concave. For all the remaining terms, it suffices to apply Lemma C.3 for each one. Note that we have used the assumption that $\rho_{w1}, \rho_{w2}, \alpha_j \geq 1$ for all $w \in [W]$ and $j \in [K]$. Therefore, the posterior is log-concave in $\vec{z}$ as claimed. □

# D  The Remaining Theoretical Results

We now prove the remaining results: the form of the BBF prediction, consistency, and rates of convergence. From here onwards, we will drop the vector notation for $\eta, g, z, y_w$ when we will only need to reference the entire vector. There will be the odd situation where we'll need to reference a vector within a vector, e.g. $\vec{\eta}_i \in \Delta_K$, the true conditional label distribution for $x_i$.

Recall that we take the mode of the modified BBF posterior label distribution. That is, $g^{bbf}$ is the solution to the following convex program:

$$\min_{z \in \Delta_K^N}\left[z^\top \log z - \sum_{w=1}^{W}\left[(\rho_{w1} - 1)\log\left(\frac{1}{N(\vec{y}_w)}y_w^\top z\right) + (\rho_{w2} - 1)\log\left(1 - \frac{1}{N(\vec{y}_w)}y_w^\top z\right)\right]\right.$$
$$\left. - \sum_{j=1}^{K}(\alpha_j - 1)\log\left(\frac{1}{N}e_j^\top z\right)\right]. \quad (7)$$

I.e. the modified log-posterior was log-concave, so we just negated it to get a minimization problem.

With the convex program to solve clearly defined, the overarching goal is to bound the distance between $\eta$ and $g^{bbf}$. Recall that we are using KL divergence so that

$$d(\vec{\eta}, \vec{g}^{bbf}) = \sum_{i=1}^{N} KL(\vec{\eta}_i, \vec{g}_i^{bbf}).$$

Once $\eta$ and the LF predictions are fixed, we will derive the relationship between $\vec{\alpha}, \vec{\rho}$, the hyperparameters we control, and $d(\eta, g^{bbf})$. In the process of deriving such a bound, we'll be able to prove the remaining claimed results.

To set the stage, Lemma 5 of [An and Dasgupta, 2024] shows that

$$d(\eta, g^{bbf}) = d(\eta, g^*) + d(g^*, g^{bbf}).$$

If we can provide an upper bound for the LHS in the form of

$$d(\eta, g^{bbf}) \leq d(\eta, g^*) + \mathcal{A} + \mathcal{B},$$

then we know that $d(g^*, g^{bbf}) \leq \mathcal{A} + \mathcal{B}$. Since

$$d(\eta, g^{bbf}) = -\eta^\top \log g^{bbf} + \eta^\top \log \eta,$$

it suffices to demonstrate what the values of $\mathcal{A}$ and $\mathcal{B}$ are so that

$$-\eta^\top \log g^{bbf} \leq -g^{bbf\top} \log g^{bbf} + \mathcal{A} + \mathcal{B}.$$

**Theorem D.1** (Informal version of Theorem 5.3).

$$-\eta^\top \log g^{bbf} \leq -g^{bbf\top} \log g^{bbf} + \mathcal{A} + \mathcal{B}.$$

*for*

$$\mathcal{A} = |\theta^{bbf}|^\top |b^* - Ag^{bbf}| \quad and \quad \mathcal{B} = |\theta^*|^\top |b^* - Ag^{bbf}|$$

*where the absolute values are elementwise and $\theta^{bbf}, \theta^* \in \mathbb{R}^{W+K}$ are the respective weights bringing about the predictions $g^{bbf} = g^{(\theta^{bbf})}$ and $g^* = g^{(\theta^*)}$.*

It's after proving this result that we have the machinery to prove that $g^{bbf}$ has the same form of solution as BF and that BBF is consistent.

*Proof Sketch of Theorem D.1.* The idea is to use the analysis done in [An and Dasgupta, 2024] to bound BBF's error. Namely, the cross entropy term on the LHS of the theorem statement is not easy to analyze. Thus, we use a proxy: $-g^{bbf\top} \log g^{bbf}$ and upper bound that. However, unlike for $g^{bf}$, it's not guaranteed that

$$-\eta^\top \log g^{bbf} \leq -g^{bbf\top} \log g^{bbf}.$$

Thus, we show a term $\mathcal{A}$ which when added to the right hand side makes the above inequality always hold. Now, it suffices to upper bound $-g^{bbf\top} \log g^{bbf}$ by

$$-\eta^\top \log g^* + \mathcal{B}.$$

To do this, we'll first define a class of BF problems with polytope $P_{bar}$, which has a specific form. (This class of BF problems will depend on the same LF predictions and $\eta$ as BBF). For fixed BBF hyperparameters $\vec{\alpha}, \vec{\rho}$, we'll fix a $P_{bar}$. The solution to a BF problem that uses $P_{bar}$, will be called $g^{bar}$. Namely, $g^{bar}$ is the solution to

$$V_{bar} = \max_{g \in \Delta_K^N} \min_{z \in P_{bar}} z^\top \log g = \min_{z \in P_{bar}} z^\top \log z$$

with respect to $g$. We'll show that $-g^{bbf\top} \log g^{bbf} \leq -g^{bar\top} \log g^{bar}$. This is done by demonstrating that $g^{bbf}$ almost satisfies the KKT conditions of $V_{bar}$. The KKT conditions are 'almost' satisfied in a specific way so that we can use techniques showing how an interior point method can be used to solve convex programs to show the above inequality. I.e. the terms after the entropy term in the BBF objective (Equation 7) are a log-barrier approximation of $P_{bar}$.

Following the analysis of An and Dasgupta [2024], we'll then show that

$$-g^{bar\top} \log g^{bar} \leq -\eta^\top \log g^* + \mathcal{B}$$

where $\mathcal{B} = |\theta^*|^\top |b^* - Ag^{bbf}|$.

Putting everything together, the chain of inequalities is as follows:

$$-\eta \log g^{bbf} \leq -g^{bbf\top} \log g^{bbf} + \mathcal{A} \leq -g^{bar\top} \log g^{bar} + \mathcal{A} \leq -\eta^\top \log g^* + \mathcal{A} + \mathcal{B}.$$

□

Once we specify the BF problem $V_{bar}$, we can show that the BBF solution has the same functional form as the BF solution. Recall that we can define a set of predictions $\mathcal{G} = \{g^{(\theta)} \colon \theta \in \mathbb{R}^{W+K}\}$ where

$$g_{ij}^{(\theta)} = \frac{\exp((A^\top \theta)_{ij})}{\exp(\sum_{j'=1}^{K}(A^\top \theta)_{ij'})} \propto \exp((A^\top \theta)_{ij}).$$

**Lemma D.2** (Lemma 5.1). $g^{bbf} \in \mathcal{G}$.

*Proof Sketch.* In the last proof sketch, we mentioned how $g^{bbf}$ almost satisfies all KKT conditions for $V_{bar}$. In particular, it satisfies the zero-gradient condition. The zero-gradient condition is what determines the form of solution. So, since BF and BBF satisfy the same condition, they have the same form of solution. An and Dasgupta [2024] show (in Theorem 2) that $g^{bf} \in \mathcal{G}$, so $g^{bbf} \in \mathcal{G}$. $\square$

Analysis of the KKT conditions that $g^{bbf}$ satisfies is what will let us show BBF's consistency. What we'll need is that the prior distributions of BBF to concentrate around the empirical LF accuracies and empirical class frequency distribution. What will happen is that polytope being approximated in the BBF objective will tend to $P^* = \{z \in \Delta_K^N \colon Az = b^*\}$, where

$$g^* = \arg\min_{z \in P^*} z^\top \log z.$$

(The above result was shown in Theorem 6 by An and Dasgupta [2024].) This means that $g^{bbf} \to g^*$, the best approximator in KL divergence to $\eta$.

**Theorem D.3** (Informal version of Theorem 5.2). *If $\vec{\alpha}$ and $\vec{\rho}$ each tend to infinity elementwise and grow in such a way that $Ag^{bbf} \to b^*$, then $g^{bbf} \to g^*$.*

*Proof Sketch.* We'll show that as the BBF hyperparameters grow as in the theorem statement, the KKT conditions $g^{bbf}$ satisfies tends to the KKT conditions that $g^*$ satisfy. I.e. the KKT conditions for

$$g^* = \arg\min_{z \in P^*} z^\top \log z.$$

Thus, $g^{bbf} \to g^*$. $\square$

The steps of the above proofs are laid out as follows.

- Subsection D.1, demonstration of what $\mathcal{A}$ is.
- Subsection D.2, definition of $P_{bar}$ and associated BF problem.
- Subsection D.3, some properties of BF with $P_{bar}$, specifically the dual program and KKT conditions.
- Subsection D.4, some preliminaries on log-barrier approximations of a convex program's feasible region. This will help simplify the argument showing how BBF is related to BF with $P_{bar}$.
- Subsection D.5, demonstration of how BBF's objective contains a log-barrier approximation of $P_{bar}$. It'll be proven here that $-g^{bbf\top} \log g^{bbf} \leq -g^{bar\top} \log g^{bar}$ where $g^{bar}$ is the solution to BF with $P_{bar}$. This is also where we'll show that $g^{bbf}$ has the same functional form as the BF solutions (Lemma 5.1, in Subsubsection D.5.2).
- Subsection D.6, upper bounding $-g^{bar\top} \log g^{bar}$, i.e. deriving the value of $\mathcal{B}$.
- Subsection D.7, formal proof of Theorem D.1.
- Subsection D.8, proof of BBF's consistency (Theorem 5.2).

## D.1 Bounding the Value of $\mathcal{A}$

Say that the accuracies gotten from labeling $g^{bbf}$ is $\widehat{b} := Ag^{bbf}$. From [An and Dasgupta, 2024] Corollary 28, we know that $Ag^* = b^*$. From that same paper, Lemma 26 provides the following useful identity.

**Lemma D.4.** *Suppose $g = g^{(\theta)} \in \mathcal{G}$ and $y, y' \in \Delta_K^N$ are arbitrary.*

$$(y - y')^\top \log g = (Ay - Ay')^\top \theta.$$

**Lemma D.5.** $\mathcal{A} = |\theta^{bbf}|^\top |b^* - \widehat{b}|$

*Proof.* For $-g^{bbf\top} \log g^{bbf}$ to be an upper bound for $-\eta^\top \log g^{bbf}$, we would need

$$-g^{bbf\top} \log g^{bbf} + \eta^\top \log g^{bbf} \geq 0.$$

Choosing $y = \eta$ and $y' = g^{bbf}$ from Lemma D.4, the above inequality is equivalent to

$$(b^* - \widehat{b})^\top \theta^{bbf} \geq 0$$

where $A\eta = b^*$ and $Ag^{bbf} = \widehat{b}$. While that quantity could be non-negative in and of itself, the following bound always holds.

$$-g^{bbf\top} \log g^{bbf} + |\theta^{bbf}|^\top |b^* - \widehat{b}| \geq -\eta^\top \log g^{bbf}$$

The second term on the left hand side is our $\mathcal{A}$. $\qquad\square$

### D.2 BF with Polytope $P_{bar}$

In this section, we want to define a class of BF problems by way of a polytope with a specific form. This will allow us to use techniques associated with the barrier method (an interior point method used to solve convex programs, [Boyd and Vandenberghe, 2004], Chapter 11) to Bayesian BF.

Recall that given the LF predictions and $\eta$, the polytope that BF uses is constructed by bounding the LF accuracies and the class frequency distribution. These bounds can be written as linear inequalities where any point satisfying all inequalities lies within the polytope. Moreover, we want to have $\eta$ be in the polytope. Equivalently, we want any estimate of LF accuracies/class frequencies to contain $b^*$.

So, say that the LF accuracies/class frequencies as estimated by the Bayesian BF solution is $\widehat{b} := Ag^{bbf}$. $P_{bar}$ will be defined by taking the constraint $b^* \leq Az \leq b^*$ and expanding the bounds until said bounds barely contain $\widehat{b}$. Specifically, for an arbitrary $1 \gg \delta > 0$ and each $i \in [W + K]$, define

$$\epsilon_i^- := \mathbf{1}(b_i^* > \widehat{b}_i)(b_i^* - \widehat{b}_i) + \delta \quad \text{and} \quad \epsilon_i^+ := \mathbf{1}(\widehat{b}_i > b_i^*)(\widehat{b}_i - b_i^*) + \delta.$$

We claim that $\widehat{b} \in (b^* - \epsilon^-, b^* + \epsilon^+)$. For a fixed index $i$, if $b_i^* > \widehat{b}_i$, then

$$\epsilon_i^- = b^* - \widehat{b}_i + \delta \quad \text{and} \quad \epsilon_i^+ = \delta.$$

The $i^{th}$ interval is

$$(b_i^* - \epsilon_i^-, b_i^* + \epsilon_i^+) = (b_i^* - b_i^* + \widehat{b}_i - \delta, b_i^* + \delta) = (\widehat{b}_i - \delta, b^* + \delta)$$

which clearly contains $\widehat{b}_i$. A similar argument can be made when $\widehat{b}_i > b_i^*$ and the case of $\widehat{b}_i = b_i^*$ is trivial.

Formally, the polytope $P_{bar}$ which brings about $g^{bar}$ as the optimal BF solution is

$$P_{bar} = \{z \in \Delta_K^N \colon b^* - \epsilon^- \leq Az \leq b^* + \epsilon^+\}.$$

The BF problem that $g^{bar}$ is the solution to is as follows

$$g^{bar} := \arg\min_{z \in P_{bar}} z^\top \log z. \tag{8}$$

We shall connect the Lagrangian of the Bayesian BF objective (Equation 7) at optimality to the KKT conditions of the above BF problem. Before we do that, we provide an easy example that is illustrative of the argument that will be used.

### D.3  Dual Program and KKT conditions for BF with $P_{bar}$

In this section, we will show the dual problem of the BF problem in consideration as well as the KKT conditions. We'll need these in order to relate BBF to BF with $P_{bar}$.

The dual problem is easily derivable following the argument given for BF (following Theorem 2 of [An and Dasgupta, 2024]). Say that $\vec{0}_{W+K}$ is the zeros vector of length $W + K$.

**Lemma D.6.** *The dual problem for Equation 8 is*

$$\max_{\sigma,\sigma' \geq \vec{0}_{W+K}} \left[ \sigma^\top \left( -b^* - \epsilon^+ \right) + \sigma'^\top \left( b^* - \epsilon^- \right) - \sum_{i=1}^{N} \log \left( \sum_{j=1}^{K} \exp \left( [A^\top (\sigma' - \sigma)]_{ij} \right) \right) \right]$$

Before stating the KKT conditions for the problem in Equation 8, define $D \in \{0,1\}^{N \times NK}$ where $[Dz]_1 = \sum_{j=1}^{K} z_{1j}$. The constraint $D\vec{z} = \vec{1}_N$ ensures that each of the $\vec{z}_i$'s sum to 1. We will not need non-negativity constraints for each of the $z_{ij}$'s because their functional form guarantees their non-negativity.

**Lemma D.7.** *The KKT conditions for the BF problem in Equation 8 are:*

$$\frac{1}{N(\vec{y}_w)} y_w^\top z - b^* - \epsilon^+ \leq 0, \quad w \in [W + K]$$

$$-\frac{1}{N(\vec{y}_w)} y_w^\top z + b^* - \epsilon^- \leq 0, \quad w \in [W + K]$$

$$Dz - \vec{1}_n = \vec{0}_N$$

$$\sigma, \sigma' \geq \vec{0}_{W+K}$$

$$\sigma_w([Az]_w - b_w^* - \epsilon_w^+) = 0, \quad w \in [W + K]$$

$$\sigma'_w(-[Az]_w + b_w^* - \epsilon_w^-) = 0, \quad w \in [W + K]$$

$$\nabla_z \big[ z \cdot \log z + \sigma^\top (Az - b^* - \epsilon^+) + \sigma'^\top (-Az + b^* - \epsilon^-)$$

$$+ \xi^\top (Dz - \vec{1}_n) \big] = \vec{0}_{NK}.$$

Recall that notation has been abused and that $b^* \in [0,1]^{W+K}$ contains all empirical LF accuracies *and* empirical class frequencies.

### D.4  Preliminaries for Barrier Method Type Analysis

The content in this section closely follows Boyd and Vandenberghe [2004] Section 11.2.2. This exposes the general schema of the argument we'll use to relate BBF to BF, but there are differences.

Suppose we had a convex program

$$p^* = \min_{\substack{f_1(x) \leq 0 \\ Ax = b}} f_0(x) \tag{9}$$

where $x \in \mathbb{R}^d$ with feasible constraints. Call this our original problem. If we define the indicator function $I_- : \mathbb{R} \to \{0, \infty\}$ as

$$I_-(u) := \begin{cases} 0 & \text{if } u \leq 0 \\ \infty & \text{otherwise,} \end{cases}$$

then the above problem can be written as

$$p^* = \min_{Ax=b} f_0(x) + I_-(f_1(x)). \tag{10}$$

So long as the program is feasible, the indicator function forces the optimal $x$ to satisfy the constraint $f_1(x) \leq 0$. The logarithmic barrier is essentially a relaxation of the indicator function so it becomes continuous. Namely, $I_-(u)$ can be approximated by

$$\widehat{I}_-(u) := -\frac{1}{t} \log(-u)$$

for $t > 0$, which is convex and non-decreasing. If we say $\log(0) := \infty$, $\widehat{I}_-(u)$ and $I_-(u)$ are defined on the same domain. We can easily see that if $u > 0$ is fixed, then as $t \to \infty$, $\widehat{I}_-(u) \to I_-(u)$.

We may approximate the problem with the indicator function (Equation 10) with

$$p(t) = \min_{Ax=b} f_0(x) - \frac{1}{t}\log(-f_1(x)). \tag{11}$$

Then, as $t \to \infty$, $p(t) \to p^*$. Call the above problem the log-barrier problem.

Let $x^*$ be the optimal solution of Equation 9 and $x^*(t)$ be the optimal solution of Equation 11. One can show the following result by analyzing the Lagrangian of Equation 11 at the point $x^*(t)$. This argument below is the one we use for our results about $g^{bbf}$ and $g^{bar}$.

**Lemma D.8.** *Suppose $x^*(t)$ is the optimal solution to the convex program with log-barrier constraints (Equation 11). Then,*

1. *$x^*(t)$ is primal feasible for the original convex program (Equation 9)*

2. *$x^*(t)$ shows the existence of dual feasible variables $\sigma(t) = -\frac{1}{t}f_1(x^*(t))$ (for $f_1(x) \leq 0$) and $\widehat{\nu}$ (for $Ax = b$) for the original problem.*

3. *$f_0(x^*) \leq f_0(x^*(t)) \leq f_0(x^*) + \frac{1}{t}$.*

*Proof.* Consider the program in Equation 11 at optimality. There, its Lagrangian must have zero gradient at $x^*(t)$. We mean that there exists a $\widehat{\nu} \in \mathbb{R}^d$ where

$$\nabla_x \widehat{L}(x, \widehat{\nu}) = \nabla_x \left[ f_0(x) - \frac{1}{t}\log(-f_1(x)) + \widehat{\nu}^\top(Ax - b) \right] = \vec{0}_d$$

when $x^*(t)$ is substituted for $x$. Evaluating, that's

$$\nabla f_0(x) - \frac{1}{t f_1(x^*(t))}\nabla f_1(x^*(t)) + A^\top \widehat{\nu} = \vec{0}_d$$

Since the original problem has a feasible solution, $f_1(x^*(t)) < 0$ because that is the only time $\widehat{I}_-$ is not infinite or undefined. By assumption, $t > 0$ so that $\sigma(t) = -1/(t f_1(x^*(t))) > 0$. Because $f_1(x^*(t)) < 0$, $x^*(t)$ is also primal feasible for the original problem. If we substitute $x^*(t)$ for $x$, $\sigma(t)$ for $\sigma$ and $\widehat{\nu}$ for $\nu$, we have shown the following equations are satisfied.

$$f_1(x) \leq 0$$
$$Ax - b = \vec{0}$$
$$\sigma \geq 0$$
$$\sigma f_1(x) = -\frac{1}{t}$$
$$\nabla \left[ f_0(x) + \sigma f_1(x) + \nu^\top(Ax - b) \right] = \vec{0}_d.$$

One immediately observes that these are the KKT conditions for the original problem (Equation 9) where instead of the usual complementary slackness condition $\sigma f_1(x) = 0$, we have it equal to $-\frac{1}{t}$.

We now want to show that $\sigma(t), \widehat{\nu}$ are dual feasible for the original problem. The Lagrangian for the original problem is
$$L(x, \sigma, \nu) = f_0(x) + \sigma f_1(x) + \nu^\top(Ax - b).$$
The zero gradient condition that is satisfied above means that

$$\nabla L(x^*(t), \sigma(t), \widehat{\nu}) = \vec{0}_d.$$

Since $x^*(t)$ is the minimizer of $L(\cdot, \sigma(t), \widehat{\nu})$, $\sigma(t)$ and $\widehat{\nu}$ are dual feasible for the original problem.

Now, we show the final claim, bounding $f_0(x^*(t))$. Because $\sigma(t)$ and $\widehat{\nu}$ are dual feasible, the dual function for the original program at those values is

$$g(\sigma(t), \widehat{\nu}) = f_0(x^*(t)) + \sigma(t)f_1(x^*(t)) + \widehat{\nu}^\top(Ax^*(t) - b).$$

Because $Ax = b$ is one of the constraints to the log barrier function, the last term is 0. By definition, the second term will evaluate to $-\frac{1}{t}$. Thus, $g(\sigma(t), \widehat{\nu}) = f_0(x^*(t)) - \frac{1}{t}$. Now, by weak duality,

$$g(\sigma(t), \widehat{\nu}) \leq p^* = f_0(x^*) \quad \text{meaning} \quad f_0(x^*(t)) - \frac{1}{t} \leq f_0(x^*).$$

This proves the upper bound to the third claim. To see the lower bound of the third claim, recall that $x^*(t)$ is primal feasible for the original problem and therefore is no smaller than the optimal minimum of the original problem $f_0(x^*)$. $\qquad\square$

Because we will use a similar argument for $g^{bbf}$ and $g^{bar}$, we now draw equivalences for the results in the above Lemma. $g^{bar}$ will be the solution of the "original" convex program while $g^{bbf}$ is a solution to a problem that implicitly encodes log-barrier constraints. Claim 2 of the above Lemma will imply that $g^{bbf}$ can be written in the form of a BF solution. Claims 1 and 3 will not only aid in our proof of Bayesian BF's consistency, but also show that the entropy of $g^{bar}$ upper bounds the entropy of $g^{bbf}$. Namely $-g^{bbf\top} \log g^{bbf} \leq -g^{bar\top} \log g^{bar}$.

### D.5  BBF's Objective has a Log-Barrier Approximation to $P_{bar}$

In this section, we'll show that the BBF prediction $g^{bbf}$ satisfies a modified version of the KKT conditions of $V_{bar}$ (Lemma D.7). Like in the above section, the only difference is that $g^{bbf}$ will satisfy a modified version of the complementary slackness conditions. This will be done in two steps. First, we'll ignore the class frequency prior (i.e. suppose that it is constant) and show that BBF with only LF accuracy priors satisfies those modified KKT conditions. Then, we'll extend the argument so the class frequency prior is considered.

So, we start with a version of BBF where the class frequency pdf is not considered.

$$\min_{z \in \Delta_K^N} \left[ z^\top \log z - \sum_{w=1}^{W} \left[ (\rho_{w1} - 1) \log \left( \frac{1}{N(\vec{y}_w)} y_w^\top z \right) + (\rho_{w2} - 1) \log \left( 1 - \frac{1}{N(\vec{y}_w)} y_w^\top z \right) \right] \right]$$

In restricting to dealing with LF accuracies, we note that the matrix $A$ will have only $W$ rows, one for each LF prediction. To simplify the notation, define the following:

$$\gamma_w(z) := \frac{1}{N(\vec{y}_w)} y_w^\top z, \quad \alpha_w' = \rho_{w1} - 1, \quad \text{and} \quad \beta_w' = \rho_{w2} - 1$$

Then, the convex program we first analyze can be rewritten as

$$\min_{z \in \Delta_K^N} \left[ z^\top \log z - \sum_{w=1}^{W} [\alpha_w' \log (\gamma_w(z)) + \beta_w' \log (1 - \gamma_w(z))] \right] \tag{12}$$

We first argue that $g^{bbf}$ is primal feasible for the BF problem with polytope $P_{bar}$. Then, we analyze the optimal objective of Bayesian BF with LF accuracies (Equation 12) in the lens of the aforementioned problem's KKT conditions (Lemma D.7).

**Lemma D.9.** *The optimal solution $g^{bbf}$ to the Bayesian BF objective with LF accuracies (Equation 12) is primal feasible for the BF problem with $P_{bar}$ (Equation 8).*

*Proof.* This is true by construction of $P_{bar}$. One can follow the argument given in Section D.2. $\quad\square$

**Theorem D.10.** *Fix $1 \gg \delta > 0$. For each LF $w \in [W]$, define the following*

$$\epsilon_w^- = \mathbf{1}(b_w^* > \gamma_w^*)(b_w^* - \gamma_w^*) + \delta, \quad \epsilon_w^+ := \mathbf{1}(\gamma_w^* > b_w^*)(\gamma_w^* - b_w^*) + \delta,$$

$$t_w' = \frac{\gamma_w^*}{\alpha_w'(\gamma_w^* - b_w^* + \epsilon_w^-)}, \quad t_w = \frac{1 - \gamma_w^*}{\beta_w'(-\gamma_w^* + b_w^* + \epsilon_w^+)},$$

$$\sigma_w' = -\frac{1}{t_w'(-\gamma_w^* + b_w^* - \epsilon_w^-)}, \quad \text{and} \quad \sigma_w = -\frac{1}{t_w(\gamma_w^* - b_w^* - \epsilon_w^+)}.$$

*If $g^{bbf}$ is the optimal solution for Equation 12 where $\alpha_w, \beta_w > 1$ for each $w \in [W]$, then the following deformed KKT conditions of Equation 8 are satisfied.*

$$\frac{1}{N(\vec{y}_w)} y_w^\top z - b_w^* - \epsilon_w^+ \leq 0, \quad w \in [W]$$

$$-\frac{1}{N(\vec{y}_w)} y_w^\top z + b_w^* - \epsilon_w^- \leq 0, \quad w \in [W]$$

$$Dz - \vec{1}_n = \vec{0}_N$$

$$\sigma, \sigma' \geq \vec{0}_W$$

$$\sigma_w([Az]_w - b_w^* - \epsilon_w^+) = -\frac{1}{t_w} \quad w \in [W]$$

$$\sigma'_w(-[Az]_w + b_w^* - \epsilon_w^-) = -\frac{1}{t'_w} \quad w \in [W]$$

$$\nabla_z \left[ z \cdot \log z + \sigma^\top(Az - b^* - \epsilon^+) + \sigma'^\top(-Az + b^* - \epsilon^-) + \xi^\top(Dz - \vec{1}_n) \right] = \vec{0}_{NK}.$$

*Note that $w$ only ranges in $[W]$ and not $[W + K]$ (as seen in Lemma D.7) because the final $K$ indices are for the class frequency constraints, which we aren't considering right now.*

*Proof.* We argue for this by analyzing the objective of our problem (Equation 12) at optimality. This is very similar to the demonstration argument we gave in Lemma D.8.

If we evaluate the zero-gradient condition in the theorem statement, we get

$$\vec{1}_{NK} + \log z + A^\top \sigma - A^\top \sigma' + D^\top \xi = \vec{0}_{NK}. \tag{13}$$

We'll show that the gradient of the Bayesian BF objective (Equation 12) will be of this form.

When the BBF objective is at optimality, the following will hold true for some $\xi \in \mathbb{R}^N$.

$$\vec{0}_{NK} = \nabla_z \left[ z^\top \log z - \sum_{w=1}^W \left[ \alpha'_w \log\left( \frac{y_w^\top z}{N(\vec{y}_w)} \right) + \beta'_w \log\left( 1 - \frac{y_w^\top z}{N(\vec{y}_w)} \right) \right] + \xi^\top(Dz - \vec{1}_N) \right]$$

Evaluating,

$$\vec{0}_{NK} = \vec{1}_{NK} + \log z - \sum_{w=1}^W \left[ \frac{\alpha'_w}{\frac{1}{N(\vec{y}_w)} y_w^\top z} \left( \frac{y_w}{N(\vec{y}_w)} \right) + \frac{\beta'_w}{1 - \frac{1}{N(\vec{y}_w)} y_w^\top z} \left( -\frac{y_w}{N(\vec{y}_w)} \right) \right] + D^\top \xi$$

where the division is elementwise. Observe that by definition, $y_w/N(\vec{y}_w)$ corresponds to row $w$ of matrix $A$. I.e. that term is related to the bound on LF $h^{(w)}$'s accuracy. Define

$$f'_w(z) = -\gamma_w(z) + b_w^* - \epsilon_w^- \quad \text{and} \quad f_w(z) = \gamma_w(z) - b_w^* - \epsilon_w^+.$$

If one sets these functions to each be less than or equal to 0, then one can see they represent upper and lower bounds for LF $h^{(w)}$'s accuracy. Using that notation and multiplying each term in each summand by 1 gives

$$\vec{0}_{NK} = \vec{1}_{NK} + \log z + D^\top \xi$$

$$+ \sum_{w=1}^W \left[ \frac{-\alpha'_w f'_w(z)}{\gamma_w(z)} \frac{1}{-f'_w(z)} \left( -\frac{y_w}{N(\vec{y}_w)} \right) + \frac{-\beta'_w f_w(z)}{1 - \gamma_w(z)} \frac{1}{-f_w(z)} \left( \frac{y_w}{N(\vec{y}_w)} \right) \right].$$

Moving forward, we'll replace $z$ by $g^{bbf}$ and use the following shorthand of $\gamma_w^*$ to denote $\gamma_w(g^{bbf}) = y_w^\top g^{bbf}/N(\vec{y}_w)$.

Now, define

$$t'_w = -\frac{\gamma_w^*}{\alpha'_w f'_w(g^{bbf})} = \frac{\gamma_w^*}{\alpha'_w(\gamma_w^* - b_w^* + \epsilon_w^-)} \quad \text{and} \quad t_w = -\frac{1 - \gamma_w^*}{\beta'_w f_w(g^{bbf})} = \frac{1 - \gamma_w^*}{\beta'_w(-\gamma_w^* + b_w^* + \epsilon_w^+)}.$$

Using this definition, the above is simplified to

$$\vec{0}_{NK} = \vec{1}_{NK} + \log g^{bbf} + D^\top \xi + \sum_{w=1}^{W} \left[ \frac{-1}{t'_w f'_w(g^{bbf})} \left( -\frac{y_w}{N(\vec{y}_w)} \right) + \frac{-1}{t_w f_w(g^{bbf})} \left( \frac{y_w}{N(\vec{y}_w)} \right) \right].$$

Define now

$$\sigma'_w = -\frac{1}{t'_w f'_w(g^{bbf})} = -\frac{1}{t'_w(-\gamma^*_w + b^*_w - \epsilon^-_w)} \text{ and } \sigma_w = -\frac{1}{t_w f_w(g^{bbf})} = -\frac{1}{t_w(\gamma^*_w - b^*_w - \epsilon^+_w)}.$$

By Lemma D.9, $g^{bbf} \in P_{bar}$, which means that by construction, $f_w(g^{bbf}), f'_w(g^{bbf})$ are both negative. Since $\alpha_w, \beta_w > 1$ for each $w \in [W]$ by assumption, the $t$'s are strictly positive. This means that the $\sigma$ and $\sigma'$ values are all strictly positive.

We can then write

$$\vec{0}_{NK} = \vec{1}_{NK} + \log g^{bbf} + D^\top \xi + \sum_{w=1}^{W} \left[ -\sigma'_w \left( \frac{y_w}{N(\vec{y}_w)} \right) + \sigma_w \left( \frac{y_w}{N(\vec{y}_w)} \right) \right].$$

To see that this matches the form of Equation 13, observe that $y_w/N(\vec{y}_w)$ is the $w^{th}$ row of $A$ or the $w^{th}$ column of $A^\top$. This means we have verified the zero gradient condition of the deformed KKT conditions in the theorem statement.

We now verify the remaining conditions. The first three conditions state that $g^{bbf}$ must be primal feasible with respect to the BF problem with $P_{bar}$. That was shown in Lemma D.9. The third condition is a relaxation of requiring $g^{bbf}$ to be a concatenation of $N$ distributions, each over $K$ elements. It is satisfied because $g^{bbf} \in \Delta_K^N$. In defining the $\sigma$'s, we showed that they were strictly positive. Hence, the fourth condition is satisfied. Finally, one can see that the modified complementary slackness conditions hold by substituting $\gamma^*_w = y_w^\top g^{bbf}/N(\vec{y}_w)$ for $[Az]_w$ and observing the definition of $\sigma_w, \sigma'_w$. $\qquad\square$

*Remark* D.11. It doesn't matter what the value of $\xi$ is as these dual variables ensure that the solution to a BF problem is a distribution, i.e. is in $\Delta_K^N$. Thus, we will ignore $\xi$ when talking about BF's dual variables.

### D.5.1 Extending the argument for class frequencies

Now, we analyze BBF as presented in Equation 7. Abusing notation, define $\alpha'_j = \alpha_j - 1$. The modified version of our objective (coming from Equation 12) is

$$\min_{z \in \Delta_K^N} \left[ z^\top \log z - \sum_{w=1}^{W} [(\rho_{w1} - 1) \log(\gamma_w(z)) + (\rho_{w2} - 1) \log(1 - \gamma_w(z))] \right.$$
$$\left. - \sum_{j=1}^{K} \alpha'_j \log \left( \frac{1}{N} e_j^\top z \right) \right]. \quad (14)$$

The goal is to show that by adding the last sum, we satisfy a modified version of the KKT conditions with upper and lower bounds on the class frequency constraints. Unsurprisingly, the addition is $2K$ more primal constraints, the $2K$ more dual variables, and a modified zero-gradient condition.

Before stating the result, we remind the reader of some notation and introduce some notation. $\omega^*_j = \frac{1}{N} e_j^\top \eta$, the empirical class frequency. Let $A'$ be the matrix in $\{0,1\}^{K \times NK}$ where row $j$ is $\frac{1}{N} \vec{e}_j$. This will make the notation in the KKT conditions simpler. Define $\pi_j(z) = \frac{1}{N} e_j^\top z$ as the frequency of class $j$ with $z$ as the labeling. Similar to above, $\pi^*_j = \pi_j(g^{bbf})$ where we now take $g^{bbf}$ to be the solution of Equation 7. Finally, define $\alpha'_0 = \sum_{j=1}^{K} \alpha'_j$.

**Theorem D.12.** *Fix $1 \gg \delta > 0$. For each class $j \in [K]$, define the following*

$$\zeta^-_j = \mathbf{1}(\omega^*_j > \pi^*_j)(\omega^*_j - \pi^*_j) + \delta, \quad \zeta^+_j := \mathbf{1}(\pi^*_j > \omega^*_j)(\pi^*_j - \omega^*_j) + \delta,$$

$$\tau'_j = \frac{\pi^*_j}{\alpha'_0(\pi^*_j - \omega^*_j + \zeta^-_j)}, \quad \tau_j = \frac{\pi^*_j}{(\alpha'_0 - \alpha'_j)(-\pi^*_j + \omega^*_j + \zeta^+_j)},$$

$$v'_j = -\frac{1}{\tau'_j(-\pi^*_j + \omega^*_j - \zeta^-_j)}, \quad \text{and} \quad v_j = -\frac{1}{\tau_j(\pi^*_j - \omega^*_j - \zeta^+_j)}.$$

*Suppose that each element of $\vec{\rho}$ and $\vec{\alpha}$ is no smaller than 1. Then, if $g^{bbf}$ is the optimal solution to Equation 7, then the following deformed KKT conditions of Equation 8 (found at Lemma D.7) are satisfied. The $t, t', \sigma, \sigma'$ values are as defined in Theorem D.10.*

$$\frac{1}{N(\vec{y}_w)}y_w^\top z - b^*_w - \epsilon^+_w \leq 0, \quad w \in [W]$$

$$-\frac{1}{N(\vec{y}_w)}y_w^\top z + b^*_w - \epsilon^-_w \leq 0, \quad w \in [W]$$

$$\frac{1}{N}e_j^\top z - \omega^*_j - \zeta^+_j \leq 0 \quad j \in [K]$$

$$-\frac{1}{N}e_j^\top z + \omega^*_j - \zeta^-_j \leq 0 \quad j \in [K]$$

$$Dz - \vec{1}_n = \vec{0}_N$$

$$\sigma, \sigma' \geq \vec{0}_W$$

$$v, v' \geq \vec{0}_K$$

$$\sigma_w([Az]_w - b^*_w - \epsilon^+_w) = -\frac{1}{t_w} \quad w \in [W]$$

$$\sigma'_w(-[Az]_w + b^*_w - \epsilon^-_w) = -\frac{1}{t'_w} \quad w \in [W]$$

$$v_j([A'z]_j - \omega^*_j - \zeta^+_j) = -\frac{1}{\tau_j} \quad j \in [K]$$

$$v'_j(-[A'z]_j + \omega^*_j - \zeta^-_j) = -\frac{1}{\tau'_j} \quad j \in [K]$$

$$\nabla_z \Big[z \cdot \log z + \sigma^\top(Az - b^* - \epsilon^+) + \sigma'^\top(-Az + b^* - \epsilon^-)$$
$$+ v^\top(A'z - \omega^* - \zeta^+) + v'^\top(-A'z + \omega^* - \zeta^-) + \xi^\top(Dz - \vec{1}_n)\Big] = \vec{0}_{NK}.$$

*Proof.* We essentially follow the same proof as Theorem D.10. Namely, we choose the same $\epsilon$'s, $t$'s, and $\sigma$'s. For this proof, we focus on lines 3, 4, 7, 10, 11, 12. Line 12 is the zero gradient condition, which we start with. The gradient of term in the BBF objective we're interested in is

$$\nabla_z \left[-\sum_{j=1}^K \alpha'_j \log\left(\frac{1}{N}e_j^\top z\right)\right] = \sum_{j=1}^K \frac{\alpha'_j}{\frac{1}{N}e_j^\top z}\left(-\frac{1}{N}e_j\right).$$

Once again the division is elementwise. It suffices to show that the above is equal to the terms involving $v$ in the zero-gradient condition because we have shown the correspondence of the other terms in Theorem D.10. By adding 0, we can see the above is equal to

$$\sum_{j=1}^K \left[\frac{\alpha'_0}{\frac{1}{N}e_j^\top z}\left(-\frac{1}{N}e_j\right) + \frac{\alpha'_0 - \alpha'_j}{\frac{1}{N}e_j^\top z}\left(\frac{1}{N}e_j\right)\right]$$

Now, define

$$f'_j(z) = -\pi_j(z) + \omega^*_j - \zeta^-_j \quad \text{and} \quad f_j(z) = \pi_j(z) - \omega^*_j - \zeta^+_j.$$

Like before, setting these to be less than 0 form the class frequency constraints of BF. Now, it suffices to choose the $\tau$'s and $v$'s. The above expression can be rewritten as

$$\sum_{j=1}^K \left[\frac{-\alpha'_0 f'_j(z)}{\pi_j(z)}\frac{1}{-f'_j(z)}\left(-\frac{1}{N}e_j\right) + \frac{-(\alpha'_0 - \alpha'_j)f_j(z)}{\pi_j(z)}\frac{1}{-f_j(z)}\left(\frac{1}{N}e_j\right)\right]$$

As the zero-gradient condition holds for the optimal solution to Equation 7, we substitute $g^{bbf}$ for $z$. Then, with $\pi^*_j = \pi_j(g^{bbf})$, we define

$$\tau'_j = -\frac{\pi^*_j}{\alpha'_0 f'_j(g^{bbf})} = \frac{\pi^*_j}{\alpha'_0(\pi^*_j - \omega^*_j + \zeta^-_j)}$$

and

$$\tau_j = -\frac{\pi_j^*}{(\alpha_0' - \alpha_j')f_j(g^{bbf})} = \frac{\pi_j^*}{(\alpha_0' - \alpha_j')(-\pi_j^* + \omega_j^* + \zeta_j^+)}.$$

The above can then be written as

$$\sum_{j=1}^{K}\left[-\frac{1}{\tau_j' f_j'(g^{bbf})}\left(-\frac{1}{N}e_j\right) - \frac{1}{\tau_j f_j(g^{bbf})}\left(\frac{1}{N}e_j\right)\right]$$

Now, define

$$\upsilon_j' = -\frac{1}{\tau_j' f_j'(g^{bbf})} \quad \text{and} \quad \upsilon_j = -\frac{1}{\tau_j f_j(g^{bbf})}.$$

The above simplifies to

$$\sum_{j=1}^{K}\left[-\upsilon_j'\left(\frac{1}{N}\vec{e}_j\right) + \upsilon_j\left(\frac{1}{N}e_j\right)\right]$$

Since row $j$ of $A'$ is defined as $\frac{1}{N}e_j$, the zero-gradient condition is satisfied.

To see that the $\tau$'s and $\upsilon$'s are non-negative and that $g^{bbf}$ is primal feasible (lines 3 and 4 of the claimed KKT conditions), one can use the argument provided in Theorem D.10. In that same way, the modified complementary slackness conditions are also satisfied. □

**Corollary D.13.** *$\sigma, \sigma'$ as defined in Theorem D.10 and $\upsilon, \upsilon'$ from Theorem D.12 are dual feasible for BF with $P_{bar}$ (Equation 8).*

*Proof.* The zero-gradient condition in the statement of Theorem D.12 states that the Lagrangian, $L(z, \sigma, \sigma', \upsilon, \upsilon', \xi)$, has to have 0 gradient at the primal/dual optimal variables. Since we have both the primal and dual variables that make the gradient of the Lagrangian zero, it follows that $g^{bbf}$ is the minimizer of the Lagrangian $L(\cdot, \sigma, \sigma', \upsilon, \upsilon', \xi)$ for the dual variables from Theorem D.12. This implies that those $\sigma, \sigma', \upsilon, \upsilon', \xi$ are dual feasible. □

### D.5.2 $g^{bbf}$ is a BF prediction

**Corollary D.14.** *$g^{bbf}$ can be written in the BF form.*

*Proof.* This follows from the zero gradient condition in Theorem D.10 being satisfied as a consequence of Theorem D.12. I.e. if one solves for $z$ as is done in the proof of Theorem 2 in [An and Dasgupta, 2024]. We have seen that this condition for BF is how one derives the BF prediction format. Because the same condition is met here, $g^{bbf}$ has the same form. □

### D.5.3 Relating the Entropy of $g^{bbf}$ with $g^{bar}$

Like in Lemma D.8, we can bound the value of the objective ($z^\top \log z$) when evaluated at $g^{bbf}$.

**Corollary D.15.** *Let $g^{bar}$ be the optimal solution to the BF program in Equation 8.*

$$-g^{bar\top}\log g^{bar} - \sum_{w=1}^{W}\left[\frac{1}{t_w} + \frac{1}{t_w'}\right] - \sum_{j=1}^{K}\left[\frac{1}{\tau_j} + \frac{1}{\tau_j'}\right] \leq -g^{bbf\top}\log g^{bbf} \leq -g^{bar\top}\log g^{bar}$$

*for $t_w, t_w'$ defined in Theorem D.10 and $\tau_j, \tau_j'$ as defined in Theorem D.12.*

*Proof.* To see the second inequality, recall that $g^{bbf}$ is primal feasible for the BF program in Equation 8 while $g^{bar}$ is the optimal solution to that program. I.e. primal feasibility implies $g^{bbf\top}\log g^{bbf} \geq g^{bar\top}\log g^{bar}$.

For the first inequality, we would like to analyze the difference between our primal feasible solution's objective $g^{bbf\top}\log g^{bbf}$ versus the optimal minimum $g^{bar\top}\log g^{bar}$. By weak duality, $g^{bar\top}\log g^{bar}$ is larger than any dual function value. Continuing from the discussion in Corollary D.13, the (BF)

dual function is equal to $L(g^{bbf}, \sigma, \sigma', \upsilon, \upsilon', \xi)$ where the dual variables are defined in Theorem D.10 and D.12. Namely,

$$g(\sigma, \sigma', \xi) = g^{bbf\top} \log g^{bbf} + \sigma^\top (Ag^{bbf} - b^* - \epsilon^+) + \sigma'^\top (-Ag^{bbf} + b^* - \epsilon^-)$$
$$+ \upsilon^\top (A'g^{bbf} - \omega^* - \zeta^+) + \upsilon'^\top (-A'g^{bbf} + \omega^* - \zeta^-) + \xi^\top (Dg^{bbf} - \vec{1}_n).$$

By primal feasibility of $g^{bbf}$, the last term is 0. By the modified complementary slackness conditions, the above is equal to

$$g^{bbf\top} \log g^{bbf} - \sum_{w=1}^{W} \left[ \frac{1}{t_w} + \frac{1}{t'_w} \right] - \sum_{j=1}^{K} \left[ \frac{1}{\tau_j} + \frac{1}{\tau'_j} \right].$$

Putting this together with weak duality, we have

$$g^{bar\top} \log g^{bar} \geq g^{bbf\top} \log g^{bbf} - \sum_{w=1}^{W} \left[ \frac{1}{t_w} + \frac{1}{t'_w} \right] - \sum_{j=1}^{K} \left[ \frac{1}{\tau_j} + \frac{1}{\tau'_j} \right].$$

Rearranging gives us our claim. □

## D.6 Error Bound for $g^{bar}$

The last piece we need before being able to prove Theorem 5.3 is a bound on $-g^{bar\top} \log g^{bar}$. We now will show $\mathcal{B}$ such that

$$-g^{bar\top} \log g^{bar} \leq -\eta^\top \log g^* + \mathcal{B}.$$

Since $g^{bar}$ is the solution of a bona fide BF problem, we can use the analysis of [An and Dasgupta, 2024] to get a bound in the form we'd like.

**Theorem D.16.** *For $g^{bar}$ the optimal solution of Equation 8 and some $g^{ref} \in \mathcal{G}$,*

$$-\eta^\top \log g^{bar} \leq -\eta^\top \log g^{ref} + |\widehat{b} - b^*|^\top |\theta^{ref}|$$

*where $\widehat{b} := Ag^{bbf}$.*

*Proof.* This proof heavily relies on the proof of Theorem 7 in [An and Dasgupta, 2024]. It suffices to find the dual of the BF program in Equation 8, then do sensitivity analysis where the reference problem has constraints $Az = b^*$. One is able to get the following bound where $\theta^{ref} = \sigma' - \sigma$ and $\sigma, \sigma' \geq 0$.

$$-\eta^\top \log g^{bar} \leq -\eta^\top \log g^{ref} + \epsilon^{+\top} \sigma + \epsilon^{-\top} \sigma'$$

Here, the $\epsilon^\pm$ encapsulates both $\epsilon^\pm$ as defined above as well as $\zeta^\pm$ (i.e. since we concatenate the LF accuracies and class frequencies). Then, observe that $|Ag^{bbf} - b^*| + \delta = \max\{\epsilon^+, \epsilon^-\}$ where the absolute value and maximums are done elementwise. Therefore, the above bound can become

$$-\eta^\top \log g^{bar} \leq -\eta^\top \log g^{ref} + |\widehat{b} - b^*|^\top |\theta^{ref}|.$$

The $\delta$ term disappears because it can be arbitrarily small. □

In particular, we will choose $g^* = g^{ref}$.

**Corollary D.17.**

$$-\eta^\top \log g^{bar} \leq -\eta^\top \log g^* + |\widehat{b} - b^*|^\top |\theta^*|$$

*Therefore,*

$$\mathcal{B} = |\widehat{b} - b^*|^\top |\theta^*|.$$

## D.7 Formal Proof of BBF Error Bound

Now, we have all the pieces to prove Theorem 5.3. We'll explicitly state all the assumptions for completeness. This follows the proof sketch for Theorem D.1 provided earlier, but we fill in the missing details.

**Theorem D.18** (Theorem 5.3). *Fix all LF predictions and $\eta$. Also, fix $\vec{\alpha} \in \mathbb{R}^K_{\geq 1}$ and $\vec{\rho} \in \mathbb{R}^{2W}_{\geq 1}$. Say that $g^{bbf}$ is the solution to the BBF objective in Equation 7. The error of $g^{bbf}$ is such that*

$$d(g^*, g^{bbf}) \leq (|\theta^*| + |\theta^{bbf}|)^\top |b^* - Ag^{bbf}|$$

*where the absolute value is elementwise.*

*Proof.* Lemma 5 of [An and Dasgupta, 2024] shows that

$$d(\eta, g^{bbf}) = d(\eta, g^*) + d(g^*, g^{bbf}).$$

So, if we have a bound that looks like

$$d(\eta, g^{bbf}) \leq d(\eta, g^*) + \mathcal{A} + \mathcal{B},$$

then it's clear that $d(g^*, g^{bbf}) \leq \mathcal{A} + \mathcal{B}$. This is equivalent to showing a bound of the form

$$-\eta^\top \log g^{bbf} \leq -\eta^\top \log g^* + \mathcal{A} + \mathcal{B},$$

which we'll now do.

From Lemma D.5, we know that

$$-\eta^\top \log g^{bbf} \leq -g^{bbf\top} \log g^{bbf} + |\theta^{bbf}|^\top |b^* - Ag^{bbf}|.$$

Now, Corollary D.14 shows that the first term of the RHS above is upper bounded by

$$-g^{bar\top} \log g^{bar}.$$

Penultimately, Corollary D.17 shows that

$$-g^{bar\top} \log g^{bar} \leq -\eta^\top \log g^* + |\theta^*|^\top |Ag^{bbf} - b^*|.$$

Putting everything together, we have

$$-\eta^\top \log g^{bbf} \leq -\eta^\top \log g^* + (|\theta^*| + |\theta^{bbf}|)^\top |b^* - Ag^{bbf}|.$$

$\square$

## D.8 Consistency of BBF

An and Dasgupta [2024] showed that it is possible (via Theorem 6) to compute $g^*$, the prediction in $\mathcal{G}$ that minimizes $d(\eta, \cdot)$. In particular, by Theorem 6 from the above paper, $g^*$ is the BF prediction when one solves BF with polytope $P^* = \{z \in \Delta^N_K : Az = b^*\}$. Recall that for BBF to be consistent, there must be a way for its prediction to tend to $g^*$. Indeed, we'll show that under the correct conditions, BBF can approximate BF with $P^*$ arbitrarily well. This means that $g^{bbf} \to g^*$, which is what it means to be consistent.

We'll prove consistency as follows. First, we'll show that as the BBF priors concentrate in a certain way (or rather the BBF hyperparameters tend to infinity in a certain way), we'll have $Ag^{bbf} \to b^*$. (In particular, we need the LF accuracy (resp. class frequency) priors to concentrate around the empirical LF accuracies (resp. class frequency).) This fact will play an important role as it ensures all the following things happen. Then, one recalls or observes that $g^{bbf}$ can be interpreted as being an approximation to $g^{bar}$, the optimal solution of BF with $P_{bar}$. It'll then be shown that as $Ag^{bbf} \to b^*$, $g^{bbf} \to g^{bar}$. At the same time, by the way $P_{bar}$ is constructed, $g^{bar} \to g^*$, which means that $g^{bbf} \to g^*$.

Since we have abused notation in the previous sections, let $\alpha''_j$ be $\alpha_j - 1$ where $\alpha_j$ is the $j^{th}$ hyperparameter to the Dirichlet prior in BBF's generative process. Like above, we'll have $\alpha''_0 = \sum_{j=1}^K \alpha''_j$. (Recall that we had used $\alpha'_w, \beta'_w$ to denote $\rho_{w1} - 1, \rho_{w2} - 1$ respectively.)

**Lemma D.19.** *Suppose for each $w \in [W]$ and $j \in [K]$ that $\alpha'_w, \beta'_w, \alpha''_j \to \infty$. Moreover suppose that*

$$|\alpha'_w - b^*_w(\alpha'_w + \beta'_w)| \to 0 \quad \text{and} \quad \left|\alpha''_j - \omega^*_j\left(\sum_{j'=1}^{K} \alpha''_{j'}\right)\right| \to 0$$

$$\alpha'_w \to \frac{\beta'_w b^*_w}{1 - b^*_w} \quad \text{and} \quad \alpha''_j \to \frac{1}{1 - \omega^*_j}\sum_{\substack{j'=1 \\ j' \neq j}}^{K} \alpha''_{j'}$$

*for each $w$ and $j$. Then, $Ag^{bbf} \to b^*$.*

*Proof.* To simplify the argument, we'll add the Beta and Dirichlet normalizing factors into the BBF objective. This does not change the argument in a substantial way because of two factors. First, we are concerned with the mode of Beta and Dirichlet distributions, adding or removing the normalizing factor does not change the mode. Second, the proof depends on the fact that the non-entropy terms tend to infinity (whereas the entropy term remains bounded). Removing the normalization terms does not prevent the other terms from going to infinity.

So, abuse notation and say $B(\alpha, \beta)$ is the Beta function while $B(\vec{\alpha})$ is the multivariate Beta function. The modified BBF objective that we are considering is

$$\min_{z \in \Delta^N_K} \left[ z^\top \log z - \sum_{w=1}^{W} [\alpha'_w \log(\gamma_w(z)) + \beta'_w \log(1 - \gamma_w(z)) - \log B(\rho_{w1}, \rho_{w2})] \right.$$

$$\left. - \sum_{j=1}^{K} \alpha'_j \log(\pi_j(z)) + \log B(\vec{\alpha}) \right]. \quad (15)$$

Before getting started, recall that the mode of the Beta distribution is $\alpha'_w/(\alpha'_w + \beta'_w)$. Similarly, the $j^{th}$ element of the mode of a Dirichlet distribution is $\alpha''_j/\alpha''_0$. Moreover, as $\alpha'_w, \beta'_w, \alpha''_j$ increase, the variance decreases so that the density (and therefore the log density) increases unboundedly. So, suppose that $\alpha'_w, \beta'_w, \alpha''_j$ are very large (for $w \in [W], j \in [K]$). Since the BBF objective is minimization, this means that the negative log-pdfs of the Beta (resp. Dirichlet) distributions are small when the value of $\gamma_w(z)$ (resp. $\pi_j(z)$) is close to the mode. Moreover, how small these values can get is unbounded below.

Therefore, the optimal $z$ cannot just minimize the entropy term. To minimize the BBF objective, the other terms must also be made small. Moreover, minimizing these log-pdfs are the priority since they're larger in absolute magnitude compared to the entropy term. By assumption, (and with a little rearranging), one can see that the mode for the $w^{th}$ Beta function ($w^{th}$ summand) tends to $b^*_w$. Similarly, the $j^{th}$ mode of the Dirichlet mode tends to $\omega^*_j$. So, to make all the non-entropy terms small, $z$ must be chosen so that $Az$ is close to $b^*$ (which one recalls is an abuse of notation and is a concatenation of the LF accuracies $b^*_w$ and class frequencies $\omega^*_j$). Since there is a set of $z$'s such that $Az$ is 'close' to $b^*$, one can choose a $z$ from that set to minimize the entropy term. Now, as $\alpha'_w, \beta'_w, \alpha''_j$ increase, the pdfs concentrate, meaning $Az \to b^*$ in order for the objective to be minimized. $\square$

Now, we show that $g^{bbf} \to g^{bar}$.

**Lemma D.20.** *Take the same assumptions as Lemma D.19. Then, as $Ag^{bbf} \to b^*$, $g^{bbf} \to g^{bar}$.*

*Proof.* Recall from our discussion of barrier method type analysis in Subsection D.4 that the complementary slackness condition is what controls how close we are approximating the 'original problem'. That is, a convex program with a log-barrier approximation to a feasible region ($f_1(x) \leq 0$ in that example) produces an approximate solution to the original problem. Reprinting, the following is the 'original problem':

$$p^* = \min_{\substack{f_1(x) \leq 0 \\ Ax = b}} f_0(x)$$

with solution $x^*$. The problem with the log-barrier approximation to $f_1(x) \leq 0$ is

$$p(t) = \min_{Ax=b} f_0(x) - \frac{1}{t}\log(-f_1(x))$$

with solution $x^*(t)$. The result was that as $t \to \infty$, $x^*(t) \to x^*$ as the latter problem approximated the original problem arbitrarily well.

The same idea applies here for our case. To be explicit, what happens is that the problems being solved eventually match, which is why $x^*(t) \to x^*$. In our case, the KKT for the BBF problem will match the KKT conditions for BF with $P_{bar}$. Upon inspection of Theorem D.12, we must show that $t_w, t'_w, \tau_j, \tau'_j$ each tend to infinity as $Ag^{bbf} \to b^*$ for the BBF problem and BF with $P_{bar}$ to match (and to prove our claim).

We start with $t_w$ and $t'_w$. Reprinted, they are

$$t'_w = \frac{\gamma^*_w}{\alpha'_w(\gamma^*_w - b^*_w + \epsilon^-_w)}, \quad t_w = \frac{1 - \gamma^*_w}{\beta'_w(-\gamma^*_w + b^*_w + \epsilon^+_w)}.$$

For each of these terms, we are interested in the case where $\epsilon^-_w = \epsilon^+_w = \delta$. In the other case, the denominator has $\alpha'_w\delta$ or $\beta'_w\delta$, which can be arbitrarily small as $\delta$ is arbitrary. Recall from Lemma D.19 that $\gamma^*_w = y^\top_w g^{bbf}/N(\vec{y}_w) \to b^*_w$. In particular, we can write $\gamma^*_w$ in terms of $\alpha'_w$ and $\beta'_w$. Letting $\xi$ be a term to account for the deviation (as $[Ag^{bbf}]_w$ doesn't equal $b^*_w$),

$$\gamma^*_w = \frac{\alpha'_w + \xi}{\alpha'_w + \beta'_w} \to b^*_w.$$

Now, by assumption, $\alpha'_w \to b^*_w(\alpha'_w + \beta'_w)$. This means that both $\alpha'_w$ and $\alpha'_w + \xi$ tend to $b^*_w(\alpha'_w + \beta'_w)$ or $\xi \to 0$ as $\alpha'_w, \beta'_w \to \infty$. To reduce clutter, we drop the reference to $w$, noting that all quantities we're dealing with are scalars. Replacing $\gamma^*_w$ with the fraction involving $\xi$, we have

$$t' = \frac{\frac{\alpha'+\xi}{\alpha'}}{(\alpha' + \beta')(\frac{\alpha'+\xi}{\alpha'+\beta'} + b^* + \delta)} = \frac{1 + \frac{\xi}{\alpha'}}{\alpha' + \xi - b^*(\alpha' + \beta') + \delta}.$$

Recall that $\delta$ was arbitrary, meaning we can absorb anything in front of it. For the numerator, we can clearly see that it tends to 1 as $\alpha' \to \infty$. For the denominator, since $\alpha' + \xi$ tends to $b^*(\alpha' + \beta')$ and $\delta$ can be arbitrarily small, the denominator tends to 0 and $t'$ tends to $\infty$. Similarly for $t_w$, we use $1 - \gamma^*_w = 1 - (\alpha'_w + \xi)/(\alpha'_w + \beta'_w)$ to get

$$t_w = \frac{\frac{\beta'-\xi}{\beta'}}{-\alpha' - \xi + b^*(\alpha' + \beta') + \delta}$$

and the same argument from above works, this time using $\xi \to 0$ and $\alpha'_w \to b^*(\alpha'_w + \beta')_w$.

Now, we move on to $\tau_j$ and $\tau'_j$, which proceeds in a very similar way as the above. Lemma D.19 shows that $\pi^*_j = e^\top_j g^{bbf}/N \to \omega^*_j$. Like before, we write

$$\pi^*_j = \frac{\alpha'_j + \xi}{\alpha'_0}.$$

Since $\pi^*_j \to \omega^*_j$, and $\alpha'_j \to \omega^*_j\alpha'_0$, we have $\xi \to 0$. As above, we're interested in the case when the denominator is not just $\delta$. Written out,

$$\tau'_j = \frac{\pi^*_j}{\alpha'_0(\pi^*_j - \omega^*_j + \delta)} = \frac{\pi^*_j}{\alpha'_0(\frac{\alpha'_j+\xi}{\alpha'_0} - \omega^*_j + \delta)} = \frac{\pi^*_j}{(\alpha'_j + \xi - \alpha'_0\omega^*_j + \delta)}$$

The numerator tends to a constant while the first three terms in the denominator tend to 0 while $\delta$ can be made arbitrarily small. Hence $\tau'_j \to \infty$.

To see that $\tau_j$ also tends to infinity, observe that $\alpha'_0 \geq \alpha'_0 - \alpha'_j$, so that we can apply the above argument with very minor changes. $\square$

**Theorem D.21** (Theorem 5.2). *Under the conditions of Lemma D.19, BBF is consistent. I.e. $g^{bbf} \to g^*$. This happens because $\theta^{bbf} \to \theta^*$.*

Table 3: Comparison of our proposed method with baseline methods with respect to accuracy. Bolded entries are indistinguishable via two-tailed paired t-test with $p = 0.05$.

| Dataset | IMDB | YouTube | SMS | CDR | Yelp | Commercial | Tennis | TREC | SemEval | ChemProt | AG News |
|---|---|---|---|---|---|---|---|---|---|---|---|
| MV | 73.68 | 85.02 | **93.94** | 72.36 | 72.38 | 90.61 | 86.85 | 54.51 | 79.07 | 54.67 | 81.38 |
| DS | 50.31 | 83.88 | 87.18 | 45.77 | 72.97 | 93.57 | 86.54 | 47.66 | 73.53 | 52.66 | 81.79 |
| DP | 72.75 | 74.45 | 81.52 | 64.41 | 57.86 | 89.79 | 86.57 | 45.85 | 72.78 | 56.17 | 81.71 |
| FS | 74.45 | 83.61 | 91.90 | **73.62** | **74.01** | 87.31 | 85.95 | 50.10 | 12.15 | 52.37 | 81.28 |
| MeTaL | 72.43 | 70.54 | 79.67 | 60.20 | 54.47 | 89.79 | 86.57 | 43.73 | 54.58 | 55.86 | 81.84 |
| iBCC | 74.44 | 45.69 | 84.55 | 60.92 | 68.30 | 83.31 | 87.19 | 21.02 | 30.19 | 35.09 | 26.57 |
| EBCC | 74.43 | 45.69 | 84.55 | 62.98 | 69.67 | 83.31 | **87.30** | 21.02 | 30.19 | 35.09 | 27.81 |
| CLL | 72.77 | 85.19 | 93.86 | 71.24 | 72.04 | 49.99 | 49.92 | 59.34 | **84.21** | 52.37 | 80.97 |
| HyperLM | 75.03 | **90.99** | 93.86 | 72.58 | **74.42** | 88.20 | 87.20 | **60.02** | **84.21** | 52.32 | 81.38 |
| WeaSEL | 50.49 | 60.84 | 54.13 | 51.64 | 57.16 | 45.15 | 57.79 | 20.81 | 11.14 | 21.41 | 32.29 |
| Denoise | **80.26** | 81.45 | **94.02** | 73.17 | **75.04** | **94.24** | **76.87** | 49.81 | 73.53 | **57.85** | **87.61** |
| FABLE | 72.16 | 87.71 | **93.94** | 71.64 | 69.44 | 92.55 | 86.56 | 53.64 | 73.61 | 54.42 | 81.38 |
| BBF | 72.47 | 89.47 | 93.86 | 71.29 | 71.81 | 90.85 | 84.56 | 59.88 | **84.21** | 52.47 | 81.40 |
| $g^*$ | 74.50 | 93.92 | 93.90 | 76.52 | 74.77 | 93.59 | 86.64 | 63.75 | 84.21 | 61.26 | 82.24 |

*Proof.* We have just proven in Lemma D.20 that $g^{bbf} \to g^{bar}$. It suffices to show that under the assumptions of our claim, $g^{bar} \to g^*$.

To do that, it suffices to show that $P_{bar} \to P^* = \{z \in \Delta_K^N \colon Az = b^*\}$ under the assumptions of the theorem statement. This is because An and Dasgupta [2024] showed in Theorem 6 that solving BF with $P^*$ gives $g^*$. Recall that $P_{bar}$ was defined as follows:

$$P_{bar} = \{z \in \Delta_K^N \colon b^* - \epsilon^- \leq Az \leq b^* + \epsilon^+\}.$$

It would suffice to show that $\epsilon^+, \epsilon^-$ both converge to 0. We now remind the reader of the definitions of $\epsilon^+$ and $\epsilon^-$. For $i \in [W + K]$ and $\widehat{b}_i = [Ag^{bbf}]_i$,

$$\epsilon_i^- = \mathbf{1}(b_i^* > \widehat{b}_i)(b_i^* - \widehat{b}_i) + \delta \quad \text{and} \quad \epsilon_i^+ = \mathbf{1}(\widehat{b}_i > b_i^*)(\widehat{b}_i - b_i^*) + \delta.$$

By definition, at least one of these is equal to $\delta$, which is some arbitrary but small positive number. In Lemma D.19, we showed that $\widehat{b}_i \to b_i^*$. Thus, given the assumptions of the theorem, $\epsilon_i^+, \epsilon_i^-$ are each either already $\delta$, or tend to $\delta$. This means that every element in $\epsilon^+$ and $\epsilon^-$ is arbitrarily small, showing that $P_{bar} \to P^*$ as the BBF hyperparameters grow. As BBF is essentially solving the BF problem that gives $g^*$, $g^{bbf}$ will have the same weights as $g^*$, i.e. $\theta^{bbf} \to \theta^*$. □

# E   Extra Experimental results

In this section, we present extra experimental results. Specifically, the accuracies of label models for which F1 scores were reported in the main paper and the standard deviations of the F1 scores and 0-1 losses reported. Also, we present the BBF consistency experiments for all datasets used in the main paper.

## E.1   0-1 Loss and Standard Deviations

In the main paper, we followed Zhang et al. [2023] and included a mix of F1 scores and 0-1 loss depending on whether the dataset had two classes. Here, we provide the accuracies achieved by each label model on every dataset, regardless of whether there are two classes or not.

In Table 3 we see the accuracies of our proposed method and the other baseline methods. Against FABLE, our proposed method has similar or better performance on most datasets. On Commercial, Tennis, and ChemProt, the performance of BBF was slightly worse than FABLE. With respect to HyperLM, BBF was similar or better on the majority of datasets, only being worse on IMDB, YouTube, CDR, Yelp.

Here are the standard deviations for F1 score (Table 4) and 0-1 loss (Table 5). For brevity, we only show standard deviations for methods that had randomness. We see that WeaSEL had the largest standard deviation while the standard deviation for other methods were not very large.

Table 4: Comparison of F1 score standard deviations for random baseline methods.

| Dataset | IMDB | YouTube | SMS | CDR | Yelp | Commercial | Tennis |
|---------|------|---------|------|------|------|-----------|--------|
| DP | 0.07 | 1.50 | 0.73 | 7.63 | 1.66 | 0.00 | 0.00 |
| MeTaL | 0.16 | 0.00 | 0.73 | 6.18 | 0.45 | 0.00 | 0.00 |
| EBCC | 0.00 | 0.00 | 0.00 | 0.00 | 0.03 | 0.00 | 0.02 |
| WeaSEL | 10.89 | 6.67 | 6.29 | 13.42 | 7.77 | 4.63 | 17.22 |
| Denoise | 1.13 | 0.43 | 0.38 | 0.10 | 1.33 | 1.06 | 41.74 |
| FABLE | 0.26 | 0.37 | 0.00 | 0.16 | 0.16 | 0.23 | 0.06 |
| CLL | 0.00 | 0.00 | 0.00 | 0.07 | 0.00 | 0.15 | 4.02 |

Table 5: Comparison of accuracy standard deviations for random baseline methods.

| Dataset | IMDB | YouTube | SMS | CDR | Yelp | Commercial | Tennis | TREC | SemEval | ChemProt | AG News |
|---------|------|---------|------|------|------|-----------|--------|------|---------|----------|---------|
| DP | 0.16 | 2.63 | 0.55 | 1.92 | 3.90 | 0.00 | 0.00 | 0.80 | 0.25 | 0.46 | 0.25 |
| MeTaL | 0.16 | 0.00 | 0.47 | 1.23 | 1.01 | 0.00 | 0.00 | 1.96 | 3.45 | 0.51 | 0.26 |
| EBCC | 0.00 | 0.00 | 0.00 | 0.00 | 0.04 | 0.00 | 0.06 | 0.00 | 0.00 | 0.00 | 0.00 |
| WeaSEL | 4.68 | 7.63 | 12.11 | 5.73 | 3.68 | 8.93 | 5.84 | 3.40 | 2.16 | 4.91 | 2.69 |
| Denoise | 1.24 | 0.34 | 0.11 | 0.11 | 2.27 | 0.51 | 13.99 | 0.08 | 0.00 | 0.20 | 0.47 |
| Fable | 0.28 | 0.33 | 0.00 | 0.06 | 0.21 | 0.11 | 0.03 | 0.47 | 0.24 | 0.68 | 0.04 |
| CLL | 0.00 | 0.00 | 0.00 | 0.05 | 0.01 | 0.11 | 5.07 | 0.01 | 0.00 | 0.00 | 0.03 |

## E.2   Experimental Results for BBF's Consistency

In this section, we present experimental results showing the consistency of BBF on real datasets. The eleven datasets used in the main paper are considered again in the same setting – we allow BBF to label all datapoints (by combining the provided train/validation/test splits). Recall from our previous section that for $g^{bbf} \to g^*$, we need the prior distributions for LF accuracies and class frequencies to concentrate around their empirical values. This is done by using the labels of the datapoints to compute $b_w^*$ and $\omega_j^*$. The hyperparameters for the Beta distributions are set with scale factor $s$ as

$$\rho_{w1} = sb_w^* + 1, \quad \rho_{w2} = s(1 - b_w^*) + 1 \quad \text{and} \quad \alpha_j = s\omega_j^* + 1.$$

One is added so the conditions of Lemma D.19 are satisfied. We vary the scale by taking values in $\{10^1, \ldots, 10^5\}$ for 5 trials total for each dataset. To check for consistency, we measure the average KL divergence between $g^{bbf}$ and $\eta$. Namely, $\frac{1}{N}d(\eta, g^{bbf})$. The provided graphs have a log-log scale and show that the rate of convergence is linear.

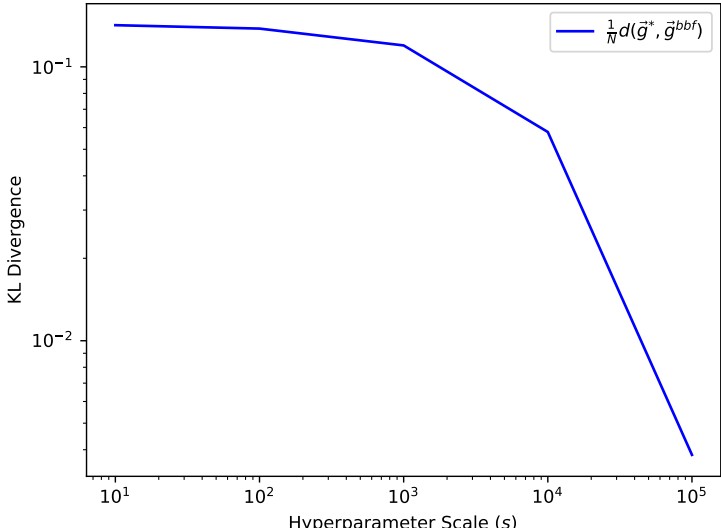

Figure 2: Convergence of $g^{bbf}$ to $g^*$ on the IMDB dataset with respect to scale factor $s$.

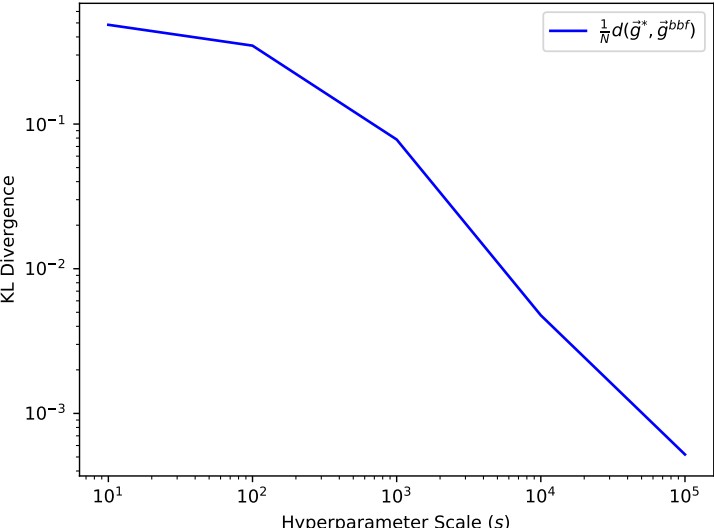

Figure 3: Convergence of $g^{bbf}$ to $g^*$ on the Youtube dataset with respect to scale factor $s$.

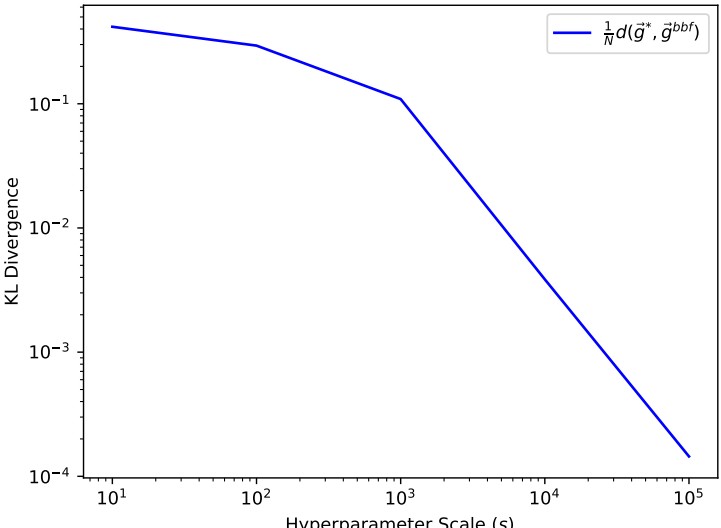

Figure 4: Convergence of $g^{bbf}$ to $g^*$ on the SMS dataset with respect to scale factor $s$.

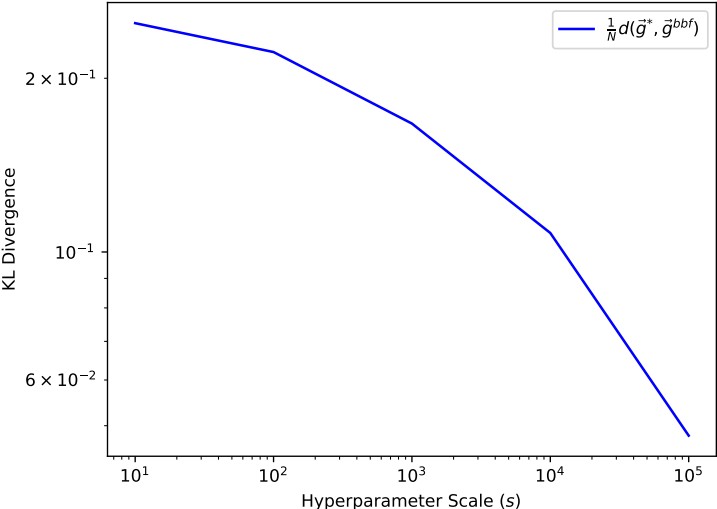

Figure 5: Convergence of $g^{bbf}$ to $g^*$ on the CDR dataset with respect to scale factor $s$.

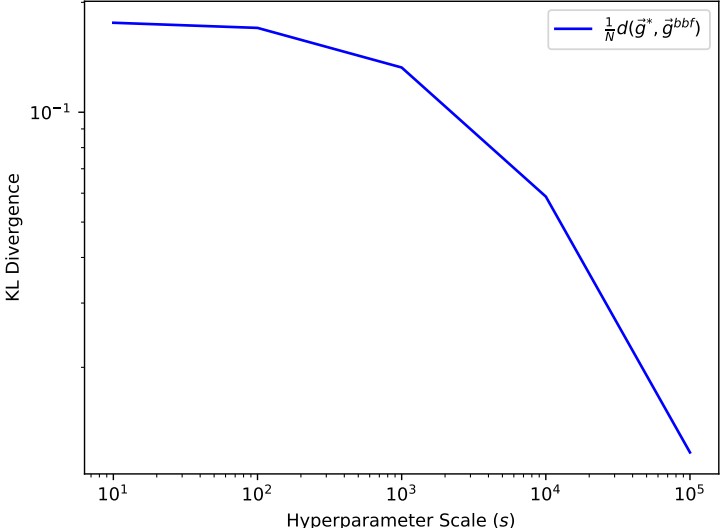

Figure 6: Convergence of $g^{bbf}$ to $g^*$ on the Yelp dataset with respect to scale factor $s$.

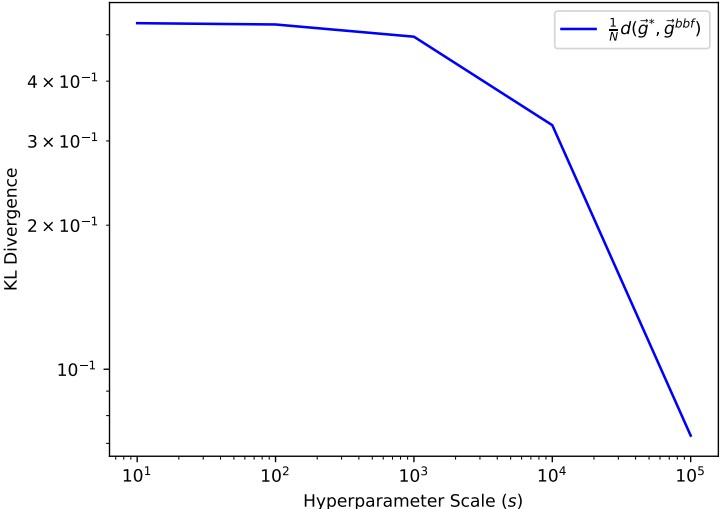

Figure 7: Convergence of $g^{bbf}$ to $g^*$ on the Commercial dataset with respect to scale factor $s$.

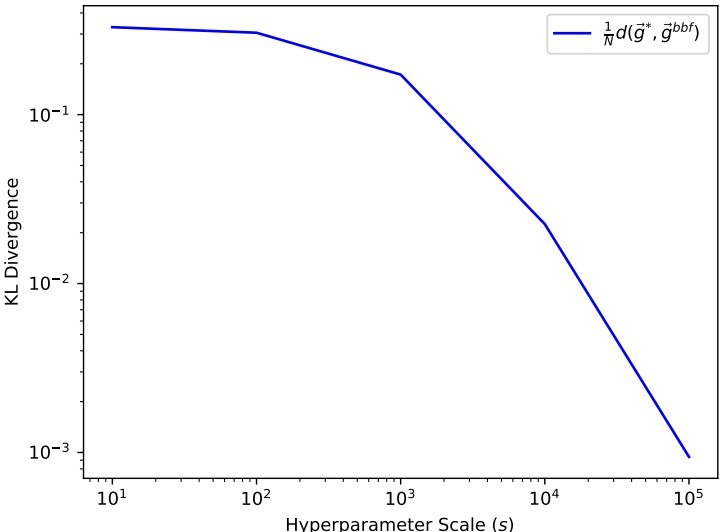

Figure 8: Convergence of $g^{bbf}$ to $g^*$ on the Tennis dataset with respect to scale factor $s$.

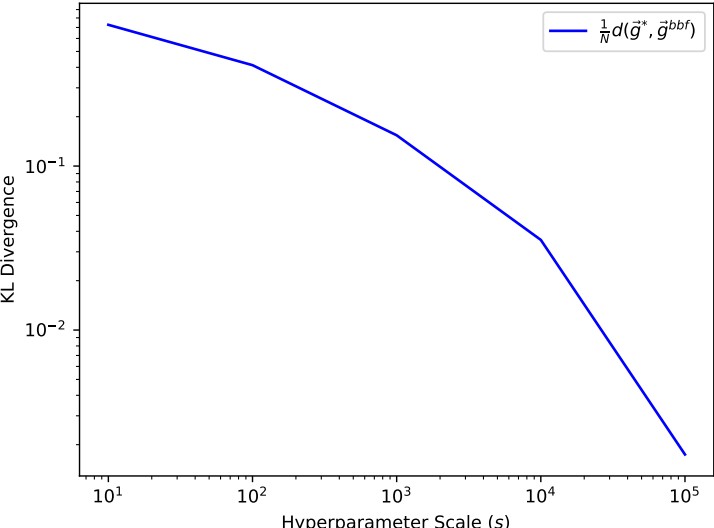

Figure 9: Convergence of $g^{bbf}$ to $g^*$ on the TREC dataset with respect to scale factor $s$.

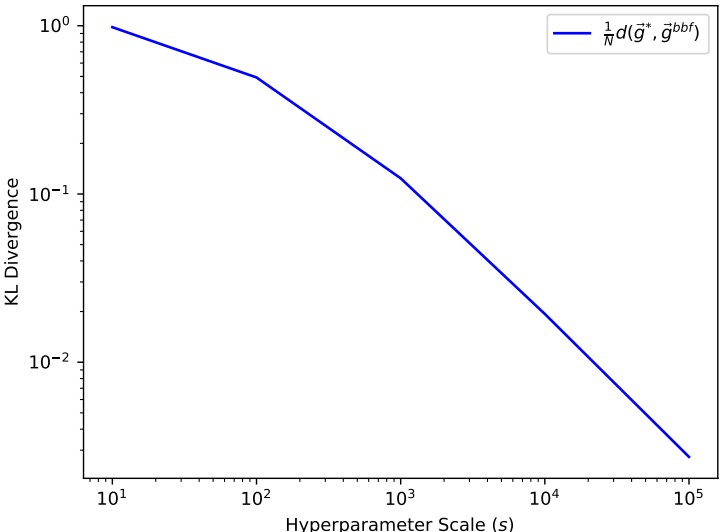

Figure 10: Convergence of $g^{bbf}$ to $g^*$ on the SemEval dataset with respect to scale factor $s$.

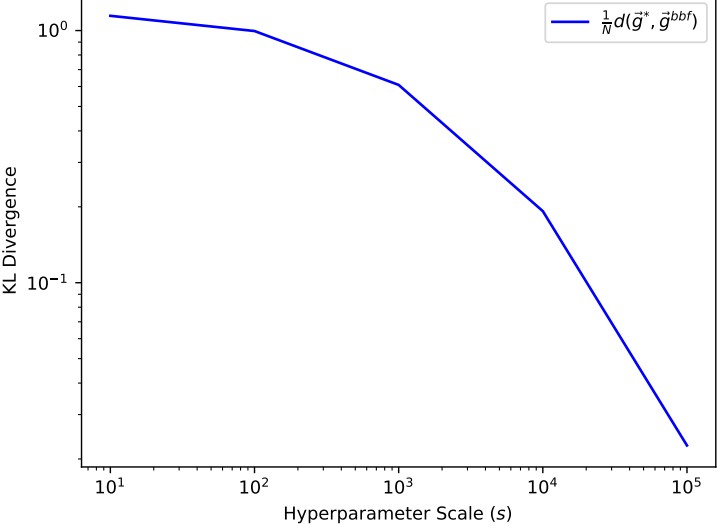

Figure 11: Convergence of $g^{bbf}$ to $g^*$ on the ChemProt dataset with respect to scale factor $s$.

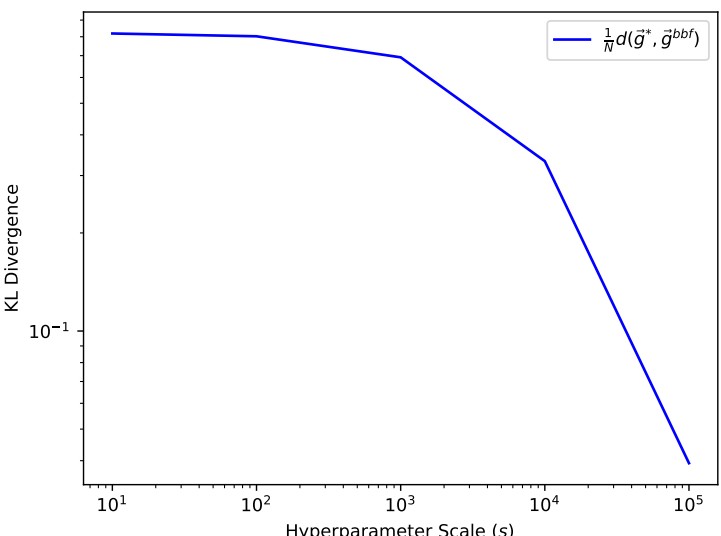

Figure 12: Convergence of $g^{bbf}$ to $g^*$ on the AG News dataset with respect to scale factor $s$.

