# OpenReview forum: "Statistical Analysis of an Adversarial Bayesian Weak Supervision Method"
_NeurIPS.cc/2025/Conference — NeurIPS 2025 poster_

### Official Review · Reviewer_BbRK · 2025-06-17

**Clarity:** 2
**Significance:** 3
**Originality:** 3
**Rating:** 5
**Confidence:** 3

**Summary:**

The paper suggests BBF, a label model that aggregates labeling functions predictions for weak supervision.
The paper investigates the model theoretically and empirically.
In empirical evaluation the model outperforms other Bayesian label models.

**Questions:**

The assumptions on the labeling functions themselves are not clearly given.
Should they all be weak classifiers?
Schapire, Robert E. "The strength of weak learnability." Machine learning 5 (1990): 197-227.
Should they be independent given the concept? Any other constraint preventing using the same labeling function ten times?


Assumptions never exactly match reality.
It will be interesting to add lower bounds of the models on the datasets and their comparison to actual performance in Table 2.

The EM algorithm influenced the entire area, via Dawid & Skene and more.
It will be nice to add a comparison to it.
Especially, is the process a single step or iterative?
Dempster, Arthur P., Nan M. Laird, and Donald B. Rubin. "Maximum likelihood from incomplete data via the EM algorithm." Journal of the royal statistical society: series B (methodological) 39.1 (1977): 1-22.
Dawid, Alexander Philip, and Allan M. Skene. "Maximum likelihood estimation of observer error‐rates using the EM algorithm." Journal of the Royal Statistical Society: Series C (Applied Statistics) 28.1 (1979): 20-28.

The learning framework is not clear.
You report F1 which requires labels, yet possibly after the modeling, which is OK.
However, you mention using accuracy in the modeling yet write “No labeled data is provided to the label models.”
If you need labels in the modeling, this is a semi-supervised framework so please specify how many labels are needed.
If you use a different method (e.g., disagreement minimization Amit, Idan, Eyal Firstenberg, and Yinnon Meshi. "Framework for semi-supervised learning when no labeled data is given." U.S. Patent No. 11,468,358. 11 Oct. 2022.) then the term accuracy is confusing and method should be explained more.
If it is simply accuracy with respect to a proxy, it should be written when accuracy is used.

HyperLM and Denoise share the top results.
It will be beneficial to try and explain the intuition of these results.
It is especially interesting since it seems that when one of the models is not the top performer, its performance tends to be very bad.
It hints that the datasets differ in some properties with a large impact on performance.

**Ethical Concerns:**

["NO or VERY MINOR ethics concerns only"]

**Final Justification:**

Clarity was the main problem, now being resolved.
Hence, I'm happy to change my recommendation to accept.

**Limitations:**

See weaknesses

**Paper Formatting Concerns:**

Why do you report F1 and not the more natural accuracy or few metrics?

“Labeling functions” seems to be a more common name than “label functions”

“better than random” - better to use the term weak classifier

“class frequency distribution” - formula seems like just class probability - did I  miss something?

“any labeling of the N datapoints that attains accuracy b on X is equally likely to be selected as h’s predictions” - and labeling with higher accuracy are more likely to be selected than those with lower accuracy? Please clarify.

“we’ll favor labelings with large entropy” - please explain why.

Majority Vote is a simple model and a good benchmark to other models.
I suggest adding to Table 2 a line describing the difference between the best model and MV.
Also, I would add that a supervised model with access to the labels in the training and the difference between its performance and the top model.

“Method” seems to be in the dataset location in Table 2.

**Quality:**

3

**Strengths And Weaknesses:**

The authors do a mathematical analysis of their model.
On the other hand they also do intensive empirical evaluation.

The benchmark uses many models and datasets, and the method shows improvement over Bayesian models.

Assumptions, setting,  and therefore the proof are not presented clearly.
I tried to point out a few unclear points.

Also, there might be certain assumptions and constraints of the models used in the benchmark, hence their straight forward usage might be under performing.
For example, “On Tennis, the three LFs that each predicted on 99% of the datapoints closely matched each other” hints that the labeling functions are not conditionally independent given the concept. Hence, performance might be better on a subset of the functions.

A general comment, not part of the judgment:
“then regardless of what h(w) predicts, its accuracy can only be 1/K.” - Isn’t it since some labels can be positive for a sample yet the LF can hit only a single label?
If so, adjusting them might help improve performance and bound.

I like the paper and I think that its weaknesses can be taken care of during the rebuttal.
If so, I’ll be happy to accept.

---

> ### Author Rebuttal · Authors · 2025-07-29
>
> Thank you for your time and effort in reading our paper and writing this review.
>
> **Paper formatting concerns**
> * For concerns not explicitly addressed here, we will take your suggestions and revise the paper.
> * We showed the F1 score to follow the FABLE paper [1].  We have recorded the accuracies in the Supplementary Material (code/results/error_tables.txt) and will add a table in the Appendix wherein only the accuracies will be reported.
> * Class frequency distribution is just class probability
> * Whether or not a labeling with higher accuracy is more likely to be selected than one of lower accuracy depends on the prior accuracy and the generated label.  Once an accuracy \\(b\\) has been generated for an LF, the only labelings with accuracy \\(b\\) can be generated.
> * We favor labelings with high entropy so that the posterior label distribution has a unique mode.  That is, the prior distributions and LF predictions define the polytope that BBF is implicitly constructing.  We want to somehow define a point in the interior and having the label distribution be proportional to its entropy is one way of achieving this in a non-ad hoc fashion.  I.e. suppose that the BBF priors were super concentrated.  Then, the implicit polytope being constructed by BBF looks like  \\(A\vec{g} = b\\) for some vector of LF accuracies \\(b\\).  If the labelings were not proportional their entropy, then the posterior label density would have equal density on every \\(\vec{g}\\) satisfying that equation.
> * We will add a line in Table 2 for difference between best method & majority vote.
> * With respect to the addition of a supervised model, we would contend that the last line in Table 2 is similar in spirit.  It is the best possible labeling that Balsubramani-Freund (BF) or Bayesian Balsubramani-Freund (BBF) can output.  Further all datapoints with labels (known to us) in the datasets are given to the label models we tested.  I.e. we merged the train/test/validation splits a la FABLE [1] and gave the models the labeling function (LF) predictions on those datapoints (and datapoint features \\(x_i\\) if the label model could use them).  Thus, we don’t think we can give an apples to apples comparison with a supervised method without doing a split on the merged dataset and rerunning all the experiments.  Something similar to what you’re suggesting was done in [2].  There, BF performed quite well against supervised methods where each method was given the same training data.
>
> **On the weaknesses you bring up**
> * There are no assumptions on the labeling functions, except that they predict on at least 1 datapoint.  Since we show that BBF's solution (i.e. the posterior labeling's mode) is the solution to a slightly deformed version of the Balsubramani-Freund (BF) model's KKT conditions, one can have as many duplicates of LFs as they would like without harming the model performance.
> * We agree it would be interesting to see lower bounds for each model’s performance.  Unfortunately, we are not aware of any.
> * We provide a row in Table 2 for Dawid-Skene (DS) where EM is run to convergence.  An and Dasgupta [3] provide a theoretical comparison between (BF) and the one-coin version of DS (each labeling function (LF) is modeled as a biased coin) where 1 E step is taken to get the DS prediction.  In particular, the two methods draw their predictions from the same set and BF’s consistency implies there is no problem (set of LF predictions and underlying label distribution \\(\\vec{\\eta}\\) where DS always outperforms BF.
> * The learning framework is as follows.  We are given datapoints \\(\\{x_1, \\ldots, x_N\\}\\) and some LFs \\(h^{(1)}, …, h^{(W)}\\) and are expected to provide labels \\(\\{z_1, \\ldots, z_N\\}\\) for those datapoints.   For BBF in specific, we only need to give it the LF predictions on \\(\\{x_1, \\ldots, x_N\\}\\) and set the hyperparameters for its Beta and Dirichlet priors.  It will then output its predictions of the datapoint labels.  To model the LF accuracies without any labeled data, we assume that every LF has made 4 correct predictions and 1 wrong one for a \\(Beta(4,1)\\) prior.  Something similar is done in EBCC [4].  There, LFs are modeled as confusion matrices.  They assume that each LF has made 4 correct predictions under every class and made every mistake once.  For the \\(K=2\\) class case, each LF is assumed to correct predict \\(+\\) 4 times (4 true positives), predict \\(-\\) when the class was \\(+\\) once (1 false negative), correctly predict \\(-\\) 4 times (4 true negatives), and incorrectly predict \\(+\\) when the class was \\(-\\) once (1 false positive).  They use a row stochastic confusion matrix with Dirichlet priors -- \\(Dirichlet(4,1)\\) for the first row and \\(Dirichlet(1,4)\\) for the second row.   In the experiments, the labels \\(\\{z_1, \\ldots, z_N\\}\\) are only used to compute the F1 scores/accuracies once every label model has outputted their predictions.  One could use labeled data to set the Beta and Dirichlet priors for BBF (and similarly for EBCC), but we do not pursue this path for the experiments in our paper.
> * To differentiate HyperLM and Denoise, we first want to emphasize that HyperLM only uses the LF predictions \\(h^{(w)}(x_i)\\) and does not use the datapoint features \\(x_i\\).  Denoise on the other hand uses both LF predictions and datapoint features.
>     + With respect to HyperLM, we checked whether its modeling assumption was met for each of the eleven datasets.  We did not find a pattern between the LFs satisfying HyperLM’s assumption and whether HyperLM performed better/worse than Denoise.  On the datasets where HyperLM decidedly performed worse than Denoise, i.e. IMDB, Commercial, ChemProt, AGNews, we found that the performance was similar to that of Majority Vote.  We hypothesize that the “center” of the polytope constructed by HyperLM might be close to the majority vote labeling.  It is unclear to us what might cause this, because there are datasets w/ a low number of LFs where HyperLM does well (Tennis), and there’s a dataset where the LF accuracies are high but HyperLM has performance similar to Majority Vote (AGNews, 82% average accuracy for LFs).  Thus, we don’t think it’s a combination of whether the HyperLM modeling assumption was met/how many LFs there are/how accurate the LFs are.
>     + On the other hand,  Denoise utilizes an attention mechanism to provide the weights for a convex combination of the LF predictions.  To train the attention mechanism, they minimize the cross entropy between the current prediction and some “pseudo-clean” labeling.  When the algorithm starts, the majority vote labeling is taken as the “pseudo-clean” labeling.  As the model gets trained, the label model's old predictions gets used as the “pseudo-clean" labeling.  This similar to bootstrapped co-training [4].  One possible explanation is that Denoise gets stuck in poor local optima — which could explain its poor performance against HyperLM and even majority vote.
>
> [1]  Zhang, J., Song, L., Ratner, A., Leveraging Instance Features for Label Aggregation in Programmatic Weak Supervision, AISTATS 2023
>
> [2] Balsubramani, A., Freund, Y., Scalable Semi-Supervised Aggregation of Classifiers, NeurIPS 2015
>
> [3] An, S. and Dasgupta, S., Convergence Behavior of an Adversarial Weak Supervision Method, UAI 2024
>
> [4] Blum, A., Mitchell, T., Combining labeled and unlabeled data with co-training, COLT 1998

---

> > ### Comment · Reviewer_BbRK · 2025-08-02
> > **Clarity of your assumptions**
> >
> > I recommended the current version.The main reason was the clarity problem, which hurts the ability to understand the proof.I believe that the authors did solid work yet the clarity prevents me from being convinced in that, forcing me to do a big leap of faith.
> > Take for example the assumptions about the labeling functions.In the rebuttal you say: "Since we show that BBF's solution (i.e. the posterior labeling's mode) is the solution to a slightly deformed version of the Balsubramani-Freund (BF) model's KKT conditions, one can have as many duplicates of LFs as they would like without harming the model performance. "That is important. You should clearly state that you assume the KKT conditions and explain why not only that the functions need not be independent given the concept (eof examples), they can also be duplicated (which is a big advantage).
> > Similarly, in the rebuttal  you say that " we assume that every LF has made 4 correct predictions and 1 wrong one for a  prior."This is a strong assumption (e.g., compared to weak learnability).I suggest starting it and since the bar is so high, address results given lower performance.
> > I do believe that the problems are due to clarity and I hope that they will be fixed during this phase and the paper will be accepted.

---

> > > ### Author Response · Authors · 2025-08-02
> > >
> > > Thank you for your response.  We appreciate your comments and will revise the paper to add the things discussed to improve the overall clarity of our work.

---

> > > > ### Comment · Reviewer_BbRK · 2025-08-04
> > > >
> > > > Good.
> > > > From my point of view, clarity is the main problem.
> > > >
> > > > Please post it here, with focus on assumptions.

---

> > > > > ### Author Response · Authors · 2025-08-05
> > > > >
> > > > > We wish to qualify the claim that BBF is tolerant of arbitrarily many duplicates as what we originally claimed (in the rebuttal) is not always correct.  To be clear, our analysis does not require any assumptions on the LFs and will work regardless.  However, we believe that if enough duplicate LFs are added, BBF’s performance can be harmed.  We should have said that BBF can tolerate arbitrarily many duplicate LFs in the limiting case (under the assumptions of Theorem 5.2). We now try and explain the error.  Recall we have said BBF’s labeling satisfies a modified version of BF’s KKT conditions.  One can show that BF is never affected by duplicate LFs.  If one looks at BF’s KKT conditions, the reason it is tolerate of duplicate LFs is because complementary slackness is satisfied.  We show in the Appendix that BBF in general does not satisfy complementary slackness, meaning it is not tolerant against duplicate LFs in general.  Only under the setting of Theorem 5.2 does BBF satisfy complementary slackness.
> > > > >
> > > > > We regret this error, and see that it underscores your request for greater clarity.  Below, we have included our revisions.  We aim to achieve the following with these revisions:
> > > > >
> > > > > 1. Improve clarity of assumptions on LFs throughout the paper (including the subtleties brought up above)
> > > > > 2. Improve the clarity of the experimental setup
> > > > > 3. Add additional discussion regarding BBF/HyperLM/Denoise’s performance
> > > > >
> > > > > Since we are not able to post a PDF, we list the specific changes that were made.  Section name and line numbers are provided for every change.
> > > > >
> > > > > ### Section 3: Preliminaries:
> > > > >
> > > > > In line 105, before the sentence starting with “We’ll abbreviate”, add the sentence:
> > > > >
> > > > > “In particular, our analysis makes no assumptions about the LFs, e.g. no restrictions on their multiplicity, no requirements about their independence given certain events, etc.”
> > > > >
> > > > > ### Section 5: Statistical Analysis of Our Proposed Method:
> > > > >
> > > > > In line 235, insert the following sentences at the beginning of the line.
> > > > >
> > > > > “Note that unlike BF, BBF’s performance _is_ affected by duplicate LFs in most cases.  It is only in a special case that BBF is not affected by duplicate LFs.  However, the following results for BBF do not depend on having a set of unique LFs.”
> > > > >
> > > > > ### Section 5.2: Consistency:
> > > > >
> > > > > At the end of line 256, add the following sentences.
> > > > >
> > > > > “It is here that BBF becomes unaffected by duplicate LFs.  At a high level, this is because BBF will satisfy the KKT conditions associated with BF when it outputs \\(\\vec{g}^\*\\).”
> > > > >
> > > > > ### Section 5.3: Rates of Convergence:
> > > > >
> > > > > At the end of line 272, add the following sentences.
> > > > >
> > > > > “Moreover, one should note that duplicate LFs will increase the size of the bound as each duplicate LF adds an extra term.”
> > > > >
> > > > > ### Section 6.1: Methods:
> > > > >
> > > > > In line 281, modify the sentence starting with “This follows” so that it reads
> > > > >
> > > > >  “This follows the initialization … their Bayesian method, i.e. we assume each LF makes 4 correct predictions and 1 wrong prediction.”
> > > > >
> > > > > ### Section 6.2: Datasets:
> > > > >
> > > > > In line 302, right before the sentence starting with “No labeled data”, add the following sentence:
> > > > >
> > > > > “Each label model is given LF predictions on the datapoints (and the datapoint features themselves if used).”
> > > > >
> > > > > In line 303, before the sentence starting with “Tasks covered”, add the following sentences:
> > > > >
> > > > > “Label models are expected to be initialized and provide a labeling without using any labels.  Labels are only used to judge the quality of each label model’s predictions by way of F1 Score/0-1 loss.”
> > > > >
> > > > > ### Section 6.3 Comparison with Other Label Models:
> > > > >
> > > > > In line 319, append the following sentences:
> > > > >
> > > > > “For ChemProt specifically, we think BBF is worse than Denoise because the average LF accuracy is just under \\(0.47\\), which differs greatly from our mean prior accuracy of \\(0.8\\).  For almost every other dataset, the average accuracy of the LFs was within \\(0.11\\) of \\(0.8\\).  SemEval was the outlier with average LF accuracy of \\(0.97\\)”
> > > > >
> > > > > Before line 320, add the following paragraph:
> > > > >
> > > > > “Since BBF employs a similar strategy to HyperLM, we now quickly discuss the performance differences between HyperLM and Denoise.  On IMDB, Commercial, ChemProt, AGNews, HyperLM performs worse than Denoise, so much so that it’s close to majority vote in performance.  We hypothesize that the “center” of the polytope constructed by HyperLM might be close to the majority vote labeling.  This is because we could not find a pattern between whether HyperLM’s assumptions were met, the number and/or quality of the LFs provided, and whether HyperLM underperformed against Denoise.  In the cases where Denoise underperformed against HyperLM, we believe this is because Denoise might get stuck in poor local optima as its strategy for learning is similar to that of bootstrapped co-training [Blum and Mitchell, 1998].”

---

> > > > > > ### Comment · Reviewer_BbRK · 2025-08-06
> > > > > >
> > > > > > Very good!
> > > > > >
> > > > > > I recommend stating KKT conditions.If the minimal accuracy should be at least 80%, please state it too.
> > > > > > The duplicate functions are thought experiments; nobody will do it on purpose.Hence the performance there is more illustrative than practical.However, very correlated though not duplicated functions might be used.

---

> > > > > > > ### Author Response · Authors · 2025-08-07
> > > > > > >
> > > > > > > We are glad we were able to address your concerns re clarity.
> > > > > > >
> > > > > > > Unfortunately, the modified KKT conditions for BBF would be too long to include in the main paper as it takes up almost an entire page.  They're stated on pages 23-24 in the version of the paper that has its Appendix included (see Supplementary Material).
> > > > > > >
> > > > > > > There is no requirement that the minimal accuracy be 80%.  In our latest comment to Reviewer nt5S, we include experimental results for BBF where the prior hyperparameters are initialized using the majority vote labeling.  There, the performance of BBF is quite close to what was presented in the main paper.

---

> > > > > > > > ### Comment · Reviewer_BbRK · 2025-08-07
> > > > > > > >
> > > > > > > > If it is in the appendix it is good.
> > > > > > > > However, please state your main assumption in the for paper and send to the appendix just the details.
> > > > > > > >
> > > > > > > > Please add there the function performance expectation too.

---

> > > > > > > > > ### Author Response · Authors · 2025-08-07
> > > > > > > > >
> > > > > > > > > Here are our extra revisions regarding your comments:
> > > > > > > > >
> > > > > > > > > ### Section 5.2: Consistency
> > > > > > > > >
> > > > > > > > > On line 255, delete the sentence starting with "We'll see".  Also, delete the sentence we added to this section in our previous list of revisions.  Then, add the following:
> > > > > > > > >
> > > > > > > > > "In the Appendix, we show that BBF satisfies a modified version of BF's KKT conditions (for a fixed implicit polytope).  Specifically, the complementary slackness condition is not satisfied.  However, as we make BBF's implicit polytope tend to \\(P^\*\\), the complementary slackness conditions will be satisfied, meaning BBF replicates the behavior of BF with \\(P^\*\\).  Only then is BBF unaffected by duplicate LFs."
> > > > > > > > >
> > > > > > > > > ### Section 6.1: Methods
> > > > > > > > >
> > > > > > > > > At the end of line 281, add the following sentences:
> > > > > > > > >
> > > > > > > > > "Note that there are no requirements on the LF accuracies for BBF.  Across all LFs used in the experiments, the accuracies range from \\(0.0366\\) to \\(1\\)."

---

### Official Review · Reviewer_K6Pv · 2025-06-30

**Clarity:** 4
**Significance:** 3
**Originality:** 4
**Rating:** 5
**Confidence:** 5

**Summary:**

The paper theoretically studies a new label model that is neither probabilistic nor adversarial but performant in practice by proposing BBF. A series of statistical results are presented for BBF, including log-concavity of its posterior, consistency and rates of convergence. Empirically, BBF compares favorably over other label models. Overall, the paper offers valuable theoretical insights of a new category of performant label model, and proposed a practical method, it makes a solid contribution both theoretically and empirically.

**Questions:**

see W1

**Ethical Concerns:**

["NO or VERY MINOR ethics concerns only"]

**Final Justification:**

paper looks good to me.

**Limitations:**

yes

**Quality:**

4

**Strengths And Weaknesses:**

Strong points:
1. The paper offers novel theoretical insights into HyperLM’s strong performance in real-world by proposing Bayesian Balsubramani-Freund (BBF). This make a solid contribution to the programmatic weak supervision field as this is the first theoretical analysis of a new category of label model that is performant in practice.
2. The paper shows BBF has some nice properties including log-concavity of posterior, form of exponential family solution, consistency, and rates of convergence.
3. Writing is clear and also strong empirical results are also shown.

Weak points:
1. I did not find any major weakness for this paper and one minor point is that, in experiments, since some methods involve randomness, it would also be nice to present the variance as in practical applications variance is also important.

---

> ### Author Rebuttal · Authors · 2025-07-29
>
> Thank you for your time and effort in reading our paper and writing this review.
>
> We will include a table in the Appendix of the standard deviations of the errors (as that is what was recorded).  The standard deviations are already in the Supplementary Material, but one has to go dataset by dataset, label model by label model.  We have however consolidated them into one table which we paste below.  Label models which are deterministic have their rows omitted.
>
> 	\begin{tabular}{cccccccccccc}
> 		\toprule
> 		Dataset & IMDB & YouTube & SMS & CDR & Yelp & Commercial & Tennis & Trec & SemEval & ChemProt & AGNews \\
> 		\midrule
> 		Snorkel & $0.07$ & $1.50$ & $0.73$ & $7.63$ & $1.66$ & $0.00$ & $0.00$ & $0.80$ & $0.25$ & $0.46$ & $0.25$ \\
> 		MeTaL & $0.16$ & $0.00$ & $0.73$ & $6.18$ & $0.45$ & $0.00$ & $0.00$ & $1.96$ & $3.45$ & $0.51$ & $0.26$ \\
> 		EBCC & $0.00$ & $0.00$ & $0.00$ & $0.00$ & $0.03$ & $0.00$ & $0.02$ & $0.00$ & $0.00$ & $0.00$ & $0.00$ \\
> 		WeaSEL & $10.89$ & $6.67$ & $6.29$ & $13.42$ & $7.77$ & $4.63$ & $17.22$ & $3.40$ & $2.16$ & $4.91$ & $2.69$ \\
> 		Denoise & $1.13$ & $0.43$ & $0.38$ & $0.10$ & $1.33$ & $1.06$ & $41.74$ & $0.08$ & $0.00$ & $0.20$ & $0.47$ \\
> 		FABLE & $0.26$ & $0.37$ & $0.00$ & $0.16$ & $0.16$ & $0.23$ & $0.06$ & $0.47$ & $0.24$ & $0.68$ & $0.04$ \\
> 		\bottomrule
> 	\end{tabular}

---

> > ### Comment · Reviewer_K6Pv · 2025-08-08
> >
> > thanks for the response. it looks good to me.

---

### Official Review · Reviewer_nt5S · 2025-06-30

**Clarity:** 3
**Significance:** 2
**Originality:** 3
**Rating:** 3
**Confidence:** 4

**Summary:**

This paper introduces a Bayesian label model for programmatic weak supervision. This approach (named BBF) combines the polytope‐center strategy of HyperLM with the minimax guarantees of Balsubramani–Freund in a fully unsupervised Bayesian approach.

The authors first show that BBF has a log-concave posterior distribution over labels. They also show that the learned solution of their approach is contained within an exponential family of weighted combinations of weak labeler outputs. The authors provide consistency guarantees on their approach.

**Questions:**

Q1: Can you provide guidelines for choosing the hyperparameters for each of the priors in practice?

Q2: Could the authors compare against some subset of the minimax based approaches, assuming a fixed constant accuracy for each labeler (e.g., say 0.75 for binary classification tasks)?

**Ethical Concerns:**

["NO or VERY MINOR ethics concerns only"]

**Final Justification:**

While the authors provided some clarifications in the rebuttal, I still feel that the framing of the paper around how the approach doesn't require labeled data, while alternative adversarial PWS approaches do to estimate labeler accuracies, is incorrect. The paper requires pre-specifying a prior distribution around LF accuracies (which can only be estimated with labeled data for a principled approach). The author's argument about empirical performance can similarly be made about [1], and no empirical comparison with [1] is made. As such, I choose to maintain my score as a borderline reject: there are interesting contributions from the paper, but the claims about how the method do not require labeled data while every other adversarial PWS approaches do is **too strong**, and the paper lacks comparisons to such approaches.

[1] Arachie, C. and Huang, B., Constrained labeling for weakly supervised learning.

**Limitations:**

Yes

**Quality:**

2

**Strengths And Weaknesses:**

Strengths
* The provided Bayesian approach is well motivated and the authors provide a theoretical analysis on (a) what the posterior form is and (b) the consistency of the resulting method.

Weaknesses
* The authors only show consistency when the priors of labeler accuracies concentrates around the empirical accuracy. While the authors claim this approach to be fully unsupervised (contrasting with prior minimax approaches that require labeled data to compute labeler accuracies), this consistency only arises when priors are properly specificed – which would indeed likely require labeled data without other domain knowledge.
* Lack of discussion with [1, 2], which both are relevant constrained optimization or minimax approaches (albeit not Bayesian in nature) in PWS and both provide guarantees
* No comparison with any of the minimax-based approaches above or in the related work – while their formulation does require labeled data, they can be run assuming a fixed accuracy as a hyperparameter, which is not too dissimilar from specifying the uniformative prior distributions in BBF; I believe such an experiment would greatly strengthen the support for using such a proposed Bayesian approach for PWS
* As highlighted by the authors, scalability is a concern – the number of parameters of the approach scales with the input data, which is certainly undesirable
* Empirically, the performance of the approach isn’t that strong – it’s almost always outperformed by HyperLM and does not noticeably improve upon the MV baseline

[1] Arachie, et. al. Constrained labeling for weakly supervised learning.

[2] Mazetto, et. al. Adversarial multi class learning under weak supervision with performance guarantees.

---

> ### Author Rebuttal · Authors · 2025-07-29
>
> Thank you for your time and effort in reading our paper and writing this review.
>
> **With respect to points made in the weaknesses section:**
>
> We will add some discussion to the paper with respect to the papers you bring up.
>
> Arachie and Huang [1] provide a guarantee, though it only holds when the empirical LF error rate is known, our \\(b^\*\\).  In that paper, they are silent about how the performance of their method changes as the quality of the LF accuracy estimate improves.
>
> Mazzetto [2] is going for a more classic generalization bound scenario where the points are drawn iid.  Our bound is more general insofar as that requirement is not necessary for the proof to work.  Moreover, the setup for their minimax game is more akin to the setup of Balsubramani-Freund [3] rather than HyperLM's.
>
> **With respect to your questions:**
>
> **Q1:**
> If one has labeled data at their disposal, they may use those to instantiate the Betas/Dirichlet priors. E.g. lets say a labeling function (LF) gets \\(v_w\\) out of \\(v\\) labeled points correct.  The Beta prior for \\(h^{(w)}\\)'s accuracy could be \\(\rho_{w1} = 1+v_w\\), \\(\rho_{w2} = 1+(v-v_w)\\).  If there are no labeled datapoints available, one can use our choices of hyperparameters (\\(\rho_{w1} = 4\\), \\(\rho_{w2} = 1\\)), which was also used by EBCC [4] and seems to work well.  One pitfall to avoid is to have the hyperparameters be too large in size when there is no guarantee that it is tending toward the empirical accuracies.  This can happen if one uses the majority vote (MV) labeling as the ground truth.  For example, say that \\(h^{(w)}\\) gets \\(m_w\\) points “correct” when using the MV labeling as the ground truth.  One could instantiate the Beta hyperparameters as \\(\rho_{w1} = 1+m_w\\), \\(\rho_{w2} = 1+(N-m_w)\\) which will give you a bad result.  This is consistent with the theory because if $N$ is large, then the Beta is very concentrated and the resulting BBF prediction will have accuracy close to \\(A\vec{g}^{MV}\\), \\(\vec{g}^{MV}\\) being the majority vote labeling, which might be far from \\(b^\*\\), the empirical accuracies of the LFs.  This also holds true for instantiating the Dirichlet hyperparameters for the class frequency distribution.  The strategy of using the majority vote labeling to instantiate the Dirichlet hyperparameters is done in EBCC and won’t work for our method.
>
> **Q2:** We can’t just set the accuracy to 0.75 and run these adversarial methods that need accuracy estimates.  As Arachie and Huang [1] note, this will result in an infeasible problem.  This is akin to having an underdetermined system \\(Ax=b^\*\\) and trying to propose a new RHS \\(b\\) that will ensure the system has >=1 solution.  For the methods that have been discussed it seems that Arachie and Huang's method is the only one that can tolerate choices arbitrary choices accuracies.  That is, they learn some labeling \\(\vec{g}^{CLL}\\) by minimizing \\(||A\vec{g}^{CLL}-b||_2^2\\).  For Balsubramani-Freund (in An and Dasgupta [5]) and Mazzetto et. al. [2], the feasible set cannot be empty as they choose predictions from the set of \\(x\\) that satisfy \\(Ax=b\\).   Moreover, the size of the feasible set heavily influences the performance of the method.  E.g. suppose we started with \\([0.75, 0.75]\\) as our accuracy interval for each LF and widened it until the the feasible sets for BF/Mazzetto et. al. become non-empty.  That’s akin to choosing \\(\epsilon_w\\) for each LF to get intervals of \\([0.75-\epsilon_w, 0.75 + \epsilon_w]\\).  From our own toy experiments, the size of \\(\epsilon_w\\) heavily affects the performance of BF.  E.g. for \\(K=2\\) classes, if the accuracy interval is too wide, then the resulting max entropy solution will be close to the uniform label distribution.  In that case, if the intervals are so large that accuracies of \\(0.5\\) are included, choosing the uniform label distribution will satisfy the constraints and give the maximum entropy.   How to choose the \\(\epsilon_w\\)'s without any labeled data while maintaining good performance is an interesting but, in our opinion, difficult open question.
>
> [1] Arachie, C. and Huang, B., Constrained labeling for weakly supervised learning, UAI 2021
>
> [2] Mazzetto, A. et. al., Adversarial Multiclass Learning under Weak Supervision
> with Performance Guarantees, ICML 2021
>
> [3] Balsubramani, A. and Freund, Y., Optimally Combining Classifiers Using Unlabeled Data, COLT 2015
>
> [4] Li, Y., Rubinstein, B., Cohn, T., Exploiting Worker Correlation for Label Aggregation in Crowdsourcing, ICML 2019
>
> [5] An, S. and Dasgupta, S., Convergence Behavior of an Adversarial Weak Supervision Method, UAI 2024

---

> > ### Comment · Reviewer_nt5S · 2025-08-04
> >
> > Thank you for your clarification, especially regarding Q2.
> >
> > My concerns remain, with regards to only achieving "consistency when the priors of labeler accuracies concentrate around the empirical accuracy". While empirically the values taken from other papers work well, this requirement seems to go against the overall narrative that this approach works in a fully unsupervised manner (contrasting with the other PWS approaches, such as that of Arachie, C. and Huang [1]). I also believe that a comparison with [1] would significantly strengthen the case for this paper, as well as addressing the first weakness, much more explicitly in the overall motivation.

---

> ### Author Response · Authors · 2025-08-05
>
> Thank you for your response.
>
> We believe that the requirement that the LF accuracy priors concentrate around the empirical accuracy for the consistency result is not in tension with our proposed method being fully unsupervised.
>
> We claim that the convergence rather than consistency result is the theoretical justification for BBF’s empirical performance.  We would describe consistency as the limiting behavior of the method while the rates of convergence better describe the non-asymptotic behavior.  Specifically, we believe that Theorem 5.3 can provide an explanation for the observed empirical results.  I.e. one may ask why BBF gets good experimental results despite us always initializing the LF accuracy priors as \\(Beta(4,1)\\).  Recall that Theorem 5.3 shows that if the priors are such that \\(|A\\vec{g}^{bbf} - b^*|\\) is small element wise, then the excess error incurred by BBF won’t be too large.  Thus, we would hope that \\(A\\vec{g}^{bbf}\\) is close to \\(b^\*\\) in practice so that BBF’s performance can be explained by Theorem 5.3, i.e. the BBF prediction does not incur too much excess loss.  In the experimental results, we see that BBF often does not incur a lot of excess loss. I.e. for just over half the datasets, the loss from the BBF prediction is close to \\(\\vec{g}^\*\\), the best prediction BBF can output.  Note that Theorem 5.3 is silent on whether one needs labeled data to get a good initialization of the LF accuracy priors.  Consistency is achieved only in the case where \\(A\\vec{g}^{bbf}=b^\*\\) elementwise.
>
> To bolster the above claim, we compare the consistency results of BBF and Dawid-Skene’s (DS).   Specifically, DS is an unsupervised method with a consistency result that’s quite similar to BBFs.  Thus, if it’s agreed upon that DS being unsupervised is not in tension with its consistency result, we think the same can be said for BBF.  One can easily show that one-coin DS is consistent in the following sense.  Suppose the labels and LF predictions are generated with the DS generative process.  Then, as the LF accuracies (and class frequencies) tend to their _underlying_ values, the DS prediction gotten from using one E step (converting the accuracies to LF weights \\(\\log(\\frac{b}{1-b})\\) for \\(K=2\\) classes, then taking the weighted majority vote) is the optimal prediction.  This differs from the BBF consistency result in two ways.  First, we do not require that the labels/LF predictions be generated according to a specific generative process.  Second, we want the LF accuracy estimates to tend to the _empirical_ rather than underlying quantities.  If we would like to achieve the consistency result for DS in practice (without any prior knowledge), it would involve gathering large amounts of labeled or unlabeled data to estimate the LF accuracies.  In that sense it would be impractical to achieve.  For BBF is it also impractical in most scenarios to use the consistency result as it would involve gathering large amounts of labeled data as you note.  It is our opinion that BBF and DS are sufficiently similar with respect to their consistency results so that either consistency is in tension with being unsupervised for both models, or there’s no tension for either model.
>
> We will revise the paper to more explicitly address the first weakness and compare/contrast our proposed method against other PWS approaches.

---

> > ### Author Response · Authors · 2025-08-05
> >
> > Here, we list out our revisions, focusing specifically on your suggestions.  We aim to address the following:
> > 1. Add more discussion about other PWS methods
> > 2. Add a comparison between Arachie and Huang’s method ([1] above) to ours
> > 3. Address how one can theoretically justify BBF's performance in an unsupervised setting
> >
> > Since we cannot post a pdf, we have included the changes, where the section and line number is specified for every change.
> >
> > ### Section 2: Related Work
> >
> > In line 91, before the sentence starting with “Alternate formulations”, insert the sentence:
> >
> > “Arachie and Huang [2021] estimate the LF accuracies and then try to find the minimum norm labeling that induces those accuracies from the LF predictions. "
> >
> > In line 94, append the following at the end of the line:
> >
> > “Balsubramani and Freund [2015a,b], Mazuelas et al. [2020, 2022], and An and Dasgupta [2024] all provide convergence results in terms of how well the LF accuracies are estimated, but only the latter show consistency.  Mazzetto et al. [2021] provide a generalization bound, though they require datapoints be drawn i.i.d.  In contrast, the result of Arachie and Huang [2021] result requires that one know the actual LF accuracies on the unlabeled datapoints and for there to be an increasing number of LFs to drive down the prediction error.  In this paper, we do not consider the setting of adding more LFs and do not require the actual LF accuracies to be known for our analysis.”
> >
> > ### Section 5.3: Rates of Convergence
> >
> > On line 272, append the following:
> >
> > “However, if an unsupervised instantiation of \\(\\vec{\\alpha}, \\vec{\\rho}\\) induces \\(\\vec{g}^{bbf}\\) such that \\(A\\vec{g}^{bbf}\\) is close to \\(b^\*\\), then the excess error will be small.  Experimental results show that BBF performs well in the unsupervised setting and often has small excess error compared to \\(\\vec{g}^\*\\).”

---

> > > ### Comment · Reviewer_nt5S · 2025-08-06
> > >
> > > Thanks for your detailed clarification on consistency. My concern still remains in that:
> > >
> > > * (1) The method requires the LF accuracy priors -- which to be set properly, would require labeled data to get accurate estimates (with high probability). While empirically the chosen Beta(4, 1) prior seems to work, the same could be said about Arachie, C. and Huang, B. [1] with a pre-specified error rate
> > >
> > > * (2) No empirical comparisons with [1] are provided
> > >
> > > As such, I choose to maintain my score.

---

> > > > ### Author Response · Authors · 2025-08-07
> > > >
> > > > Thank you for your consideration.  We ask you to reconsider your score as we now provide your requested experimental comparison and address your first concern.  Below are some extra rows (BBF MV10 will be used later).  Arachie and Huang's method is denoted CLL.  To initialize the LF errors, we used their proposed strategy: for all text datasets/datasets where the LF coverage (# of points each LF predicts on) was low, we set each LF to have error of \\(0.01\\).  For datasets where the LF coverage was high, we set the LFs errors to be \\(1/K\\).  CLL was run 10 times per dataset.
> > > >
> > > > Our results with CLL are consistent with [1].  That is, CLL often matches the performance of HyperLM.  We make a few observations:
> > > > 1. On IMDB, SMS, Yelp, TREC, SemEval, ChemProt, CLL matched the performance of BBF
> > > > 2. On YouTube, Commercial, Tennis it was beaten by BBF
> > > > 3. On CDR, it beat BBF
> > > >
> > > > For Commercial and Tennis, the LF errors were initialized to be \\(0.5\\) as \\(K=2\\) for because the LF coverage was high.  If instead, the LF errors are initialized to be \\(0.01\\), the resulting F1 scores are 84.85 and 81.37 respectively.  There, it beats BBF on Tennis, but is worse than BBF on Commercial.  We contend that BBF is an improvement.
> > > >
> > > > 			Dataset & IMDB & YouTube & SMS & CDR & Yelp & Commercial & Tennis & TREC & Semeval & ChemProt & AGNews \\
> > > > 			\midrule
> > > > 			CLL & $74.80$ & $84.79$ & \textbf{78.83} & $64.33$ & $77.80$ & $36.38$ & $42.31$ & $59.34$ & \textbf{84.21} & $52.37$ & $80.97$ \\
> > > > 			HyperLM & $75.29$ & \textbf{91.40} & \textbf{78.90} & $70.54$ & $78.73$ & $82.60$ & \textbf{84.21} & \textbf{60.02} & \textbf{84.21} & $52.32$ & $81.38$ \\
> > > >             \midrule
> > > > 			BBF & $74.67$ & $89.71$ & $78.77$ & $62.39$ & $77.67$ & $85.02$ & $79.87$ & $59.88$ & \textbf{84.21} & $52.47$ & $81.40$ \\
> > > > 			BBF MV10 & $74.67$ & $89.48$ & $78.77$ & $62.40$ & $77.61$ & $85.31$ & $79.89$ & $59.88$ & \textbf{84.21} & $52.63$ & $81.40$ \\
> > > >
> > > > ----------
> > > >
> > > > With respect to your first concern, we point to the one-coin Dawid-Skene (OCDS) model [2] with the EM algorithm.  There, we can use the label model in two ways.  We can either use labels to estimate the LF accuracies and provide those estimates to OCDS.  Or, we can estimate the accuracies by using the majority vote labeling as a proxy to the ground truth labeling.  We think it is reasonable to say that OCDS in the first scenario is semi-supervised (see [4] for an example of DS being used this way).  Moreover, it is in this semi-supervised scenario where one can achieve OCDS’ consistency result (outputting the best possible labeling).  (If the OCDS generative assumption holds, then one can also guarantee that EM will output a good labeling at convergence if the initialization of the LF accuracies is good, see [3]).  On the other hand, we think many would agree that OCDS in the second scenario is _unsupervised_.  I.e. being able to use labeled data does not preclude OCDS from being used in an unsupervised way.
> > > >
> > > > The same can be said about BBF.  One can use labeled data to estimate the LF accuracies and then set the Beta priors accordingly.  This would make it semi-supervised.  One can also set the Beta priors by using the majority vote labeling as a proxy to the ground truth.  We’ll explain by an example. If there are \\(N=1000\\) datapoints and the majority vote labeling is such that an LF gets 600 points “correct” and “400” wrong, one might initialize the Beta prior as Beta(6, 4) for BBF.  If one does Beta(600, 400), then BBF will perform poorly unless the empirical accuracy of that LF is \\(0.6\\).  I.e. a prior concentrated around something that’s not the LF’s empirical accuracy will incentivize the BBF to assign something other than the empirical accuracy to that LF, which is deleterious to performance.
> > > >
> > > > We performed experiments where we initialize BBF in this way, so the first Beta argument is \\(10 \\times\\) (LF accuracy from MV labeling), the second is \\(10 \\times \\)(1 - LF accuracy from MV).  The \\(10\\) was chosen arbitrarily to keep the Beta priors relatively flat.  The majority vote labeling was also used to instantiate the Dirichlet prior for class frequencies (and the rescaling was also done).  The results are added as a row in the above table as “BBF MV10” and we see the results are quite similar to the hyperparameter initialization proposed in the paper.
> > > >
> > > > Thus, we contend that BBF can be fully unsupervised, or as unsupervised as the Dawid-Skene model can conceivably be.
> > > >
> > > > [1] Wu et al., Learning Hyper Label Model for Programmatic Weak Supervision, ICLR 2023
> > > >
> > > > [2] Li, H. and Yu, B., Error Rate Bounds and Iterative Weighted Majority Voting for Crowdsourcing, ArXiV 2014
> > > >
> > > > [3] Balakrishnan, S., Wainwright, M. J., and Yu, B., Statistical Guarantees For the EM Algorithm: From Population to Sample-Based Analaysis, The Annals of Statistics 2017
> > > >
> > > > [4] Mazzetto, A., et al., Semi-Supervised Aggregation of Dependent Weak Supervision Sources With Performance Guarantees, AISTATS 2021

---

### Official Review · Reviewer_uvqR · 2025-07-01

**Clarity:** 1
**Significance:** 2
**Originality:** 3
**Rating:** 3
**Confidence:** 2

**Summary:**

This paper introduces the BBF method (for Bayesian Balsubramani-Freund), an unsupervised Bayesian label model for weak supervision, where inputs $x_i$ with unknown labels $y_i$ are annotated by multiple noisy label functions (LFs). It weights the various LF to create a posterior on the $y_i$, by first drawing class priors, sampling latents $(y_i)$, and compute accuracy posterior for each LF on these latents before combining them in some adversarial fashion.

The method comes with theoretical guarantees and offers decent empirical performance.
If I understand correctly, Theorem 5.2 states that, if the class prior is a Dirac on  the right class proportion and if the accuracy posterior for each LF concentrates to the right one, then this method find the optimal blend of the LFs (based on the KL divergence to the true labeling). Theorem 5.3 seems to offer some speed for this convergence, it seems to corresponds to how much the estimated accuracies of the LFs implied by the model (which is captured by $Ag^{bbf}$) matches the true ones ($b^*$).

**Questions:**

How would this paper potentially lead to a groundbreaking finding?

**Ethical Concerns:**

["NO or VERY MINOR ethics concerns only"]

**Final Justification:**

While I appreciate the authors effort to answer my concerns, I still have trouble to "get it" and understand why this paper matters. I do not doubt that this paper is technically solid, but motivations and clarity should be improved imho in order to meet the NeurIPS bar.

**Limitations:**

Yes

**Quality:**

2

**Strengths And Weaknesses:**

**Strength**:
- A method with theoretical guarantee and decent empirical performance.

**Weaknesses**:
- It was hard for me to follow the paper, understand the motivations of both the setting, and the various steps in the algorithm.
- The empirical results are decent but not groundbreaking
- I do not really know how "nice" the consistency and convergence results are, they seems to depend on quite restrictive assumptions. Basically, it is assumed that the priors having already done the heavy lifting. And the convergence result is more of a "calibration result" as per Bartlett et al. JASA 2006 (with this paper in mind, the consistency result can be viewed as a "Fisher consistency" result)

---

> ### Author Rebuttal · Authors · 2025-07-29
>
> Thank you for your time and effort in reading our paper and writing this review.
>
> With respect to the experimental results, we would like to provide more context/some alternate perspectives and discuss how they might inform future research.  We provide three perspectives and then discuss them together.
>
> **Bayesian**: Bayesian Balsubramani-Freund (BBF) shows that one does not need to create more complex generative processes to improve performance.  Out of the Bayesian label models we are aware of in the literature, FABLE [1] is the best.  BBF is quite competitive with FABLE and is often better despite the fact that it has a very simple model for labeling functions (LFs) and uses less information.  Specifically, FABLE models each LF with *multiple* confusion matrices (of size \\(K^2\\) each) and also uses the datapoint *features* \\(x_i\\).  Compare that to BBF which uses models an LF by its accuracy only and only uses the LF predictions \\(h^{(w)}(x_i)\\) and does *not* use the datapoint features.  (The other Bayesian models do not use datapoint features. iBCC [2] models each LF as a single confusion matrix, while EBCC [3] models each LF w/multiple confusion matrices.)
>
> **Adversarial**:  In the last row of Table 2, we see \\(\vec{g}^\*\\), the best approximation of \\(\vec{\eta}\\) in KL divergence.  We’ll take that as a proxy to the best possible F1 score/0-1 loss that can be achieved by BBF.  For many of the datasets, we see that even \\(\vec{g}^\*\\), the best case scenario for BBF, is still not good enough to beat Denoise.  In other words, the ceiling for BBF’s performance is too low if the goal is to always beat Denoise in performance. Thus, we would need to raise the ceiling,  This is done by modeling the LFs in a more complex way.   I.e. have more general constraints (and not just a constraint on an LF’s accuracy).  For example, Mazuelas et al. [4] provide a framework to accommodate more general constraints.  BBF is a good first step — being competitive with Denoise, a label model that is a neural network that uses the datapoint features \\(x_i\\) as part of its input, something BBF doesn’t have.
>
> **Improving labels models in general**: BBF can be thought of as a member of the class of label models that use the “HyperLM strategy” of selecting a labeling.  Specifically, it constructs a polytope of labelings then picks something inside of it.  BBF’s empirical performance, in our opinion, shows that it is a decent if not good representative of the “HyperLM strategy”.  This gives credence to the idea that analyzing BBF theoretically will help us get a formal grasp on what the “HyperLM” strategy is.  In our case, reducing it to a mix of Bayesian/adversarial label model.
>
> **Discussion**: One direction of research is to change the generative process of BBF to make it more complex.  One possible route to follow is to create analogues of iBCC, EBCC, and even FABLE.  (Mazuela’s [4] formulation the polytope of labelings is quite general and many other approaches can be taken.) The hope is that this will raise the ceiling of performance enough so that the resulting label model can exceed Denoise’s performance.  In a more abstract direction, we hope that our theoretical contributions will provide more insight on the “HyperLM strategy”, which in turn may lead to new label models that use the same strategy but have better performance than Denoise.   All in all, BBF’s experimental results provide another piece of evidence that there is still more work to be done in the scenario where label models are only given LF predictions and *not* datapoint features.
>
> ————
>
> With respect to consistency our consistency result, we would like to provide more context.   The majority of label models proposed in the Programmatic Weak Supervision setting are probabilistic in nature.  And as we are interested in adversarial label models, we wish to see them used in practice more.  However, one of the concerns for adversarial models is that they may be too pessimistic.  One line of argument one can provide is via consistency/convergence results, e.g. like from our paper or from An and Dasgupta [5].  Those results show that as the estimates for LF accuracies (and class frequencies) get more and more accurate, the outputs of BBF and Balsubramani-Freund (BF) improve.  Thus, anyone considering using either model knows that they do reasonable things.
>
> In particular, An and Dasgupta [5] use the consistency result for BF to show that there doesn’t exist a problem (a set of LF predictions and underlying label distribution (\\(\vec{\eta}\\)) where BF is always outperformed by the Dawid-Skene model [6].
>
> [1] Zhang, J., Song, L., Ratner, A., Leveraging Instance Features for Label Aggregation in Programmatic Weak Supervision, AISTATS 2023
>
> [2] Kim, H.-C. and Ghahramani, C., Bayesian Classifier Combination, AISTATS 2012
>
> [3] Li, Y., Rubinstein, B., Cohn, T., Exploiting Worker Correlation for Label Aggregation in Crowdsourcing, ICML 2019
>
> [4] Mazuelas, S., Romero, M., Grunwald, P., Minimax Risk Classifiers with 0-1 Loss, JMLR 2023
>
> [5] An, S. and Dasgupta, S., Convergence Behavior of an Adversarial Weak Supervision Method, UAI 2024
>
> [6] Dawid, A. P., Skene, A. M., Maximum Likelihood Estimation of Observer Error-Rates Using the EM Algorithm, Applied Statistics 1979

---

> > ### Comment · Reviewer_uvqR · 2025-08-04
> > **Thank you**
> >
> > Thank you for your detail answer.
> > I am still unclear about how groundbreaking this stream of research is, in particular when taking in consideration the “bitter lesson”.

---

> ### Author Response · Authors · 2025-08-05
>
> Thank you for your response.  To answer your question, we would like to once again reframe the discussion.  Our takeaway from the “bitter lesson” essay is that large amounts of compute power and a “good algorithm” to take advantage of such power is jointly sufficient to achieve “good performance” on your task.
>
> We think that our work broadly follows the suggestion presented by the author of "the bitter lesson" by taking advantage of the computational power available today while also being a “good algorithm”.   By algorithm, we take it to mean the label model specification and the way the parameters are learned, e.g. via gradient descent, mean field variation inference, etc.  To justify this, we first describe our work in the context of weighted majority vote methods.  Then, we discuss our work in a more general view of label models to reframe our previous response in the context of the “bitter lesson”.
>
> In the simplest case, one has majority vote, wherein LF is assigned weight \\(1\\) with minimal computation.  One-coin Dawid-Skene model (presented in [5] above) is more complex with each LF’s weight being a function of its accuracy.  An LF w/ accuracy \\(b\\in (0,1)\\) gets weight \\(\\log(\\frac{b}{1-b})\\) if there's \\(K=2\\) classes.  The LF accuracies can be estimated in an unsupervised way using EM, which uses more computational power but is still quite light.  BBF follows in the line of using more computational power to find weights for each LF.  (Lemma 5.1 shows that BBF is a weighted majority vote.)   For BBF, the weight is a result of a convex program being solved and to our knowledge doesn't have a simple functional form.  In a way, we are using the extra computational power (to solve a convex program) to overcome the added ambiguity of not having the form of the LF weights.  Indeed, our experiments show that BBF, is better than the previously proposed probabilistic methods, providing evidence that BBF is following in the progression proposed by "the bitter lesson".
>
> EBCC and Fable are both Bayesian methods that use complex generative models to model the label/LF prediction generation process and algorithmically estimate the posterior label distribution.  They take advantage of available compute power to run Mean Field Variation Inference for EBCC/the bespoke posterior estimation algorithm from [1] above.  I.e. these methods leverage the computation power while also being "good algorithms".  Now, if we consider EBCC/Fable to be "good algorithms" because of their empirical performance, then we'd like to argue that BBF also has a “good” algorithm.
>
> Taking a step back, we’d like to discuss HyperLM/Denoise to provide more context.  These methods are both both NNs trained on GPU(s), which arguably requires more resources than EBCC/Fable (and all label models other than WeaSEL, which itself is a NN).  Given the “bitter lesson”, it’s perhaps not surprising that HyperLM and Denoise are the top performers — they both take advantage of the computational power available and are “good algorithms”.  We want to emphasize that HyperLM is simpler conceptually compared to Denoise because it does _not_ use datapoint features.  This is important because HyperLM can compete with Denoise despite not having access to the datapoint features.  This leads us to believe that one ought to be able to achieve much better performance than Denoise given that HyperLM already gets close without datapoint features.  I.e. one avenue (that we brought up in our previous response) is that one could get a theoretical grasp of makes HyperLM good, then improve on that.
>
> Now, we believe there are two ways of viewing our theoretical/empirical contributions.  On the theoretical front, the impact (taking the “bitter lesson” into context) is more indirect than direct.  We believe the “groundbreaking” aspect of our research in this sense is its novelty in characterizing “HyperLM strategy” we mentioned before.  We hope that our work can and will be used to design a computationally intensive label model (neural network or otherwise) that exceeds the performance of HyperLM and Denoise.  I.e. be used in the creation of a new “good algorithm”.  One can also pull away from the “HyperLM strategy” and just take BBF’s approach at face value.  In this way, we have something that has characteristics of probabilistic (specifically Bayesian) and adversarial models but doesn’t cleanly fall in either camp.  To our knowledge, this type of model has not been proposed in the Programmatic Weak Supervision area before.  As in the previous paragraph, our experimental results show that BBF is competitive with the best PWS methods, which in the context of the “bitter lesson” leading us to believe that we have proposed a “good algorithm”.  To be extra explicit, BBF not only takes advantage of the computational resources available, but (we believe) is also a “good algorithm”.  I.e. it follows in the spirit of the approach proposed by the “bitter lesson” essay.

---

### Note · Authors · 2025-08-11

We thank the reviewers for their engagement and their suggestions to improve the paper.  We would like to reiterate some of the points made.

1. Reviewer uvqR's question about how groundbreaking our work is in the context of the "bitter lesson".

We see our work as following the progression observed in that essay.  Specifically, using computational power (in a smart way) to overcome the decrease of human input in the label model design.  One sees human input/bias in label model design in the one-coin Dawid-Skene (OCDS) model. [1].  That label model is a weighted majority vote where the size of the weight (for a labeling function or LF) is dependent on what the accuracy of the LF is.  I.e. the larger the accuracy, the larger the weight and vice versa.  We improve that method by *not* parameterizing the weight via LF accuracy.  (Our proposed method and OCDS are both weighted majority votes.)  While we require more computational power to compute the LF weights the experiments show that our method easily outperforms the general Dawid-Skene model.  I.e. we removed a source of human bias and supplemented it with more computation.

2. The final comments of Reviewer nt5S.

Their first concern is that the LF priors need to be set accurately (with labeled data) for our proposed method to have good empirical performance.  First, we found that the range of LF accuracies over all 11 datasets was \\(0.0366\\) to \\(1\\).  The mean of our chosen (LF accuracy) priors was \\(0.8\\), meaning that it was sometimes very far off from the real LF accuracy, yet our proposed method still did well.  We also addressed the concern that our initialization method was too ad hoc to be unsupervised.  There, we showed that our proposed method achieved the same performance while using the same initialization method as a bona fide unsupervised method.  I.e. using the majority vote labeling to estimate LF accuracies a la OCDS.

Their second concern is that we do not provide an experimental comparison to the work of Arachie and Huang [2].  In our most recent reply to them, the experimental comparison is provided and our proposed method is only worse than Arachie and Huang's method on one dataset (out of 11 total datasets).  Thus, we conclude that our proposed method is an improvement.

[1] Li, H. and Yu, B., Error Rate Bounds and Iterative Weighted Majority Voting for Crowdsourcing, ArXiV 2014

[2] Arachie, C. and Huang, B., Constrained labeling for weakly supervised learning, UAI 2021

---

### Decision · Program_Chairs · 2025-09-17

**Decision:**

Accept (poster)

**Comment:**

Claims and findings: This is a paper on weak supervision, an area that seeks to efficiently obtain high-quality labels from noisy weak sources. There has been a huge proliferation of label models (the models used to aggregate the votes from the sources), and the authors propose and analyze a new one with useful properties.

Strengths: The proposed label model is interesting. On the one hand, it is quite rich, while on the other it is amenable to theoretical analysis in a way that a lot of newer label models are not. In particular the results on the form of the posterior and the one on consistency are interesting; these resemble the more theoretically-flavored results from An and Dasgupta, but the authors consider more practical settings.

Weaknesses: The paper isn't super well-written and there's various clarity issues that have been spotted by the reviewers. However, all of this feels fixable for the camera ready.

Decision: While the paper needs to improve its writing, there is enough valuable content here to warrant accepting it.

Rebuttal discussion: The authors clarified many of the questions of the reviewers; with these clarifications the paper will be in good shape.